# Counterfactual-Augmented Importance Sampling for Semi-Offline Policy Evaluation

**Shengpu Tang, Jenna Wiens**
Computer Science & Engineering
University of Michigan, Ann Arbor, MI, USA
{tangsp,wiensj}@umich.edu

Reviewed on OpenReview: https://openreview.net/forum?id=dsH244r9fA

## Abstract

In applying reinforcement learning (RL) to high-stakes domains, quantitative and qualitative evaluation using observational data can help practitioners understand the generalization performance of new policies. However, this type of off-policy evaluation (OPE) is inherently limited since offline data may not reflect the distribution shifts resulting from the application of new policies. On the other hand, online evaluation by collecting rollouts according to the new policy is often infeasible, as deploying new policies in these domains can be unsafe. In this work, we propose a semi-offline evaluation framework as an intermediate step between offline and online evaluation, where human users provide annotations of unobserved counterfactual trajectories. While tempting to simply augment existing data with such annotations, we show that this naive approach can lead to biased results. Instead, we design a new family of OPE estimators based on importance sampling (IS) and a novel weighting scheme that incorporate counterfactual annotations without introducing additional bias. We analyze the theoretical properties of our approach, showing its potential to reduce both bias and variance compared to standard IS estimators. Our analyses reveal important practical considerations for handling biased, noisy, or missing annotations. In a series of proof-of-concept experiments involving bandits and a healthcare-inspired simulator, we demonstrate that our approach outperforms purely offline IS estimators and is robust to imperfect annotations. Our framework, combined with principled human-centered design of annotation solicitation, can enable the application of RL in high-stakes domains.

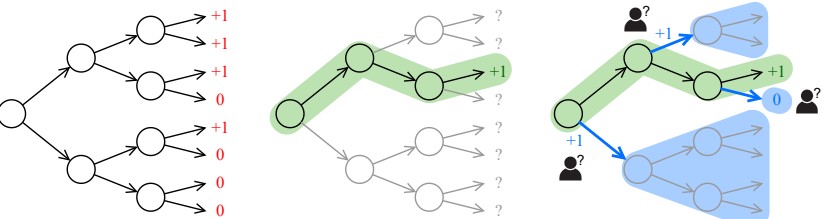

Figure 1: Left - The state transition diagram of a tree MDP with 3 steps and 2 actions. States are denoted by $\bigcirc$, actions are denoted by arrows $\{\nearrow, \searrow\}$, rewards are denoted in red and given only at terminal transitions. Center - The behavior policy takes a specific sequence of actions and leads to the factual trajectory, leaving the rest of the state-action space with poor support. Right - The counterfactual annotations provided by human annotators (indicated by 👤?) capture information (in this example, the terminal reward under any policy that takes $(\nearrow, \nearrow)$ for the second and third steps) about support-deficient regions of the state-action space not visited by the behavior policy.

37th Conference on Neural Information Processing Systems (NeurIPS 2023).

# 1 Introduction

Reinforcement learning (RL) has gained popularity in recent years for its ability to solve sequential decision-making problems in various domains [1–7]. Despite these successes, it remains challenging to deploy and use RL in highly consequential or safety-critical domains, such as healthcare, education, and public policy [8–12]. One of the major roadblocks that distinguishes RL-based systems from their supervised learning counterparts is evaluation.

Evaluation of supervised learning models often involves calculating prediction accuracy against a labeled test set [13]. In contrast, evaluation of RL policies is less straightforward and often involves interacting with the environment [2, 3, 14–19]. For domains that lack accurate simulators, this means deploying new policies in the real environment. For instance, in healthcare, online evaluation would require clinicians to follow RL recommendations in selecting treatments for real patients. While mathematically sound, this presents clear safety issues and potential disruptions to workflows. Therefore, most work in these areas has relied exclusively on retrospective evaluations using observational data [20–22], focusing on both quantitative and qualitative aspects. Quantitative evaluations make use of statistical off-policy evaluation (OPE) methods to account for the distribution shift resulting from the application of new policies [23–25]. Despite their wide use, OPE is fundamentally limited by the available offline data. In particular, past work has noted that unexpected bias and large variance [20] among other reasons make these approaches unreliable [22, 26]. On the other hand, qualitative evaluations typically aim to verify with domain experts whether the RL recommendations are reasonable, but are difficult to standardize and may be susceptible to confirmation bias [20].

In this work, we consider an intermediate step before prospective deployment that improves upon offline evaluation of RL policies. Specifically, we assume human domain experts can provide annotations of unobserved counterfactual trajectories that are small deviations of the observed trajectory (Figure 1), where each annotation is some summary of the expected outcomes of counterfactual trajectories. For example, in healthcare domains, such annotations may be obtained by asking clinicians what they think would happen to the patient if a different treatment were to be used. Intuitively, these counterfactual annotations can make up for regions of the state-action space with poor support in the offline dataset. However, as we demonstrate, simply adding the annotations as new trajectories to the offline dataset will change the state distribution and lead to biased results. Thus, we design a new OPE estimator based on importance sampling (IS) that incorporates both the offline factual data and counterfactual annotations without introducing additional bias. We analyze the theoretical properties of our proposed estimator, noting its advantages over standard IS. Specifically, our estimator requires a weaker condition on support to achieve unbiasedness and has the potential to reduce variance. Through a series of proof-of-concept experiments using toy problems and a healthcare-inspired simulator, we show the benefits of our approach in making use of counterfactual annotations to enable better evaluations of RL policies, even when annotations are biased, noisy, or missing. Our semi-offline evaluation framework represents an important step that complements offline evaluations by providing additional confidence in RL policies.

# 2 Problem Setup

We consider Markov decision processes (MDPs) defined by a tuple $\mathcal{M} = (\mathcal{S}, \mathcal{A}, P, R, d_1, \gamma, T)$, where $\mathcal{S}$ and $\mathcal{A}$ are the state and action spaces, $P : \mathcal{S} \times \mathcal{A} \to \Delta(\mathcal{S})$ and $R : \mathcal{S} \times \mathcal{A} \to \Delta(\mathbb{R})$ are the transition and reward functions, $d_1 \in \Delta(\mathcal{S})$ is the initial state distribution, $\gamma \in [0, 1]$ is the discount factor, $T \in \mathbb{Z}^+$ is the fixed horizon. $p(s'|s, a)$ denotes the probability density function of $P$, and $\bar{R}(s, a)$ denotes the expected reward. A policy $\pi : \mathcal{S} \to \Delta(\mathcal{A})$ specifies a mapping from each state to a probability distribution over actions. A $T$-step trajectory following policy $\pi$ is denoted by $\tau = [(s_t, a_t, r_t)]_{t=1}^T$ where $s_1 \sim d_1, a_t \sim \pi(s_t), r_t \sim R(s_t, a_t), s_{t+1} \sim p(s_t, a_t)$. Here, $a \sim \pi(s)$ is short for $a \sim \pi(\cdot|s)$ and $s' \sim p(s, a)$ for $s' \sim p(\cdot|s, a)$. Let $J = \sum_{t=1}^T \gamma^{t-1} r_t$ denote the return of the trajectory, which is the discounted sum of rewards. The value of a policy $\pi$ is the expected return, defined as $v(\pi) = \mathbb{E}_\pi[J]$. The value function of policy $\pi$, denoted by $V^\pi : \mathcal{S} \to \mathbb{R}$, maps each state to the expected return starting from that state following policy $\pi$. Similarly, the action-value function (i.e., the Q-function), $Q^\pi : \mathcal{S} \times \mathcal{A} \to \mathbb{R}$, is defined by further restricting the action taken from the starting state. Formally, $V^\pi(s) = \mathbb{E}_\pi[J|s_1 = s]$, and $Q^\pi(s, a) = \mathbb{E}_\pi[J|s_1 = s, a_1 = a]$. We also consider value functions at specific horizons: $V_{t:T}^\pi(s) = \mathbb{E}_\pi[\sum_{t'=t}^T \gamma^{t'-1} r_{t'}|s_t = s]$, and $Q_{t:T}^\pi(s, a) = \mathbb{E}_\pi[\sum_{t'=t}^T \gamma^{t'-1} r_{t'}|s_t = s, a_t = a]$. Throughout the paper we also consider the non-sequential, bandit setting with horizon $T = 1$. In this case, a "trajectory" (or, a sample) is denoted by $\tau = (s, a, r)$ where we omit the time step subscript.

Our goal is to estimate $v(\pi_e)$, the value of an evaluation policy $\pi_e$, given data that were previously collected by some behavior policy $\pi_b$ in the same environment defined by $\mathcal{M}$. Let $\mathcal{D} = \{\tau^{(i)}\}_{i=1}^N$ denote the dataset containing $N$ independent trajectories drawn according to $\pi_b$ and $\mathcal{M}$.

**OPE.** The typical approach to this problem relies on off-policy evaluation (OPE). Importance sampling (IS) is a common OPE approach that reweights samples based on how likely they are to occur under $\pi_e$ relative to $\pi_b$. Given a trajectory $\tau$, the 1-step and cumulative IS ratios are defined as $\rho_t = \frac{\pi_e(a_t|s_t)}{\pi_b(a_t|s_t)}$ and $\rho_{1:t} = \prod_{t'=1}^t \rho_{t'}$. The per-decision IS estimator, $\hat{v}^{\text{PDIS}} = \sum_{t=1}^T \rho_{1:t}\gamma^{t-1}r_t$, is an unbiased estimator of $v(\pi_e)$ [27, 28]. We also consider its recursive definition: $\hat{v}^{\text{PDIS}} = v_T$ where $v_0 = 0$, $v_{T-t+1} = \rho_t(r_t + \gamma v_{T-t})$. In this paper, we discuss the properties of IS-based estimators over a single trajectory; our results naturally generalize to dataset $\mathcal{D}$ containing $N$ trajectories where the final estimator is the average over trajectories. For the bandit setting, we refer to PDIS simply as the IS estimator, $\hat{v}^{\text{IS}} = \rho r = \frac{\pi_e(a|s)}{\pi_b(a|s)}r$.

**Counterfactual Annotations.** In addition to the offline dataset $\mathcal{D}$, our semi-offline framework assumes access to accompanying counterfactual annotations. To introduce the notation, we start with the non-sequential, bandit setting where $T = 1$, dropping the time step subscripts. Given a factual sample $\tau = (s, a, r)$, let $c^{\tilde{a}} \in \{0, 1\}$ be a binary indicator for whether the counterfactual action $\tilde{a} \in \mathcal{A} \setminus \{a\}$ is associated with an annotation, and let the annotation be $g^{\tilde{a}} \in \mathbb{R}$. We use $G : \mathcal{S} \times \mathcal{A} \to \Delta(\mathbb{R})$ to denote the annotation function such that $g^{\tilde{a}} \sim G(s, \tilde{a})$. A counterfactual-augmented sample $\tau^+ = (\tau, \boldsymbol{g})$ consists of the factual sample $\tau$ and counterfactual annotations $\boldsymbol{g} = \{g^{\tilde{a}} : c^{\tilde{a}} = 1\}$, where each $g^{\tilde{a}} \sim G(s, \tilde{a})$. Intuitively, a "good" annotation should reflect the scenario where the counterfactual action $\tilde{a}$ is taken and the reward $g^{\tilde{a}} \sim R(s, \tilde{a})$ is observed.

**Assumption 1** (Perfect annotation, bandit). $\mathbb{E}_{g \sim G(s,a)}[g] = \bar{R}(s, a), \forall s \in \mathcal{S}, a \in \mathcal{A}$.

For the sequential setting, we define the corresponding notation with time step subscripts: for $(s_t, a_t, r_t)$ occurring at step $t$ of trajectory $\tau$, we define counterfactual indicators $c_t^{\tilde{a}}$ for $\tilde{a} \in \mathcal{A} \setminus \{a_t\}$ and annotations $\boldsymbol{g}_t = \{g_t^{\tilde{a}} : c_t^{\tilde{a}} = 1\}$. Figure 2 provides an example trajectory with counterfactual annotations. Here, each $g_t^{\tilde{a}} \sim G_t(s_t, \tilde{a})$ is drawn from the horizon-$t$ annotation function $G_t$. While the general notion of counterfactual annotations could be used to capture different information (e.g., the instantaneous reward of the counterfactual action, $R(s, \tilde{a})$), in this work, we study a specific version that allows us to extend the theory of the bandit setting. Specifically, the annotation for counterfactual action $\tilde{a}$ summarizes the annotator's belief of the expected future return (sum

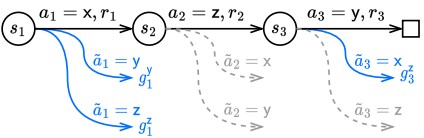

Figure 2: A trajectory augmented with counterfactual annotations, where the action space is $\mathcal{A} = \{\mathsf{x}, \mathsf{y}, \mathsf{z}\}$. The factual trajectory $\tau$ is shown in black. Solid blue arrows indicate the counterfactual annotations were queried and obtained; dashed gray arrows indicate the annotations are not available. Each transition arrow is labeled with (action, value), where value is either an observed immediate reward or a counterfactual annotation.

of rewards) in the remaining $T - t + 1$ steps after taking the counterfactual action $\tilde{a}$ from state $s_t$, and then following the evaluation policy $\pi_e$. In other words, the annotation plays the same role as the Q-function. This leads to a more refined assumption on the horizon-specific annotation function $G_t$.

**Assumption 2** (Perfect annotation, MDP). $\mathbb{E}_{g \sim G_t(s,a)}[g] = Q_{t:T}^{\pi_e}(s, a), \forall s \in \mathcal{S}, a \in \mathcal{A}$.

Under Assumption 2, if we obtained *infinitely many* annotations for *all* initial states and *all* actions, then evaluation becomes trivial (we essentially recover the Q-function of all initial states). However, we consider the non-asymptotic regime where not every annotation is available, as certain annotations might be difficult to obtain. For example, annotating initial states requires reasoning about the full horizon $T$. Furthermore, since this is a rather strong assumption (we need different annotations for each $\pi_e$), later we explore a relaxation where the annotations reflect the behavior policy $\pi_b$ instead.

## 3   Methods

To motivate our approach, we begin with a didactic bandit example to illustrate how the naive incorporation of counterfactual annotations can yield biased estimates. In order to address this issue, we propose a modification of IS estimators that reweights the factual data and counterfactual annotations. We formally describe how this idea applies to IS (in the bandit setting) and PDIS (in the sequential RL setting), giving rise to a family of semi-offline counterfactual-augmented IS estimators. We study the impact of different assumptions regarding the annotations on the performance of our proposed estimators both theoretically (Section 4) and empirically (Section 5).

## 3.1 Intuition

Consider a one-step bandit (Figure 3a) with two states $\{s_1, s_2\}$ (drawn with equal probability) and two actions, up ($\nearrow$) and down ($\searrow$). The reward from $s_1$ is $+1$ and from $s_2$ is $0$ (i.e., rewards do not depend on the action), meaning all policies have an expected reward of $0.5$. Suppose the behavior policy always selects $\nearrow$, generating a dataset with poor support for policies that assign nonzero probabilities to $\searrow$ (Figure 3b). Now suppose we also have access to human-provided annotations of counterfactual actions, but not all counterfactual annotations are available (either because they were never queried or the users declined to provide annotations). In our example (Figure 3c), one annotation is collected for the counterfactual action $\searrow$ at state $s_1$, indicating that the human annotator believes the reward for taking action $\searrow$ from state $s_1$ is $+1$ (which is the true reward). To make use of this information, one might be tempted to add the counterfactual annotation as a new sample. The augmented dataset (Figure 3d) would allow us to evaluate policies (e.g., using IS) that assign non-zero probabilities to $\searrow$ in state $s_1$. While seemingly plausible, this naive approach inadvertently changes the state distribution and results in a dataset inconsistent with the original problem (it looks like state $s_1$ is seen more often than reality). A quick calculation reveals that applying IS to this unweighted augmented dataset gives a biased estimate of a new policy as $2/3$ instead of $0.5$ (see Appendix B). To address this issue, in Section 3.2 we propose a new reweighting procedure that maintains the state distribution of the original dataset while incorporating counterfactual annotations.

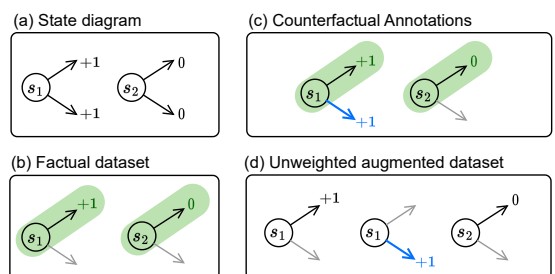

(a) State diagram  (c) Counterfactual Annotations
(b) Factual dataset  (d) Unweighted augmented dataset

Figure 3: (a) The state diagram of a bandit problem with two states and two actions. (b) A factual dataset containing two samples. (c) The factual samples augmented with counterfactual annotations. (d) The (unweighted) augmented dataset constructed from factual samples and counterfactual annotations. Compared to the original factual dataset, the relative frequency of $s_1$ vs $s_2$ has changed from $1 : 1$ to $2 : 1$.

## 3.2 Augmenting IS Estimators with Counterfactual Annotations

To avoid the bias issue described in Section 3.1, informally, we want to split the contribution of each sample between the factual data and counterfactual annotations. Given a factual sample $(s, a, r)$ and the associated counterfactual annotations $\boldsymbol{g}$, let $\boldsymbol{w} = \{w^a\} \cup \{w^{\tilde{a}} : \tilde{a} \in \mathcal{A} \setminus \{a\}\}$ be a set of user-defined non-negative weights that satisfy $w^a + \sum_{\tilde{a} \in \mathcal{A} \setminus \{a\}} w^{\tilde{a}} = 1$. These weights specify how much we want the estimator to "listen" to the counterfactual annotations ($w^{\tilde{a}}$) relative to the factual data ($w^a$). We restrict $w^{\tilde{a}} = 0$ when $c^{\tilde{a}} = 0$, i.e., non-zero weight is only allowed when the annotation is available. In general, one may assign different weights for each occurrence of $(s, a)$ (e.g., the counterfactual annotation is obtained for one instance but missing for another); let $\bar{W}(\tilde{a}|s, a) = \mathbb{E}[w^{\tilde{a}}]$ denote the average weight assigned to $\tilde{a}$ when the factual data is $(s, a)$ (see example in Appendix B.1). After reweighting, the state distribution is maintained (since the weights associated with each sample sum to 1) but the state-conditional action distributions have changed; this "weighted" augmented dataset can be seen as if it was generated using a different behavior policy.

**Definition 1** (Augmented behavior policy).
$$\pi_{b^+}(a|s) = \bar{W}(a|s, a)\pi_b(a|s) + \sum_{\check{a} \in \mathcal{A} \setminus \{a\}} \bar{W}(a|s, \check{a})\pi_b(\check{a}|s).$$

Here, $\pi_{b^+}(a|s)$ represents the probability that information about action $a$ is observed for state $s$ (similar to how an "average policy" may be defined for multiple behavior policies [29]), either as a factual action in the dataset, or as an annotated counterfactual action when some other action $\check{a}$ is the factual action. Next, we define our proposed estimators (for bandits) based on IS.

**Definition 2** (Counterfactual-augmented IS). Given a counterfactual-augmented sample $\tau^+ = (\tau, \boldsymbol{g})$ and weights $\boldsymbol{w} = \{w^{\tilde{a}} : \tilde{a} \in \mathcal{A}\}$, where $\tau = (s, a, r)$, $\boldsymbol{g} = \{g^{\tilde{a}} : c^{\tilde{a}} = 1\}$, the C-IS estimator is $\hat{v}^{\text{C-IS}} = w^a \rho^a r + \sum_{\tilde{a} \in \mathcal{A} \setminus \{a\}} w^{\tilde{a}} \rho^{\tilde{a}} g^{\tilde{a}}$, where $\rho^{\tilde{a}} = \frac{\pi_e(\tilde{a}|s)}{\pi_{b^+}(\tilde{a}|s)}$ for each $\tilde{a} \in \mathcal{A}$.

The C-IS estimator is a weighted convex combination of the factual IS estimate $\rho^a r$ and the counterfactual IS estimates $\rho^{\tilde{a}} g^{\tilde{a}}$ for all counterfactual actions $\tilde{a} \in \mathcal{A} \setminus \{a\}$. We also study a special case where all annotations are available and the weights are split equally among actions, such that $w^a = w^{\tilde{a}} = 1/|\mathcal{A}|$. Then, $\pi_{b^+}$ becomes the uniformly random policy, and after substituting into Definition 2, we obtain the following estimator.

**Definition 3** (C-IS with equal weights). Given a counterfactual-augmented sample $\tau^+ = (\tau, \boldsymbol{g})$, the C*-IS estimator is $\hat{v}^{\text{C*-IS}} = \pi_e(a|s)r + \sum_{\tilde{a} \in \mathcal{A} \setminus \{a\}} \pi_e(\tilde{a}|s)g^{\tilde{a}}$.

*Remark.* Definition 3 provides an alternative interpretation of the estimator when using equal weights: if Assumption 1 holds (i.e., the annotation function $G$ is the true reward function $R$), we effectively observe both the factual and counterfactual rewards from $R$. Then, we can directly use the definition of the value function to calculate the expected reward under $\pi_e$ using the action probabilities $\pi_e(\cdot|s)$.

For the sequential setting, given a trajectory with $T$ steps, we define the collection of weights over all time steps, $\boldsymbol{w} = \{w_t^{a_t} : t = 1...T\} \cup \{w_t^{\tilde{a}} : \tilde{a} \in \mathcal{A} \setminus \{a_t\}, t = 1...T\}$. The augmented behavior policy $\pi_{b^+}$ is similarly defined (see Definition 1). By extending the recursive definition of PDIS, we obtain the following two estimators (assuming either arbitrary weights or equal weights).

**Definition 4** (Counterfactual-augmented PDIS). Given a counterfactual-augmented trajectory $\tau^+ = (\tau, \boldsymbol{g})$ and weights $\boldsymbol{w}$ as defined above, where $\tau = [(s_t, a_t, r_t)]_{t=1}^T$, $\boldsymbol{g} = \{g_t^{\tilde{a}} : c_t^{\tilde{a}} = 1\}$, the C-PDIS estimator is $\hat{v}^{\text{C-PDIS}} = v_T$, with $v_T$ defined recursively as $v_0 = 0$, $v_{T-t+1} = w_t^{a_t} \rho_t^{a_t}(r_t + \gamma v_{T-t}) + \sum_{\tilde{a} \in \mathcal{A} \setminus \{a_t\}} w_t^{\tilde{a}} \rho_t^{\tilde{a}} g_t^{\tilde{a}}$ for $t = T...1$, where $\rho_t^{\tilde{a}} = \frac{\pi_e(\tilde{a}|s_t)}{\pi_{b^+}(\tilde{a}|s_t)}$ for each $\tilde{a} \in \mathcal{A}$.

**Definition 5** (C-PDIS with equal weights). Given a counterfactual-augmented trajectory $\tau^+ = (\tau, \boldsymbol{g})$, the C*-PDIS estimator is $\hat{v}^{\text{C*-PDIS}} = v_T$, with $v_T$ defined recursively as $v_0 = 0$, $v_{T-t+1} = \pi_e(a_t|s_t)(r_t + \gamma v_{T-t}) + \sum_{\tilde{a} \in \mathcal{A} \setminus \{a_t\}} \pi_e(\tilde{a}|s_t)g_t^{\tilde{a}}$ for $t = T...1$.

Next, we study the theoretical properties of our proposed estimators, relating their OPE performance (in terms of bias and variance) to assumptions on counterfactual annotations and the offline dataset.

## 4 Theoretical Analyses

We first present results for the bandit setting, where we study and compare the properties of the C-IS estimator with standard IS in terms of bias, variance, and the assumptions required, highlighting scenarios where bias and variance reduction is guaranteed. We then show how these results generalize to C-PDIS in the sequential RL setting. Finally, we discuss practical implications of the theoretical results. Full derivations are in Appendix C.

To begin, we review existing results for IS. Recall the following assumption of common support.

**Assumption 3** (Common support). $\pi_e(a|s) > 0 \rightarrow \pi_b(a|s) > 0, \forall s \in \mathcal{S}, a \in \mathcal{A}$.

If Assumption 3 holds, IS is unbiased (i.e., $\mathbb{E}_\tau[\hat{v}^{\text{IS}}] = v(\pi_e)$), and its variance is [27]:

$$\mathbb{V}[\hat{v}^{\text{IS}}] = \mathbb{V}_{s \sim d_1}[V^{\pi_e}(s)] + \mathbb{E}_{s \sim d_1}\left[\mathbb{V}_{a \sim \pi_b(s)}[\rho(a|s)\bar{R}(s,a)]\right] + \mathbb{E}_{s \sim d_1}\left[\mathbb{E}_{a \sim \pi_b(s)}[\rho(a|s)^2 \sigma_R(s,a)^2]\right] (1)$$

where $\sigma_R(s,a)^2 = \mathbb{V}_{r \sim R(s,a)}[r]$ is the variance associated with the reward function $R(s,a)$. The first term reflects the inherent randomness from the state distribution not related to importance sampling. The second term reflects the randomness in the behavior policy, whereas the third term reflects the randomness in rewards; these two terms are affected by the distribution of importance ratios $\rho(a|s)$. When Assumption 3 is not satisfied, the IS estimator is biased [30], where the bias is related to actions with no support: $\text{Bias}[\hat{v}^{\text{IS}}] = \mathbb{E}[\hat{v}^{\text{IS}}] - v(\pi_e) = \mathbb{E}_{s \sim d_1}\left[-\sum_{a \in \mathcal{U}(s, \pi_b)} \pi_e(a|s)\bar{R}(s,a)\right]$, with $\mathcal{U}(s, \pi_b) = \{a : \pi_b(a|s) = 0\}$ denoting the set of unsupported actions.

Intuitively, when Assumption 3 does not hold, the C-IS estimator can make use of information from the counterfactual annotations for unsupported actions, thereby reducing bias compared to IS (Section 4.1). For cases when IS is already unbiased, counterfactual annotations play the role of additional data and should help further reduce variance (Section 4.2).

### 4.1 Bias Analyses for C-IS

To formalize the effect of counterfactual annotations on support, we state the following assumption.

**Assumption 4** (Common support with annotations). $\pi_e(a|s) > 0 \rightarrow \pi_{b^+}(a|s) > 0, \forall s \in \mathcal{S}, a \in \mathcal{A}$.

Assumption 4 is a weaker version of Assumption 3, because $\pi_{b^+}(a|s) > 0$ requires either $\bar{W}(a|s,a)\pi_b(a|s) > 0$ (same as Assumption 3, assuming $\bar{W}(a|s,a) \neq 0$) or $\bar{W}(a|s,\check{a})\pi_b(\check{a}|s) > 0$ for at least some $\check{a} \in \mathcal{A} \setminus \{a\}$. In other words, information about action $a$ can be from either a factual sample or counterfactual annotations (recall Definition 1). Next, we state the main results for the bias of C-IS (unless specified otherwise, expectations are taken with respect to $\mathbb{E}_{\tau^+, \boldsymbol{w}}$). These results hold for any nonzero $\boldsymbol{w}$ and directly generalize to the special case of C*-IS where the weights are $1/|\mathcal{A}|$.

**Theorem 1** (Unbiasedness of C-IS). *In the bandit setting, when both Assumptions 1 and 4 hold, the C-IS estimator is unbiased, $\mathbb{E}[\hat{v}^{\text{C-IS}}] = v(\pi_e)$.*

**Proposition 2** (Bias of C-IS due to support). *When Assumption 1 holds but Assumption 4 is violated,* $\text{Bias}[\hat{v}^{\text{C-IS}}] = \mathbb{E}_{s \sim d_1}\big[-\sum_{a \in \mathcal{U}(s, \pi_{b^+})} \pi_e(a|s)\bar{R}(s,a)\big]$ *where* $\mathcal{U}(s, \pi_{b^+}) = \{a : \pi_{b^+}(a|s) = 0\}$ *are unsupported actions in the counterfactual-augmented dataset.*

**Proposition 3** (Bias of C-IS due to imperfect annotations). *When Assumption 4 holds but Assumption 1 is violated,* $\text{Bias}[\hat{v}^{\text{C-IS}}] = \mathbb{E}_{s \sim d_1} \mathbb{E}_{a \sim \pi_e(s)}\big[\delta_W(s,a)\,\epsilon_G(s,a)\big]$*, where we measure violation of Assumption 1 as* $\epsilon_G(s,a) = \mathbb{E}_{g \sim G(s,a)}[g] - \bar{R}(s,a)$*, and* $\delta_W(s,a) = \big(1 - \frac{\bar{W}(a|s,a)\pi_b(a|s)}{\pi_{b^+}(a|s)}\big)$*.*

**Proposition 4** (A sufficient condition for bias reduction). *If Assumption 1 holds (but Assumption 4 is violated),* $\bar{R}(s,a) \geq 0$ *for all* $s \in \mathcal{S}, a \in \mathcal{A}$*, and there exists* $(s,a)$ *such that* $\pi_b(a|s) = 0$*,* $\pi_{b^+}(a|s) > 0$*,* $\pi_e(a|s) > 0$*,* $\bar{R}(s,a) > 0$*, then* $|\text{Bias}[\hat{v}^{\text{C-IS}}]| < |\text{Bias}[\hat{v}^{\text{IS}}]|$*.*

There are two sources of bias for C-IS: missing annotations contribute to the bias as the rewards of unsupported actions (Proposition 2), whereas imperfect annotations contribute to the bias as the annotation error over supported actions (Proposition 3). If both assumptions are violated, the resulting bias is the combination of the two (see Appendix C). If both assumptions hold, C-IS is unbiased (Theorem 1). Even when not all counterfactual annotations are collected (Assumption 4 is violated), C-IS can evaluate more policies without bias (assuming perfect annotations), because there is a larger space of policies "supported" by the counterfactual-augmented dataset. In particular, if there is at least one counterfactual annotation for an action with no support in the factual data, C-IS has less bias than IS (under mild conditions, Proposition 4). Lastly, we note the a useful corollary of Theorem 1.

**Corollary 5** (Expectation of augmented importance ratios). *Let* $\rho_W^+ = w^a \rho^a + \sum_{\tilde{a} \in \mathcal{A} \backslash \{a\}} w^{\tilde{a}} \rho^{\tilde{a}}$ *given* $\tau$ *and* $w$*. Under Assumption 4,* $\mathbb{E}[\rho_W^+] = 1$*.*

*Remark.* Corollary 5 suggests that for each sample, $\rho_W^+$ plays a similar role as the standard importance ratio $\rho$ in IS, which may be used for calculating the effective sample size (ESS) [31]. Naturally, we can also create a weighted version of our proposed estimators (e.g., C-WIS), with the normalization factor defined using $\rho_W^+$.

## 4.2 Variance Analyses for C-IS

Compared to the bias analyses above, the variance of C-IS has a more involved dependence on weights $w$ as well as the variance of the annotation function, $\sigma_G(s,a)^2 = \mathbb{V}_{g \sim G(s,a)}[g]$. For clarity, we defer the full derivations to Appendix C.3; here, we present results for C*-IS where the weights are all set to $1/|\mathcal{A}|$ and the annotation function has the same variance as the reward function.

**Theorem 6** (Variance of C*-IS). *Assuming* $\sigma_G(s,a)^2 = \sigma_R(s,a)^2$*, under Assumptions 1 and 4,*

$$\mathbb{V}[\hat{v}^{\text{C*-IS}}] = \mathbb{V}_{s \sim d_1}[V^{\pi_e}(s)] + \mathbb{E}_{s \sim d_1} \mathbb{E}_{a \sim \pi_b(s)}\big[\pi_b(a|s)\,\rho(a|s)^2\,\sigma_R(s,a)^2\big] \tag{2}$$

*where* $\rho(a|s) = \frac{\pi_e(a|s)}{\pi_b(a|s)}$ *is the importance ratio under the original behavior policy.*

**Proposition 7** (Variance Reduction of C*-IS). *Under the premise of Theorem 6,* $\mathbb{V}[\hat{v}^{\text{C*-IS}}] \leq \mathbb{V}[\hat{v}^{\text{IS}}]$*.*

Comparing Eqn. (2) with the three terms of the variance decomposition of IS in Eqn. (1), we note that the first term $\mathbb{V}_{s \sim d_1}[V^{\pi_e}(s)]$ is identical, the dropped second term (of Eqn. (1)) is a non-negative variance term, and the third term is scaled by a factor $\pi_b(a|s) \leq 1$ (for each instantiation of the expression inside the expectation), leading to a guaranteed variance reduction (Proposition 7). We derive the full variance decomposition for C-IS in Theorem 13. Unlike C*-IS, variance reduction is not guaranteed for C-IS. This is due to additional non-negative terms that depend on the variance/covariance of weights $w$ and terms that depend on the difference in variance between annotations and rewards $\sigma_G(s,a)^2 - \sigma_R(s,a)^2$ (could be positive or negative); these terms all vanish to zero in the case of C*-IS where weights are constant ($1/|\mathcal{A}|$) and $\sigma_G(s,a)^2 = \sigma_R(s,a)^2$.

## 4.3 Extensions to C-PDIS

We note that the corresponding results in the bandit setting can be derived for the MDP setting using an induction-style proof, with similar interpretations of the factors that contribute to reduced bias and reduced variance when compared to standard PDIS. Below, we briefly demonstrate how the unbiasedness result (Theorem 1) extends to the MDP setting. We further explore the sequential RL setting in the empirical experiments.

**Theorem 8** (Unbiasedness of C-PDIS). *In the MDP setting, when both [Assumptions 2 and 4](#) hold, the C-PDIS estimator is unbiased, $\mathbb{E}[\hat{v}^{\text{C-PDIS}}] = v(\pi_e)$.*

*Proof Sketch.* Here, we explain the intuition behind the proof for C*-PDIS, which proceeds via a backward induction ([Figure 4](#)). In the recursive definition ([Definition 4](#)), we aim to show that every $v_{T-t+1}$ is an unbiased estimator of the horizon-$t$ value function. At each horizon $t$, we can view the problem as a one-step bandit problem (reducing the estimator to C*-IS), where the factual action leads to a factual trajectory and the counterfactual action(s) leads to the counterfactual annotation(s), both of which are used to construct unbiased estimates of horizon-$t$ Q-values (for $a_t$ and $\tilde{a}$, respectively). In the end, the estimates from the two branches are combined according to $\pi_e$, resulting in the correct expectation of the horizon-$t$ value of state $s$. See full proof for C-PDIS in [Appendix C.4](#). $\qquad\square$

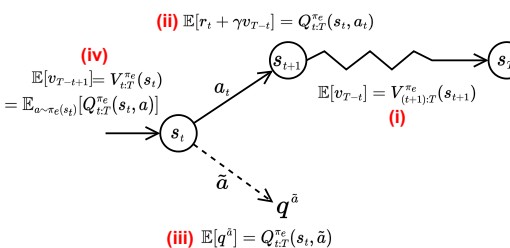

Figure 4: Illustration of proof idea. (i) The factual estimate $v_{T-t}$ provides an unbiased estimate of the horizon-$(t+1)$ value of state $s_{t+1}$. (ii) When combined with the factual reward $r_t$, we obtain an unbiased estimate of the horizon-$t$ Q-value of $(s_t, a_t)$. (iii) By assumption, the counterfactual annotations provide unbiased estimates of the horizon-$t$ Q-value of $(s_t, \tilde{a})$. (iv) The factual and counterfactual estimates are combined using $\pi_e$. For clarity, we omit details about state distributions here (see appendix).

## 4.4 Practical Implications

So far, we have been focusing on the theoretical framework for incorporating counterfactual annotations into OPE. However, the actual implementation of this approach poses several practical challenges. We believe this underscores the fact that this is a rich area for research with many potential directions. In this section, we address several real-world scenarios that do not adhere to the theoretical assumptions – specifically, when annotations are biased, noisy, or missing.

**Correcting annotation bias.** Comparing [Assumptions 1](#) and [2](#), we note an important distinction between the bandit setting and the sequential RL setting. For bandits, we simply want $G$ to mimic the reward model $R$. In contrast, for the RL setting, $G_t$ should ideally mimic the Q-function of the evaluation policy $Q_{t:T}^{\pi_e}$. Such annotations can be difficult if not impossible to obtain in practice (e.g., for healthcare, it would require asking clinicians to reason about a *sequence* of counterfactual actions and predict the outcome). Thus, we additionally consider a relaxation of [Assumption 2](#) where instead $\mathbb{E}_{g \sim G_t(s,a)}[g] = Q_{t:T}^{\pi_b}$ reflecting expected return under the behavior policy (which is more likely in practice). This annotation bias $\epsilon_G = Q_{t:T}^{\pi_b} - Q_{t:T}^{\pi_e}$ results in a biased estimator ([Proposition 3](#)). To correct this bias, we suggest first estimating the annotation bias function $\hat{\epsilon}_G(s, a)$ using an approximate MDP built from offline data, and then mapping each annotation as $\hat{g}^{\tilde{a}} = g^{\tilde{a}} - \hat{\epsilon}_G(s, \tilde{a})$ (see [Appendix D.1](#) for details). In [Section 5.2](#), we empirically measure the impact of this alternative assumption and the bias-correction procedure (which we denote as $Q^{\pi_b} \mapsto \hat{Q}^{\pi_e}$) on OPE performance.

**Reweighting noisy annotations.** So long as [Assumption 1](#) or [2](#) is satisfied, the noise (i.e., variance) of the annotations does not affect unbiasedness; however, as shown in [Theorem 13](#), the variance of our proposed estimators directly depends on how noisy the annotations are. Intuitively, if the annotation variance is smaller than the reward variance, we want the final estimate to "listen" more to the annotations and less to the factual data (and vice versa). We empirically explore the impact of annotation noise in both [Section 5](#) and [Appendix E](#), noting that while using equal weights (as in C*-IS) outperforms standard IS, adjusting weights based on the relative magnitudes of $\sigma_R^2$ and $\sigma_G^2$ can further improve OPE performance.

**Imputing missing annotations.** For many real-world domains, it is unlikely that we can obtain an annotation for every counterfactual action at every time step (the total number of annotations needed is $(|\mathcal{A}| - 1)NT$). While desirable to use equal weights as in C*-IS (and C*-PDIS) due to its variance reduction guarantee, this is not possible if some annotations are missing (in which case the factual data must have $w^a = 1$), and this can actually lead to higher variance (see [Appendix D.2](#) for an example). To alleviate this variance increase, we suggest estimating an annotation model $\hat{G}$ from available annotations and using it to impute the missing annotations (see [Appendix D.2](#)). Although $\hat{G}$ may be a biased estimator of $G$ (due to annotation noise) and introduce additional bias to the final estimate, we empirically observe a favorable bias-variance trade-off ([Section 5.2](#)).

In general, one would expect more counterfactual annotations to reduce both bias and variance; at the same time, these annotations may be imperfect (biased or noisy) and directly increase bias or variance. Our analyses in Section 4 and Appendix C show that both bias and variance depend on weights $w$, suggesting the weighting scheme as a key mechanism to achieve optimal bias and variance. In Appendix D.3 we explore an analytical approach for solving the variance-minimizing weighting scheme and note the solution is highly non-trivial. We empirically explore the impact of different weights in Appendix E.1 and note that using equal weights (as in C*-IS) is a promising heuristic, since it achieves good performance in most settings. We believe that optimizing the weights can further improve OPE performance and is an interesting direction for future work.

## 5 Experiments

First, through a suite of simple bandit problems, we verify the theoretical properties of C-IS. Then, we apply our approach to a healthcare-inspired RL simulation domain, where we compare the performance of our proposed approach, C-PDIS, to several baselines in terms of their OPE accuracy and ability to rank policies, and explore robustness to bias, noise, and missingness in the annotations.

### 5.1 Synthetic Domains - Bandits

We consider a class of bandit problems with two states $\{s_1, s_2\}$ (drawn with equal probability), two actions $\mathcal{A} = \{\nearrow, \searrow\}$ (recall Figure 3), and corresponding reward distributions $R(s_i, a) \sim \mathcal{N}(\bar{R}_{(s_i, a)}, \sigma^2)$. Without loss of generality, we assume $\nearrow$ is always taken from $s_2$ by both $\pi_b$ and $\pi_e$. For $s_1$, we consider deterministic policies in which one action is always taken, and a stochastic policy that takes the two actions with equal probability (see column/row header in Table 1). Given $(\pi_b, \pi_e)$, we draw $1,000$ samples following $\pi_b$ and then evaluate $\pi_e$ using various estimators, including standard IS, the naive baseline of adding counterfactual annotations as new samples (Section 3.1), and C*-IS. We assume that counterfactual annotations are only available for $s_1$, and all annotations are drawn from the true reward function. We measure the bias, standard deviation (std, the square root of variance), and root mean-squared error (RMSE) of the estimators with respect to $v(\pi_e)$.

**Naive baseline fails due to bias, whereas C*-IS can reduce bias and/or variance compared to IS.** In Table 1, we display the results for $\bar{R}_{(s_1, \nearrow)} = 1$, $\bar{R}_{(s_1, \searrow)} = 2$, $\bar{R}_{(s_2, \cdot)} = 1$, $\sigma = 0.5$ (other settings in Appendix E.1). The naive baseline fails to improve upon IS (and often underperforms IS in terms of RMSE) and can have a nonzero bias even when IS is unbiased (Table 1, row 2, column 2). C*-IS achieves a lower RMSE than IS across all settings considered. The benefits of C*-IS align with our theoretical analyses. (i) *Bias Reduction for Support-Deficient Data.* When $\pi_b$ is deterministic (first row), the unselected action has poor support and IS has a nonzero bias whereas C*-IS is unbiased. Though C*-IS sometimes has a larger variance than IS, the bias reduction outweighs the variance increase and leads to overall lower RMSE. (ii) *Variance Reduction for Well-Supported Data.* In the second row, data generated by $\pi_b$ has full support. Both IS and C*-IS are unbiased, but C*-IS leverages counterfactual annotations to achieve lower variance and lower RMSE.

Table 1: Summary of performance on the bandit problem for various $\pi_b$ (rows) and $\pi_e$ (columns), where each policy is denoted by its probabilities assigned to the two actions from $s_1$. Each cell of the table corresponds to a $(\pi_b, \pi_e)$ combination, for which we report (bias, std, RMSE) for three estimators: IS in the top row, naive in the middle row, and C*-IS in the bottom row. Settings with $\pi_b(s_1) = [0, 1]$ are omitted due to symmetry.

| $\pi_b$ \ $\pi_e$ | [ 1 , 0 ] | [ 0 , 1 ] | [0.5, 0.5] |
|---|---|---|---|
| [ 1 , 0 ] | - | −1 0.6 1.2 
 0.2 1.8 1.8 
 0 0.7 0.7 | −0.5 0.5 0.7 
 0.1 0.7 0.7 
 0 0.5 0.5 |
| [0.5, 0.5] | 0 0.9 0.9 
 0 1 1 
 0 0.5 0.5 | 0 1.6 1.6 
 0.2 1.8 1.8 
 0 0.7 0.7 | - |

We vary the assumptions (e.g., weights in C-IS, noisy/missing annotations) and present further experiments in Appendix E.1. These results suggest that (i) **Equal weights (in C*-IS) is a good heuristic though not always "optimal"** and (ii) **Imputing missing annotations can reduce variance**.

### 5.2 Healthcare Domain - Sepsis Simulator

Next, we apply our approach to evaluate policies in a simulated RL domain modeled after the physiology of sepsis patients [32]. Following prior work [22], we collected 50 offline datasets from the sepsis simulator (using different random seeds) each with 1000 episodes by following an $\epsilon$-greedy behavior policy with respect to the optimal policy where $\epsilon = 0.1$. We considered a set of deterministic policies (including the optimal policy) as evaluation policies, which have different performance and

Table 2: Comparison of baseline and proposed estimators in terms of OPE performance (RMSE, ESS), ranking performance (Spearman's rank correlation) and binary classification performance (accuracy, FPR, FNR) on the sepsis simulator, reported as mean $\pm$ std from 50 repeated runs. **Bolded** results are the best for each metric, whereas highlighted results outperform all baselines.

| | Estimator | $\downarrow$ RMSE | $\uparrow$ ESS | $\uparrow$ Spearman | $\uparrow$ %Accuracy | $\downarrow$ %FPR | $\downarrow$ %FNR | |
|---|---|---|---|---|---|---|---|---|
| Baseline | PDIS (w/o annot.) | 0.113 $\pm0.038$ | 76.8 $\pm44.0$ | 0.596 $\pm0.110$ | 76.5 $\pm3.5$ | 33.7 $\pm8.7$ | 15.9 $\pm4.6$ | |
| | Naive unweighted $(G = Q^{\pi_e})$ | 0.128 $\pm0.006$ | 207.2 $\pm91.5$ | 0.089 $\pm0.089$ | 50.0 $\pm6.0$ | 11.6 $\pm8.3$ | 78.1 $\pm13.6$ | |
| | Naive weighted $\quad(G = Q^{\pi_e})$ | 0.097 $\pm0.006$ | 300.8 $\pm117.6$ | 0.420 $\pm0.097$ | 64.3 $\pm4.7$ | 24.0 $\pm12.7$ | 44.3 $\pm11.4$ | |
| Proposed | C*-PDIS $(G = Q^{\pi_e})$ | **0.013** $\pm0.005$ | **994.0** $\pm10.1$ | **0.995** $\pm0.003$ | **95.7** $\pm3.1$ | 4.5 $\pm6.9$ | **4.2** $\pm5.3$ | $\}$ $\star$ ideal case |
| | C*-PDIS $(G = Q^{\pi_b})$ | 0.070 $\pm0.003$ | **994.0** $\pm10.1$ | 0.961 $\pm0.011$ | 86.8 $\pm8.2$ | 22.0 $\pm20.1$ | 8.2 $\pm11.3$ | $\}$ relaxing |
| | C*-PDIS $(G = Q^{\pi_b} \mapsto \hat{Q}^{\pi_e})$ | 0.028 $\pm0.007$ | **994.0** $\pm10.1$ | 0.979 $\pm0.010$ | 90.1 $\pm5.4$ | **4.2** $\pm6.6$ | 14.1 $\pm9.7$ | Assump. 2 |

Figure 5: (Left) RMSE of C*-PDIS vs. distance to $\pi_b$ (in terms of KL divergence) for each $\pi_e$, plotted with linear trend lines. OPE error increases as $\pi_e$ becomes more different from behavior. (Center&Right) Performance of our proposed approach under noisy and missing annotations. Trend lines show average of 50 runs $\pm$ one std. C*-PDIS is generally robust to noise, and imputing the missing annotations can help maintain competitive performance (relative to the ideal setting) even in the presence of high degrees of missingness.

varying degrees of similarity vs behavior. We compared our proposed estimator (with different annotation functions) with a set of baselines, including standard PDIS (without annotations) and two naive baselines (with perfect annotations): "naive unweighted" simply adds counterfactual annotations as new trajectories and has the same issue discussed in Section 3.1, whereas "naive weighted" reweights the annotations at the trajectory level instead of per-decision. See Appendix E.2 for detailed experimental setup. As the main OPE metric, we report RMSE of value estimates vs true values as well as the effective sample size (ESS). Additionally, we report metrics for two downstream uses of OPE for model selection. (i) When used to rank policies. We report the Spearman's rank correlation between $\hat{v}(\pi_e)$ and $v(\pi_e)$ (computed over all $\pi_e$'s) [22, 26]. (ii) When used to determine whether $\pi_e$ is better or worse than $\pi_b$. We formulate a binary classification problem of $v(\pi_e) \geq v(\pi_b)$ vs $v(\pi_e) < v(\pi_b)$, and report the accuracy, false positive rate (FPR) and false negative rate (FNR).

**C*-PDIS outperforms all baselines across all metrics in the ideal setting.** As shown in Table 2, when all counterfactuals are available and annotated with the evaluation policy's Q-function ($G = Q^{\pi_e}$), C*-PDIS outperforms baseline PDIS (without annotations) in all metrics, demonstrating that it provides more accurate OPE estimates. In contrast, the two naive approaches fail to provide accurate estimates and often underperform standard PDIS.

**C*-PDIS is robust to biased annotations.** Under the more realistic scenario where $G = Q^{\pi_b}$, i.e., annotations summarize the future returns under $\pi_b$ rather than $\pi_e$, we observe a degradation in all metrics compared to the ideal case, though C*-PDIS is still superior to PDIS (Table 2). Applying the bias correction procedure ($G = Q^{\pi_b} \mapsto \hat{Q}^{\pi_e}$, see Appendix D.1) helps in recovering performance closer to the ideal case, especially when $\pi_e$ is far from $\pi_b$ (Figure 5-left).

**Variance reduction of C*-PDIS outweighs the effect of noisy annotations.** When annotations are perturbed with increasing amounts of noise (Figure 5-center), performance degradation is minimal even at the highest level of noise tested (with a std of 1, which is large relative to the reward range $[-1, 1]$, and larger than 0.31 the std of initial state values). Our estimator remains competitive relative to baselines, suggesting that the benefit of variance reduction from additional data (through counterfactual annotations) outweighs the variance increase from annotation noise, even when annotations are much noisier than factual data. See Appendix E.2 for variations of this experiment.

**Collecting more annotations and imputing missing annotations improves performance.** As the amount of available annotations increases (Figure 5-right), our approach interpolates between baseline PDIS and the ideal case of C*-PDIS with a monotonic improvement in performance. Furthermore, imputing annotations achieves better performance, suggesting it is a promising strategy when not all annotations can be collected in practice. See Appendix E.2 for variations of this experiment where the imputed annotations have varying degrees of bias due to annotation noise.

## 6    Related Work

There is a rich literature on statistical methods for offline policy evaluation (OPE), including direct methods (DM), importance sampling (IS), and doubly-robust (DR) approaches [24, 26]. DM directly uses offline data to learn the value function of the evaluation policy (e.g., using model-based or model-free approaches) [33, 34]. We do not consider DM in our work since it often involves function approximators whose bias and variance may be difficult to analyze [35]. On the other hand, IS uses a weighted average of trajectory returns to correct the distributional mismatch between the evaluation and behaviour policies [28, 27]. Our proposed estimator directly builds on IS and uses counterfactual annotations to reduce its variance (and bias), while carefully addressing the nuances involved with reweighting the counterfactual annotations to maintain the unbiasedness property. Finally, DR approaches combine IS and DM and reduce the variance of IS by using the estimates from DM as a control variate [27, 36–38]. Our approach is a complementary source of variance reduction and may be combined with DR approaches by modifying our current definitions to include a control variate. We note that other approaches exist for managing the variance of IS in long-horizon settings by considering the stationary or marginalized state distributions [39–41]. Incorporating counterfactual annotations into these estimators is an interesting direction of future research.

Our work focuses on OPE rather than policy learning, but the broader theme of human input for RL has recently gained renewed attention, particularly in natural language processing tasks [42–44]. In many of these problems, human input is in the form of preferences (or rankings) over actions, states, or (sub-)trajectories [45]. Other related annotation approaches also exist in non-RL areas, where past work has proposed to ask annotators to alter text to match a counterfactual target label [46] and incorporating annotations of potential unmeasured confounders [47]. In contrast, our work investigates the role of a specific form of human input, counterfactual annotations in offline RL, in improving OPE performance. We focus on offline *evaluation* due to its practical importance in high-stakes RL domains such as healthcare, though our insights could also potentially benefit offline *learning* in these domains. While we did not discuss how counterfactual annotations are obtained, our theoretical analysis establishes a thorough understanding of their desirable characteristics. This can help motivate methods for converting different forms of human feedback into useful counterfactual annotations (e.g., learning reward/annotation models from human preferences [42]). Conversely, progress in the field of preference learning and learning-to-rank may benefit our approach by providing mechanisms to solicit high-quality annotations [48].

## 7    Discussion & Conclusion

In this paper, we propose a novel semi-offline policy evaluation framework that incorporates counter-factual annotations into traditional IS estimators. We emphasize that the naive approach of viewing annotations as additional data can lead to bias and propose a simple reweighting scheme to address this issue. We formally study the theoretical properties of our approach, identifying scenarios where bias and variance reduction is guaranteed. Driven by a deep understanding of these theoretical properties, we further propose practically motivated strategies to handle biased, noisy, or missing annotations. Through proof-of-concept experiments on bandits and a healthcare-inspired RL simulator, we demonstrate that our approach outperforms standard IS and is robust to imperfect annotations. Our semi-offline framework serves as an intermediate step between offline and online evaluations and has the potential to enable practical applications of RL in high-stakes domains. Though motivated by current limitations of offline evaluation, we caution that our approach is not meant to replace existing OPE methods, but rather to complement them. Collecting annotations from domain experts comes at a cost (of real human time and labor), and thus, our approach should only be applied after a policy has passed all checks on retrospective data. While not explored in this paper, future work should focus on assessing the quality of counterfactual annotations in the domains of interest through human experiments. See Appendix A for more detailed discussions on limitations, societal impacts, and future directions. Overall, we believe our contributions will inspire further investigations into the practical obstacles that emerge in semi-offline evaluation (e.g., devising human-centered strategies for soliciting counterfactual annotations that align with theoretical assumptions) and will bring RL closer to reality in healthcare and other high-stakes decision-making domains.

## Acknowledgments

This work was supported by the National Library of Medicine of the National Institutes of Health (grant R01LM013325 to JW). The views and conclusions in this document are those of the authors and should not be interpreted as necessarily representing the official policies, either expressed or implied, of the National Institutes of Health. The authors would like to thank Michael Sjoding and members of the MLD3 group for helpful discussions regarding this work, as well as the anonymous reviewers for constructive feedback.

## Data and Code Availability

The code for all experiments is available at https://github.com/MLD3/CounterfactualAnnot-SemiOPE.

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

# A  Further Discussions

**Limitations.** This work proposes and theoretically studies a new framework of semi-offline evaluation of RL policies. While our core idea is to incorporate human annotations into the evaluation process, we did not make use of real human annotations in our experiments and relied on simulations instead. Given the costs associated with user studies, we opted not to conduct user studies before we have a thorough understanding of when and why our approach works (or does not work) and how it can be implemented in practice. Our present paper focuses on describing and analyzing a formal mathematical framework of counterfactual annotations, which provides valuable insights as to what types of annotations are useful, how they should be incorporated into OPE, and which factors impact performance (see method description in Section 3 and theoretical analyses in Section 4). Our experiments further demonstrate the robustness of our approach when we deviate from the ideal setting (see Section 5). Despite best efforts, our experiments do not capture all possible scenarios in the real world. We list several important practical implications in Section 4.4 and encourage future research into these directions. Additional areas of interest that we did not explore include: alternative forms of annotations (e.g., preference, immediate reward), whether these other annotation types could be converted to match our assumptions so that our framework still applies, how best to design the question phrasing for the annotators, evaluating annotation quality, targeted annotation solicitation and optimizing annotation solicitation under budget constraint.

**Ethical Considerations and Societal Impact.** Since our experiments only involved simulations and no real humans, we did not pose immediate ethical concerns or safety threats. Nonetheless, in high-stakes decision-making tasks such as healthcare, computationally-derived RL policies must be carefully validated before their final adoption. While we are motivated by the current limitations of standard offline evaluation methods, we caution that our approach is not meant to replace, but rather to complement, existing OPE approaches. Collecting these annotations from real human domain experts comes at a cost, and thus we recommend applying our approach only when a policy has passed all checks on retrospective data. We note that compared to traditional qualitative evaluations (e.g., typically done by asking domain experts whether RL recommendations make sense), our approach is less likely (though still possible) to affect the existing decision making or suffer from confirmation bias, since we do not need to reveal the evaluation policy at the annotation collection stage. Future work that adopts our framework should carefully design the annotation solicitation process (including when and how the question is posed to the annotators) so as to achieve safe, non-disruptive evaluations of offline RL policies before their prospective use. Additionally, more work is needed to understand the extent to which humans can provide accurate annotations of counterfactuals. These human experiments should be addressed in the context of specific application domains and the target groups of expert human annotators.

**Future Directions on Counterfactual Annotations.** Our present paper establishes important theoretical groundwork for using counterfactual annotations in semi-offline policy evaluation. However, we did not collect any real annotations with human annotators as those are outside of the scope of the main research question we seek to address in our current paper. To fully realize the impact of this work, real-world human experiments would be the necessary next step, with a focus on obtaining faithful annotations (with small bias and small noise). More specifically, it would be important to empirically measure if and how various factors influence annotation quality, including: horizon $t$ (annotating beginning of an episode or the terminal step), the inherent stochasticity associated with the annotated state-action pair and how often they appear in data (under $\pi_b$). Equipped with this knowledge, it is then important to select which annotations to prioritize given a limited annotation budget. Many of these questions delve into the realm of HCI and are outside the scope of this paper's methodological contributions for offline RL and OPE, and we encourage researchers and practitioners in different research areas (e.g., RL, HCI, healthcare, education) to build upon the ideas in our work.

# B  Toy Example for Intuition

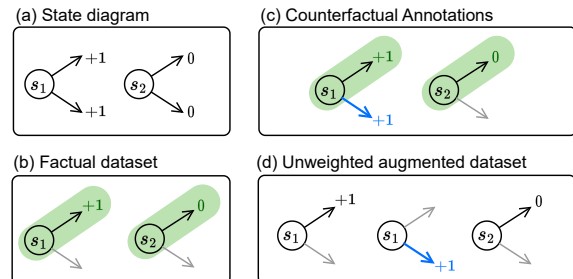

Figure 6: For ease of reference, Figure 3 is reproduced here. (a) The state diagram of a bandit problem with two states and two actions. (b) A factual dataset containing two samples. (c) The factual samples augmented with counterfactual annotations. (d) The (unweighted) augmented dataset constructed from factual samples and counterfactual annotations. Compared to the factual dataset, the relative frequency of $s_1$ vs $s_2$ changed from $1 : 1$ to $2 : 1$.

Recall the example discussed in Section 3.1 (figure reproduced here in Figure 6). The bandit problem has two states $\{s_1, s_2\}$ (drawn with equal probability) and two actions, up ($\nearrow$) and down ($\searrow$). The reward from $s_1$ is $+1$ and from $s_2$ is $0$ (i.e., rewards do not depend on the action), meaning all policies have an expected reward of $0.5$. Suppose the behavior policy always selects $\nearrow$, generating a dataset with poor support for policies that assign nonzero probabilities to $\searrow$ (Figure 6b). Now suppose we also have access to human-provided annotations of counterfactual actions, but not all counterfactual annotations are available (either because they were never queried or the users declined to provide annotations). In our example (Figure 6c), one annotation is collected for the counterfactual action $\searrow$ at state $s_1$, indicating that the human annotator believes the reward for taking action $\searrow$ from state $s_1$ is $+1$. To make use of this information, one might consider adding the counterfactual annotation as a new sample. The augmented dataset (Figure 6d) would allow us to evaluate policies (e.g., using IS) that assign non-zero probabilities to $\searrow$ in state $s_1$. Unfortunately, this naive approach leads to biased results, which we walk through in detail below.

Consider an evaluation policy that takes $\searrow$ in $s_1$ and $\nearrow$ in $s_2$, i.e., $\pi_e(s_1) = [0, 1]$ and $\pi_e(s_2) = [1, 0]$ (denoted in terms of the probabilities assigned to the two actions, $\nearrow$ and $\searrow$).

- If we use the original $\pi_b$ in the factual dataset (Figure 6b) where $\pi_b(s_1) = [1, 0]$, $\pi_b(s_2) = [1, 0]$, the IS estimator is ill-defined because we will encounter a divide-by-zero error:

$$\frac{1}{3}\left(\frac{0}{1} \times (+1) + \frac{1}{0} \times (+1) + \frac{1}{1} \times (0)\right) = \text{undefined}$$

- Instead, one may consider the behavior policy in the augmented dataset (Figure 6d), which gives $\pi_b(s_1) = [0.5, 0.5]$, $\pi_b(s_2) = [1, 0]$. The IS estimate is:

$$\frac{1}{3}\left(\frac{0}{0.5} \times (+1) + \frac{1}{0.5} \times (+1) + \frac{1}{1} \times (0)\right) = \frac{2}{3}$$

However, as stated above, all policies should have a value of $0.5$, meaning that $\frac{2}{3}$ is a biased estimate. The core of the issue is because directly adding counterfactual annotations inadvertently changes the state distribution and results in a dataset inconsistent with the original problem. In particular, comparing Figure 6b vs 6d, the relative frequency of $s_1$ vs $s_2$ has changed from $1 : 1$ to $2 : 1$.

Our proposed estimator addresses this issue by reweighting the factual data and counterfactual annotations in order to maintain the state distribution. Suppose we assign a weight $\alpha$ to $(s_1, \nearrow)$ and $1 - \alpha$ to $(s_1, \searrow)$ for some $\alpha \in (0, 1)$, i.e., the two non-negative weights associated with $s_1$ sum to 1. The sample $(s_2, \nearrow)$ receives a weight of 1 by default since it does not have an associated counterfactual annotation. In this way, the state distribution remains equally split between $s_1$ and $s_2$.

Using Definition 1, we can calculate the augmented behavior policy $\pi_{b^+}(s_1) = [\alpha, 1 - \alpha]$, $\pi_{b^+}(s_2) = [1, 0]$. Applying C-IS as defined in Definition 3, we see our proposed approach produces the correct, unbiased estimate of $0.5$:

$$\text{for } s_1, \ \left(\alpha \times \frac{0}{\alpha} \times (+1) + (1-\alpha) \times \frac{1}{1-\alpha} \times (+1)\right) = 1$$

$$\text{for } s_2, \ \left(1 \times \frac{1}{1} \times (0)\right) = 0$$

$$\text{overall: } \frac{1}{2}(1+0) = 0.5$$

## B.1 Another Example Illustrating the Weighting Scheme

Suppose similar to Figure 6, we now have a bandit problem with only a single state $\{s\}$ such that all policies have an expected reward of 1. Suppose the behavior policy always selects $\nearrow$, generating a dataset containing two samples: sample #1 $(s, \nearrow)$ with an annotation for $\searrow$, sample #2 is simply $(s, \nearrow)$ with no annotation for $\searrow$. In applying our approach, for sample #1 we can assign weights $[\alpha, 1-\alpha]$ for some $\alpha \in (0,1)$, i.e., the two non-negative weights associated with sample #1 sum to 1; for sample #2 the weights are simply $[1, 0]$. Then, the average weights for state $s$ are $\bar{W}(\nearrow \,|s, \nearrow) = \frac{1+\alpha}{2}$ and $\bar{W}(\searrow \,|s, \nearrow) = \frac{1-\alpha}{2}$. For example, if $\alpha = 0.8$, then the weights for sample #1 are $[0.8, 0.2]$ and the average weights associated with state $s$ are $[0.9, 0.1]$. Note that while the average weights $\bar{W}$ (used for calculating $\pi_{b^+}$ and C-IS) are *state-specific*, the user-assigned weights $w^{\tilde{a}}$ are *sample-specific*; furthermore, the sample-specific weights should not be interpreted as "random", as a user may deliberately set weights to split equally into $[0.5, 0.5]$, or to $[1, 0]$ which ignores the annotation if they believe it is of poor quality.

Using Definition 1, we can calculate the augmented behavior policy $\pi_{b^+}(s) = [\frac{1+\alpha}{2}, \frac{1-\alpha}{2}]$. Applying C-IS as defined in Definition 2, we see our proposed approach produces an unbiased estimate of 1:

$$\left(\frac{1+\alpha}{2} \times \frac{0}{\frac{1+\alpha}{2}} \times (+1) + \frac{1-\alpha}{2} \times \frac{1}{\frac{1-\alpha}{2}} \times (+1)\right) = 1$$

# C Extended Theoretical Analyses

Unless otherwise stated, the estimators are for $\pi_e$, i.e., $\hat{v} = \hat{v}(\pi_e)$.

## C.1 IS: Bias & Variance

We formally state and prove the bias and variance results for IS (informally described in Section 4). The proofs are adapted from existing literature [27, 30].

**Theorem 9** (Unbiasedness of IS). *In the bandit setting, when Assumption 3 holds,* $\mathbb{E}[\hat{v}^{\text{IS}}] = v(\pi_e)$.

**Proposition 10** (Bias of IS). *In the bandit setting, when Assumption 3 is violated,* $\text{Bias}[\hat{v}^{\text{IS}}] = \mathbb{E}_{s \sim d_1}\left[-\sum_{a \in \mathcal{U}(s,\pi_b)} \pi_e(a|s)\bar{R}(s,a)\right]$ *where* $\mathcal{U}(s, \pi_b) = \{a : \pi_b(a|s) = 0\}$ *are unsupported actions in the dataset.*

**Proposition 11** (Variance of IS). *In the bandit setting, when Assumption 3 holds, the variance of IS can be written as:*

$$\mathbb{V}[\hat{v}^{\text{IS}}] = \mathbb{V}_{s \sim d_1}[V^{\pi_e}(s)] + \mathbb{E}_{s \sim d_1}\left[\mathbb{V}_{a \sim \pi_b(s)}[\rho(a|s)\bar{R}(s,a)]\right] + \mathbb{E}_{s \sim d_1}\left[\mathbb{E}_{a \sim \pi_b(s)}[\rho(a|s)^2 \, \sigma_R(s,a)^2]\right]$$

*where* $\sigma_R(s,a)^2 = \mathbb{V}_{r \sim R(s,a)}[r]$ *is the variance associated with the reward function* $R(s,a)$.

*Derivation for Bias of IS (adapted from [30]).*

$$\text{Bias}[\hat{v}^{\text{IS}}] = \mathbb{E}[\hat{v}^{\text{IS}}] - v(\pi_e)$$

$$= \mathbb{E}_{s \sim d_1}\left[\sum_{a \in \mathcal{A}\backslash\mathcal{U}(s,\pi_b)} \pi_b(a|s)\frac{\pi_e(a|s)}{\pi_b(a|s)}\bar{R}(s,a)\right] - \mathbb{E}_{s \sim d_1}\left[\sum_{a \in \mathcal{A}} \pi_e(a|s)\bar{R}(s,a)\right]$$

$$= \mathbb{E}_{s \sim d_1}\left[\sum_{a \in \mathcal{A}\backslash\mathcal{U}(s,\pi_b)} \pi_e(a|s)\bar{R}(s,a) - \sum_{a \in \mathcal{A}} \pi_e(a|s)\bar{R}(s,a)\right]$$

$$= \mathop{\mathbb{E}}_{s\sim d_1}\left[-\sum_{a\in\mathcal{U}(s,\pi_b)}\pi_e(a|s)\bar{R}(s,a)\right]$$

If Assumption 3 holds, $\pi_e(a|s) = 0$ for $a \in \mathcal{U}(s,\pi_b)$ and thus $\mathrm{Bias}[\hat{v}^{\mathrm{IS}}] = 0$. $\qquad\square$

*Derivation for Variance of IS (adapted from [27]).* We apply the law of total variance:

$$\mathbb{V}[\hat{v}^{\mathrm{IS}}] = \mathbb{V}_{s\sim d_1, a\sim\pi_b(s), r\sim R(s,a)}[\rho r]$$

$$= \mathbb{V}_{s\sim d_1}\left[\mathbb{E}_{a\sim\pi_b(s), r\sim R(s,a)}[\rho r]\right] + \mathbb{E}_{s\sim d_1}\left[\mathbb{V}_{a\sim\pi_b(s), r\sim R(s,a)}[\rho r]\right]$$

$$= \mathbb{V}_{s\sim d_1}\left[\mathbb{E}_{a\sim\pi_b(s)}\left[\frac{\pi_e(a|s)}{\pi_b(a|s)}\mathbb{E}_{r\sim R(s,a)}[r]\right]\right] + \mathbb{E}_{s\sim d_1}\left[\mathbb{V}_{a\sim\pi_b(s)}\mathbb{E}_{r\sim R(s,a)}[\rho r] + \mathbb{E}_{a\sim\pi_b(s)}\mathbb{V}_{r\sim R(s,a)}[\rho r]\right]$$

$$= \mathbb{V}_{s\sim d_1}\left[\mathbb{E}_{a\sim\pi_e(s)}[\bar{R}(s,a)]\right] + \mathbb{E}_{s\sim d_1}\left[\mathbb{V}_{a\sim\pi_b(s)}[\rho\bar{R}(s,a)]\right] + \mathbb{E}_{s\sim d_1}\mathbb{E}_{a\sim\pi_b(s)}\left[\rho^2\mathbb{V}_{r\sim R(s,a)}[r]\right]$$

$$= \mathbb{V}_{s\sim d_1}[V^{\pi_e}(s)] + \mathbb{E}_{s\sim d_1}\left[\mathbb{V}_{a\sim\pi_b(s)}[\rho(a|s)\bar{R}(s,a)]\right] + \mathbb{E}_{s\sim d_1}\left[\mathbb{E}_{a\sim\pi_b(s)}\left[\rho(a|s)^2\sigma_R(s,a)^2\right]\right] \qquad \square$$

## C.2 C-IS: Bias Analyses

We start by showing unbiasedness of C-IS in the ideal case (Theorem 1) .

**Theorem 1** (Unbiasedness of C-IS). *In the bandit setting, when both Assumptions 1 and 4 hold, the C-IS estimator is unbiased, $\mathbb{E}[\hat{v}^{\mathrm{C\text{-}IS}}] = v(\pi_e)$.*

*Proof of Theorem 1.* Starting with Definition 2,

$$\mathbb{E}[\hat{v}^{\mathrm{C\text{-}IS}}] = \mathbb{E}\left[w^a\rho^a r + \sum_{\tilde{a}\in\mathcal{A}\setminus\{a\}}w^{\tilde{a}}\rho^{\tilde{a}}q^{\tilde{a}}\right]$$

$$= \mathop{\mathbb{E}}_{s\sim d_1}\mathop{\mathbb{E}}_{a\sim\pi_b(s)}\left[\mathbb{E}_{w^a\sim W(a|s,a)}[w^a]\frac{\pi_e(a|s)}{\pi_{b^+}(a|s)}\mathbb{E}_{r\sim R(s,a)}[r]\right.$$

$$\left. + \sum_{\tilde{a}\in\mathcal{A}\setminus\{a\}}\mathbb{E}_{w^{\tilde{a}}\sim W(\tilde{a}|s,a)}[w^{\tilde{a}}]\frac{\pi_e(\tilde{a}|s)}{\pi_{b^+}(\tilde{a}|s)}\mathbb{E}_{g^{\tilde{a}}\sim G(s,\tilde{a})}[g^{\tilde{a}}]\right]$$

$$\stackrel{(1)}{=} \mathop{\mathbb{E}}_{s\sim d_1}\left[\sum_{a\in\mathcal{A}}\pi_b(a|s)\left(\sum_{\tilde{a}\in\mathcal{A}}\bar{W}(\tilde{a}|s,a)\frac{\pi_e(\tilde{a}|s)}{\pi_{b^+}(\tilde{a}|s)}\bar{R}(s,\tilde{a})\right)\right]$$

$$\stackrel{(2)}{=} \mathop{\mathbb{E}}_{s\sim d_1}\left[\sum_{\tilde{a}\in\mathcal{A}}\left(\sum_{a\in\mathcal{A}}\pi_b(a|s)\bar{W}(\tilde{a}|s,a)\frac{\pi_e(\tilde{a}|s)}{\pi_{b^+}(\tilde{a}|s)}\bar{R}(s,\tilde{a})\right)\right]$$

$$\stackrel{(3)}{=} \mathop{\mathbb{E}}_{s\sim d_1}\left[\sum_{\tilde{a}\in\mathcal{A}}\left(\left(\sum_{a\in\mathcal{A}}\pi_b(a|s)\bar{W}(\tilde{a}|s,a)\right)\frac{\pi_e(\tilde{a}|s)}{\pi_{b^+}(\tilde{a}|s)}\bar{R}(s,\tilde{a})\right)\right]$$

$$\stackrel{(4)}{=} \mathop{\mathbb{E}}_{s\sim d_1}\left[\sum_{\tilde{a}\in\mathcal{A}}\cancel{\pi_{b^+}(\tilde{a}|s)}\frac{\pi_e(\tilde{a}|s)}{\cancel{\pi_{b^+}(\tilde{a}|s)}}\bar{R}(s,\tilde{a})\right]$$

$$= \mathop{\mathbb{E}}_{s\sim d_1}\left[\sum_{\tilde{a}\in\mathcal{A}}\pi_e(\tilde{a}|s)\bar{R}(s,\tilde{a})\right]$$

$$= \mathop{\mathbb{E}}_{s\sim d_1}\mathop{\mathbb{E}}_{\tilde{a}\sim\pi_e}\left[\bar{R}(s,\tilde{a})\right]$$

$$= \mathop{\mathbb{E}}_{s\sim d_1}[V^{\pi_e}(s)]$$

$$= v(\pi_e)$$

where in (1) we replace $\mathbb{E}_{g\sim G(s,\tilde{a})}[g]$ with $\bar{R}(s,\tilde{a})$ following Assumption 1 and combine it with $\mathbb{E}_{r\sim R(s,a)}[r]$ in a single summation over $\tilde{a}\in\mathcal{A}$, in (2) we swap the order of summations, in (3) we take out common factors that do not depend on $a$, and in (4) we use Definition 1 for $\pi_{b^+}(\tilde{a}|s)$, which cancels out the denominator in the importance ratio on the next line. $\qquad\square$

Next, we look at two factors contributing to the bias of C-IS: lack of support and imperfect annotations.

**Proposition 2** (Bias of C-IS due to support). *When Assumption 1 holds but Assumption 4 is violated,* $\text{Bias}[\hat{v}^{\text{C-IS}}] = \mathbb{E}_{s\sim d_1}\left[-\sum_{a\in\mathcal{U}(s,\pi_{b^+})}\pi_e(a|s)\bar{R}(s,a)\right]$ *where* $\mathcal{U}(s,\pi_{b^+}) = \{a : \pi_{b^+}(a|s) = 0\}$ *are unsupported actions in the counterfactual-augmented dataset.*

**Proposition 3** (Bias of C-IS due to imperfect annotations). *When Assumption 4 holds but Assumption 1 is violated,* $\text{Bias}[\hat{v}^{\text{C-IS}}] = \mathbb{E}_{s\sim d_1}\mathbb{E}_{a\sim\pi_e(s)}\left[\delta_W(s,a)\,\epsilon_G(s,a)\right]$, *where we measure violation of Assumption 1 as* $\epsilon_G(s,a) = \mathbb{E}_{g\sim G(s,a)}[g] - \bar{R}(s,a)$, *and* $\delta_W(s,a) = \left(1 - \frac{\bar{W}(a|s,a)\pi_b(a|s)}{\pi_{b^+}(a|s)}\right)$.

**Proposition 12** (Bias of C-IS, combined). *When both Assumptions 1 and 4 are violated,* $\text{Bias}[\hat{v}^{\text{C-IS}}] = \mathbb{E}_{s\sim d_1}\left[-\sum_{a\in\mathcal{U}(s,\pi_{b^+})}\pi_e(a|s)\,\bar{R}(s,a)\right]$ $+$ $\mathbb{E}_{s\sim d_1}\left[\sum_{a\in\mathcal{A}\backslash\mathcal{U}(s,\pi_b)}\delta_W(s,a)\,\pi_e(a|s)\,\epsilon_G(s,a) + \sum_{a\in\mathcal{U}(s,\pi_b)\backslash\mathcal{U}(s,\pi_{b^+})}\pi_e(a|s)\,\epsilon_G(s,a)\right]$, *where* $\mathcal{U}(s,\pi_{b^+}) = \{a : \pi_{b^+}(a|s) = 0\}$ *are unsupported actions in the counterfactual-augmented dataset,* $\epsilon_G(s,a) = \mathbb{E}_{g\sim G(s,a)}[g] - \bar{R}(s,a)$, *and* $\delta_W(s,a) = \left(1 - \frac{\bar{W}(a|s,a)\pi_b(a|s)}{\pi_{b^+}(a|s)}\right)$.

*Remark.* There are two main sources of bias for C-IS, resulting from each of the two assumptions being violated. In Proposition 12, missing annotations (i.e., violation of Assumption 4) contribute to the bias through the first term as the rewards of unsupported actions, whereas imperfect annotations (i.e., violation of Assumption 1) contribute to the bias through the second term as the annotation error over supported actions. When both assumptions hold, C-IS only requires a weaker version of the common support assumption to remain unbiased (Theorem 1); consequently, compared to IS, there is a larger space of policies that C-IS can evaluate without bias. Note that the unbiasedness property is not affected by the user-defined weighting scheme (except that the weights must not be to 0 or 1) and directly applies to C*-IS. When only Assumption 1 holds (Proposition 2), the bias is related to the negative rewards over unsupported actions. On the other hand, when only Assumption 4 holds (Proposition 3), the annotation error $\epsilon_G$ contributes to the resulting bias and this contribution is scaled by a factor of $\delta_W(s,a) \leq 1$ when the estimator can make use of (unbiased) factual samples of the state-action pair $(s,a)$ (i.e., when $\pi_b(a|s) > 0$ and $\bar{W}(a|s,a) > 0$).

We show Proposition 12 first and then discuss Propositions 2 and 3 as two special cases.

*Proof of Proposition 12.*

$$\text{Bias}[\hat{v}^{\text{C-IS}}] = \mathbb{E}[\hat{v}^{\text{C-IS}}] - v(\pi_e)$$

$$= \mathop{\mathbb{E}}_{s\sim d_1}\left[\sum_{a\in\mathcal{A}\backslash\mathcal{U}(s,\pi_b)}\pi_b(a|s)\left(\mathbb{E}_{w^a\sim W(a|s,a)}[w^a]\frac{\pi_e(a|s)}{\pi_{b^+}(a|s)}\mathbb{E}_{r\sim R(s,a)}[r]\right.\right.$$

$$\left.\left.+ \sum_{\tilde{a}\in\mathcal{A}\backslash\mathcal{U}(s,\pi_{b^+})\backslash\{a\}}\mathbb{E}_{w^{\tilde{a}}\sim W(\tilde{a}|s,a)}[w^{\tilde{a}}]\frac{\pi_e(\tilde{a}|s)}{\pi_{b^+}(\tilde{a}|s)}\mathbb{E}_{g^{\tilde{a}}\sim G(s,\tilde{a})}[g^{\tilde{a}}]\right)\right] - v(\pi_e)$$

$$= \mathop{\mathbb{E}}_{s\sim d_1}\left[\sum_{a\in\mathcal{A}\backslash\mathcal{U}(s,\pi_b)}\pi_b(a|s)\left(\sum_{\tilde{a}\in\mathcal{A}\backslash\mathcal{U}(s,\pi_{b^+})}\bar{W}(\tilde{a}|s,a)\frac{\pi_e(\tilde{a}|s)}{\pi_{b^+}(\tilde{a}|s)}\left(\bar{R}(s,\tilde{a}) + \epsilon_G(s,\tilde{a})\right)\right.\right.$$

$$\left.\left.- \bar{W}(a|s,a)\frac{\pi_e(a|s)}{\pi_{b^+}(a|s)}\epsilon_G(s,a)\right)\right] - v(\pi_e)$$

$$= \mathop{\mathbb{E}}_{s\sim d_1}\left[\sum_{a\in\mathcal{A}\backslash\mathcal{U}(s,\pi_b)}\pi_b(a|s)\sum_{\tilde{a}\in\mathcal{A}\backslash\mathcal{U}(s,\pi_{b^+})}\bar{W}(\tilde{a}|s,a)\frac{\pi_e(\tilde{a}|s)}{\pi_{b^+}(\tilde{a}|s)}\bar{R}(s,\tilde{a})\right] - v(\pi_e)$$

$$+ \mathop{\mathbb{E}}_{s\sim d_1}\left[\sum_{a\in\mathcal{A}\backslash\mathcal{U}(s,\pi_b)}\pi_b(a|s)\sum_{\tilde{a}\in\mathcal{A}\backslash\mathcal{U}(s,\pi_{b^+})}\bar{W}(\tilde{a}|s,a)\frac{\pi_e(\tilde{a}|s)}{\pi_{b^+}(\tilde{a}|s)}\epsilon_G(s,\tilde{a})\right]$$

$$- \mathop{\mathbb{E}}_{s\sim d_1}\left[\sum_{a\in\mathcal{A}\backslash\mathcal{U}(s,\pi_b)}\pi_b(a|s)\bar{W}(a|s,a)\frac{\pi_e(a|s)}{\pi_{b^+}(a|s)}\epsilon_G(s,a)\right]$$

$$= \mathop{\mathbb{E}}_{s\sim d_1}\left[\sum_{\tilde{a}\in\mathcal{A}\backslash\mathcal{U}(s,\pi_{b^+})}\left(\sum_{a\in\mathcal{A}\backslash\mathcal{U}(s,\pi_b)}\pi_b(a|s)\bar{W}(\tilde{a}|s,a)\right)\frac{\pi_e(\tilde{a}|s)}{\pi_{b^+}(\tilde{a}|s)}\bar{R}(s,\tilde{a})\right] - v(\pi_e)$$

$$+ \underset{s\sim d_1}{\mathbb{E}} \left[ \sum_{\tilde{a}\in\mathcal{A}\backslash\mathcal{U}(s,\pi_{b^+})} \left( \sum_{a\in\mathcal{A}\backslash\mathcal{U}(s,\pi_b)} \pi_b(a|s)\bar{W}(\tilde{a}|s,a) \right) \frac{\pi_e(\tilde{a}|s)}{\pi_{b^+}(\tilde{a}|s)}\epsilon_G(s,\tilde{a}) \right]$$

$$- \underset{s\sim d_1}{\mathbb{E}} \left[ \sum_{a\in\mathcal{A}\backslash\mathcal{U}(s,\pi_b)} \pi_b(a|s)\bar{W}(a|s,a)\frac{\pi_e(a|s)}{\pi_{b^+}(a|s)}\epsilon_G(s,a) \right]$$

$$= \underset{s\sim d_1}{\mathbb{E}} \left[ - \sum_{a\in\mathcal{U}(s,\pi_{b^+})} \pi_e(a|s)\bar{R}(s,a) \right] + \underset{s\sim d_1}{\mathbb{E}} \left[ \sum_{a\in\mathcal{A}\backslash\mathcal{U}(s,\pi_{b^+})} \pi_e(a|s)\epsilon_G(s,a) \right]$$

$$- \underset{s\sim d_1}{\mathbb{E}} \left[ \sum_{a\in\mathcal{A}\backslash\mathcal{U}(s,\pi_b)} \pi_e(a|s)\bar{W}(a|s,a)\frac{\pi_b(a|s)}{\pi_{b^+}(a|s)}\epsilon_G(s,a) \right]$$

$$= \underset{s\sim d_1}{\mathbb{E}} \left[ - \sum_{a\in\mathcal{U}(s,\pi_{b^+})} \pi_e(a|s)\bar{R}(s,a) \right]$$

$$+ \underset{s\sim d_1}{\mathbb{E}} \left[ \sum_{a\in\mathcal{A}\backslash\mathcal{U}(s,\pi_b)} \delta_W(s,a)\pi_e(a|s)\epsilon_G(s,a) + \sum_{a\in\mathcal{U}(s,\pi_b)\backslash\mathcal{U}(s,\pi_{b^+})} \pi_e(a|s)\epsilon_G(s,a) \right]$$

where $\delta_W(s,a) = \left(1 - \frac{\bar{W}(a|s,a)\pi_b(a|s)}{\pi_{b^+}(a|s)}\right)$.

Note that $\mathcal{U}(s,\pi_{b^+}) \subseteq \mathcal{U}(s,\pi_b)$, because $\pi_{b^+}(a|s) = 0$ implies $\pi_b(a|s) = 0$ (factual data does not contain action $a$) and $\bar{W}(a|s,\check{a}) = 0, \forall\check{a}\in\mathcal{A}$ (counterfactual annotations for other actions $\check{a}$ also do not contain action $a$). $\qquad\square$

*Proof for Proposition 2.* Given Assumption 1 but not Assumption 4,

$$\text{Bias}[\hat{v}^{\text{C-IS}}] = \mathbb{E}[\hat{v}^{\text{C-IS}}] - v(\pi_e)$$

$$= \underset{s\sim d_1}{\mathbb{E}} \left[ \sum_{a\in\mathcal{A}\backslash\mathcal{U}(s,\pi_b)} \pi_b(a|s) \left( \sum_{\tilde{a}\in\mathcal{A}\backslash\mathcal{U}(s,\pi_{b^+})} \bar{W}(\tilde{a}|s,a)\frac{\pi_e(\tilde{a}|s)}{\pi_{b^+}(\tilde{a}|s)}\bar{R}(s,\tilde{a}) \right) \right] - v(\pi_e)$$

$$= \underset{s\sim d_1}{\mathbb{E}} \left[ \sum_{\tilde{a}\in\mathcal{A}\backslash\mathcal{U}(s,\pi_{b^+})} \left( \left( \sum_{a\in\mathcal{A}\backslash\mathcal{U}(s,\pi_b)} \pi_b(a|s)\bar{W}(\tilde{a}|s,a) \right) \frac{\pi_e(\tilde{a}|s)}{\pi_{b^+}(\tilde{a}|s)}\bar{R}(s,\tilde{a}) \right) \right] - \underset{s\sim d_1}{\mathbb{E}} \left[ \sum_{a\in\mathcal{A}} \pi_e(a|s)\bar{R}(s,a) \right]$$

$$= \underset{s\sim d_1}{\mathbb{E}} \left[ \sum_{a\in\mathcal{A}\backslash\mathcal{U}(s,\pi_{b^+})} \pi_e(a|s)\bar{R}(s,a) - \sum_{a\in\mathcal{A}} \pi_e(a|s)\bar{R}(s,a) \right]$$

$$= \underset{s\sim d_1}{\mathbb{E}} \left[ -\sum_{a\in\mathcal{U}(s,\pi_{b^+})} \pi_e(a|s)\bar{R}(s,a) \right] \qquad\qquad\square$$

*Proof for Proposition 3.* Given Assumptions 3 and 4 but not Assumption 1,

$$\text{Bias}[\hat{v}^{\text{C-IS}}] = \mathbb{E}[\hat{v}^{\text{C-IS}}] - v(\pi_e)$$

$$= \underset{s\sim d_1}{\mathbb{E}} \left[ \sum_{a\in\mathcal{A}} \pi_b(a|s) \left( \bar{W}(a|s,a)\frac{\pi_e(a|s)}{\pi_{b^+}(a|s)}\bar{R}(s,\tilde{a}) + \sum_{\tilde{a}\in\mathcal{A}\backslash\{a\}} \bar{W}(\tilde{a}|s,a)\frac{\pi_e(\tilde{a}|s)}{\pi_{b^+}(\tilde{a}|s)} \left( \bar{R}(s,\tilde{a}) + \epsilon_G(s,\tilde{a}) \right) \right) \right] - v(\pi_e)$$

$$= \underset{s\sim d_1}{\mathbb{E}} \left[ \sum_{a\in\mathcal{A}} \pi_b(a|s) \left( \sum_{\tilde{a}\in\mathcal{A}} \bar{W}(\tilde{a}|s,a)\frac{\pi_e(\tilde{a}|s)}{\pi_{b^+}(\tilde{a}|s)}\bar{R}(s,\tilde{a}) + \sum_{\tilde{a}\in\mathcal{A}\backslash\{a\}} \bar{W}(\tilde{a}|s,a)\frac{\pi_e(\tilde{a}|s)}{\pi_{b^+}(\tilde{a}|s)}\epsilon_G(s,\tilde{a}) \right) \right] - v(\pi_e)$$

$$= \underset{s\sim d_1}{\mathbb{E}} \left[ \sum_{\tilde{a}\in\mathcal{A}} \left( \left( \sum_{a\in\mathcal{A}} \pi_b(a|s)\bar{W}(\tilde{a}|s,a) \right) \frac{\pi_e(\tilde{a}|s)}{\pi_{b^+}(\tilde{a}|s)}\bar{R}(s,\tilde{a}) \right) \right] - v(\pi_e)$$

$$+ \mathop{\mathbb{E}}_{s\sim d_1}\left[\sum_{\tilde{a}\in\mathcal{A}}\left(\left(\sum_{a\in\mathcal{A}}\pi_b(a|s)\bar{W}(\tilde{a}|s,a)\right)\frac{\pi_e(\tilde{a}|s)}{\pi_{b^+}(\tilde{a}|s)}\epsilon_G(s,\tilde{a})\right)\right]$$

$$- \mathop{\mathbb{E}}_{s\sim d_1}\left[\left(\sum_{a\in\mathcal{A}}\pi_b(a|s)\bar{W}(a|s,a)\frac{\pi_e(a|s)}{\pi_{b^+}(a|s)}\epsilon_G(s,a)\right)\right]$$

$$= \mathop{\mathbb{E}}_{s\sim d_1}\left[\sum_{\tilde{a}\in\mathcal{A}}\pi_e(\tilde{a}|s)\epsilon_G(s,\tilde{a})\right] - \mathop{\mathbb{E}}_{s\sim d_1}\left[\sum_{a\in\mathcal{A}}\pi_e(a|s)\bar{W}(a|s,a)\frac{\pi_b(a|s)}{\pi_{b^+}(a|s)}\epsilon_G(s,a)\right]$$

$$= \mathop{\mathbb{E}}_{s\sim d_1}\mathop{\mathbb{E}}_{a\sim\pi_e(s)}\left[\left(1 - \frac{\bar{W}(a|s,a)\pi_b(a|s)}{\pi_{b^+}(a|s)}\right)\epsilon_G(s,a)\right] \qquad \square$$

While it is generally true that the space of policies that C-IS can evaluate without bias is larger than that of IS, since the lack of support leads to C-IS being biased towards 0 (similar to the case of standard IS), the magnitude of bias of C-IS is not guaranteed to be less than standard IS without additional assumptions about the rewards. However, if all rewards are non-negative, then under mild assumptions, we can prove the following bias reduction result.

**Proposition 4** (A sufficient condition for bias reduction)**.** *If Assumption 1 holds (but Assumption 4 is violated), $\bar{R}(s,a) \geq 0$ for all $s \in \mathcal{S}, a \in \mathcal{A}$, and there exists $(s,a)$ such that $\pi_b(a|s) = 0$, $\pi_{b^+}(a|s) > 0$, $\pi_e(a|s) > 0$, $\bar{R}(s,a) > 0$, then $|\mathrm{Bias}[\hat{v}^{\text{C-IS}}]| < |\mathrm{Bias}[\hat{v}^{\text{IS}}]|$.*

*Proof.* $|\mathrm{Bias}[\hat{v}^{\text{IS}}]| - |\mathrm{Bias}[\hat{v}^{\text{C-IS}}]|$

$$= \left|\mathop{\mathbb{E}}_{s\sim d_1}\left[-\sum_{a\in\mathcal{U}(s,\pi_b)}\pi_e(a|s)\bar{R}(s,a)\right]\right| - \left|\mathop{\mathbb{E}}_{s\sim d_1}\left[-\sum_{a\in\mathcal{U}(s,\pi_{b^+})}\pi_e(a|s)\bar{R}(s,a)\right]\right|$$

$$= \mathop{\mathbb{E}}_{s\sim d_1}\left[\sum_{a\in\mathcal{U}(s,\pi_b)}\pi_e(a|s)\bar{R}(s,a)\right] - \mathop{\mathbb{E}}_{s\sim d_1}\left[\sum_{a\in\mathcal{U}(s,\pi_{b^+})}\pi_e(a|s)\bar{R}(s,a)\right]$$

$$= \mathop{\mathbb{E}}_{s\sim d_1}\left[\sum_{a\in\mathcal{U}(s,\pi_b)\setminus\mathcal{U}(s,\pi_{b^+})}\pi_e(a|s)\bar{R}(s,a)\right] > 0 \qquad \square$$

Lastly, we note a useful corollary of Theorem 1.

**Corollary 5** (Expectation of augmented importance ratios)**.** *Let $\rho_W^+ = w^a\rho^a + \sum_{\tilde{a}\in\mathcal{A}\setminus\{a\}}w^{\tilde{a}}\rho^{\tilde{a}}$ given $\tau$ and $\mathbf{w}$. Under Assumption 4, $\mathbb{E}[\rho_W^+] = 1$.*

*Proof of Corollary 5.* Starting with Theorem 1 and substituting $R(s,a) = G(s,a) = 1$ as the constant reward and annotation for all $s \in \mathcal{S}, a \in \mathcal{A}$, we have: $\mathbb{E}_{\tau^+}[w^a\rho^a + \sum_{\tilde{a}\in\mathcal{A}\setminus\{a\}}w^{\tilde{a}}\rho^{\tilde{a}}] = \mathbb{E}_{s\sim d_1}\mathbb{E}_{\tilde{a}\sim\pi_e(s)}[1] = 1$. $\qquad \square$

## C.3 C-IS: Variance Analyses

For variance analyses, we focus on the scenario where Assumptions 1 and 4 hold and bias is zero. Below, we state the variance decomposition results under a few assumptions about the weighting scheme and the annotation variance.

**Theorem 13** (Variance of C-IS). *Let $\rho^+(a|s) = \rho^a = \frac{\pi_e(a|s)}{\pi_{b^+}(a|s)}$ be the importance ratio defined using the augmented behavior policy, and assume variance of the annotation function and the variance of the reward function are related by $\sigma_R(s,a)^2 = \sigma_R(s,a)^2 + \Delta_\sigma(s,a)$ where $\Delta_\sigma(s,a) \in \mathbb{R}$, then under Assumptions 1 and 4,*

$$
\begin{aligned}
\mathbb{V}[\hat{v}^{\text{C-IS}}] = {}& \mathbb{V}_{s\sim d_1}[V^{\pi_e}(s)] + \mathbb{E}_{s\sim d_1}\Big[\mathbb{V}_{a\sim\pi_b(s)}\big[\textstyle\sum_{\tilde{a}\in\mathcal{A}} \rho^+(\tilde{a}|s)\,\bar{W}(\tilde{a}|s,a)\,\bar{R}(s,\tilde{a})\big]\Big] \\
& + \mathbb{E}_{s\sim d_1}\mathbb{E}_{a\sim\pi_b(s)}\Big[\textstyle\sum_{\tilde{a}\in\mathcal{A}} \rho^+(\tilde{a}|s)^2\,\bar{W}(\tilde{a}|s,a)^2\,\sigma_R(s,\tilde{a})^2\Big] \\
& + \mathbb{E}_{s\sim d_1}\mathbb{E}_{a\sim\pi_b(s)}\Big[\textstyle\sum_{\tilde{a}\in\mathcal{A}\setminus\{a\}} \rho^+(\tilde{a}|s)^2\bar{W}(\tilde{a}|s,a)^2\Delta_\sigma(s,\tilde{a})\Big] \\
& + \mathbb{E}_{s\sim d_1}\mathbb{E}_{a\sim\pi_b(s)}\Big[\textstyle\sum_{\tilde{a}\in\mathcal{A}} \rho^+(\tilde{a}|s)^2\,\bar{R}(s,\tilde{a})^2\,\sigma_W(\tilde{a}|s,a)^2\Big] \qquad (3) \\
& + \mathbb{E}_{s\sim d_1}\mathbb{E}_{a\sim\pi_b(s)}\Big[\textstyle\sum_{\tilde{a}\in\mathcal{A}} \rho^+(\tilde{a}|s)^2\,\sigma_R(s,\tilde{a})^2\,\sigma_W(\tilde{a}|s,a)^2\Big] \\
& + \mathbb{E}_{s\sim d_1}\mathbb{E}_{a\sim\pi_b(s)}\Big[C(s,a)\Big] \\
& + \mathbb{E}_{s\sim d_1}\mathbb{E}_{a\sim\pi_b(s)}\Big[\textstyle\sum_{\tilde{a}\in\mathcal{A}\setminus\{a\}} \rho^+(\tilde{a}|s)^2\Delta_\sigma(s,\tilde{a})\sigma_W(\tilde{a}|s,a)^2\Big]
\end{aligned}
$$

*where* $C(s,a) = 2 \displaystyle\sum_{\substack{a_i,a_j\in\mathcal{A} \\ a_i\neq a_j}}^{a_i\neq a_j} \rho^+(a_i|s)\,\rho^+(a_j|s)\,\bar{R}(s,a_i)\,\bar{R}(s,a_j)\,\mathrm{Cov}\big(W(a_i|s,a),W(a_j|s,a)\big).$

**Corollary 14** (Variance of C\*-IS). *Assuming $\sigma_G(s,a)^2 = \sigma_R(s,a)^2 + \Delta_\sigma(s,a)$ where $\Delta_\sigma(s,a) \in \mathbb{R}$, then under Assumptions 1 and 4,*

$$
\begin{aligned}
\mathbb{V}[\hat{v}^{\text{C*-IS}}] = {}& \mathbb{V}_{s\sim d_1}[V^{\pi_e}(s)] + \mathbb{E}_{s\sim d_1}\mathbb{E}_{a\sim\pi_b(s)}\big[\pi_b(a|s)\,\rho(a|s)^2\,\sigma_R(s,a)^2\big] \\
& + \mathbb{E}_{s\sim d_1}\big[\textstyle\sum_{\tilde{a}\in\mathcal{A}\setminus\{a\}} \pi_e(a|s)^2\Delta_\sigma(s,a)\big]
\end{aligned}
$$

*where* $\rho(a|s) = \frac{\pi_e(a|s)}{\pi_b(a|s)}$ *is the importance ratio under the original behavior policy.*

*Remark.* The variance decomposition in Theorem 13 contains eight terms. The first three terms correspond to the three terms in the IS variance decomposition in Eqn. (1), the fourth and eighth terms are related to the variance difference between annotations and rewards, whereas the remaining three terms are all related to the variance (and covariance) of the weight distribution $W(\cdot|s,a)$. As we will demonstrate empirically (Appendix E.1), depending on the weight distributions, variance of C-IS may actually be larger than that of IS. While it is difficult to guarantee a general variance reduction, we provide intuition for a few special cases. (i) If the weights are constant (i.e., $W(\cdot|s,a)$ is the same value for all instantiations of $(s,a)$ in the data), then the last four terms will all vanish to zero. (ii) When the annotation function and reward function have the same variance, $\Delta_\sigma(s,a) = 0$, the fourth and eight terms vanish to zero; with this assumption, we can straightforwardly obtain Theorem 6 from Corollary 14. (iii) Suppose for each state-action pair $(s,a)$, we set factual weights $W(a|s,a) = 1$ and counterfactual weights $W(\tilde{a}|s,a) = 0$. Then, we effectively ignore all counterfactual annotations, and the variance of C-IS becomes identical to that of IS.

We prove the most general case for Theorem 13 first, and then derive Corollary 14 as a special case.

*Proof of Theorem 13.* We apply the law of total variance:

$$
\mathbb{V}[\hat{v}^{\text{C-IS}}] = \mathbb{V}_{\substack{s\sim d_1,a\sim\pi_b(s),r\sim R(s,a) \\ \boldsymbol{g}\sim G(s,\cdot),\boldsymbol{w}\sim W(s,a)}} \Bigg[w^a\rho^a r + \sum_{\tilde{a}\in\mathcal{A}\setminus\{a\}} w^{\tilde{a}}\rho^{\tilde{a}}g^{\tilde{a}}\Bigg]
$$

$$
= \underbrace{\mathbb{V}_{s\sim d_1}\Bigg[\mathbb{E}_{\substack{a\sim\pi_b(s),r\sim R(s,a) \\ \boldsymbol{g}\sim G(s,\cdot),\boldsymbol{w}\sim W(s,a)}}\Big[w^a\rho^a r + \sum_{\tilde{a}\in\mathcal{A}\setminus\{a\}} w^{\tilde{a}}\rho^{\tilde{a}}g^{\tilde{a}}\Big]\Bigg]}_{(1)} + \underbrace{\mathbb{E}_{s\sim d_1}\Bigg[\mathbb{V}_{\substack{a\sim\pi_b(s),r\sim R(s,a) \\ \boldsymbol{g}\sim G(s,\cdot),\boldsymbol{w}\sim W(s,a)}}\Big[w^a\rho^a r + \sum_{\tilde{a}\in\mathcal{A}\setminus\{a\}} w^{\tilde{a}}\rho^{\tilde{a}}g^{\tilde{a}}\Big]\Bigg]}_{(1')}
$$

As shown in the bias analyses, we have $(1) = \mathbb{V}_{s \sim d_1}[V^{\pi_e}(s)]$. We further apply the law of total variance on $(1')$:

$$(1') = \mathop{\mathbb{E}}_{s \sim d_1}\left[\mathop{\mathbb{V}}_{a \sim \pi_b(s)}\left[\mathop{\mathbb{E}}_{\substack{r \sim R(s,a), \boldsymbol{g} \sim G(s,\cdot) \\ \boldsymbol{w} \sim W(s,a)}}\left[w^a \rho^a r + \sum_{\tilde{a} \in \mathcal{A} \backslash \{a\}} w^{\tilde{a}} \rho^{\tilde{a}} g^{\tilde{a}}\right]\right]\right] \dots (2)$$

$$+ \mathop{\mathbb{E}}_{s \sim d_1}\left[\mathop{\mathbb{E}}_{a \sim \pi_b(s)}\left[\mathop{\mathbb{V}}_{\substack{r \sim R(s,a), \boldsymbol{g} \sim G(s,\cdot) \\ \boldsymbol{w} \sim W(s,a)}}\left[w^a \rho^a r + \sum_{\tilde{a} \in \mathcal{A} \backslash \{a\}} w^{\tilde{a}} \rho^{\tilde{a}} g^{\tilde{a}}\right]\right]\right] \dots (2')$$

Since $(r, \boldsymbol{g})$ and $\boldsymbol{w}$ are conditionally independent given $(s, a)$,

$$(2) = \mathop{\mathbb{E}}_{s \sim d_1}\left[\mathop{\mathbb{V}}_{a \sim \pi_b(s)}\left[\rho^a \mathop{\mathbb{E}}_{\substack{w^a \sim W(a|s,a) \\ r \sim R(s,a)}}[w^a r] + \sum_{\tilde{a} \in \mathcal{A} \backslash \{a\}} \rho^{\tilde{a}} \mathop{\mathbb{E}}_{\substack{w^{\tilde{a}} \sim W(\tilde{a}|s,a) \\ g^{\tilde{a}} \sim G(s,\tilde{a})}}[w^{\tilde{a}} g^{\tilde{a}}]\right]\right]$$

$$= \mathop{\mathbb{E}}_{s \sim d_1}\left[\mathop{\mathbb{V}}_{a \sim \pi_b(s)}\left[\rho^a \mathop{\mathbb{E}}_{w^a \sim W(a|s,a)}[w^a] \mathop{\mathbb{E}}_{r \sim R(s,a)}[r] + \sum_{\tilde{a} \in \mathcal{A} \backslash \{a\}} \rho^{\tilde{a}} \mathop{\mathbb{E}}_{w^{\tilde{a}} \sim W(\tilde{a}|s,a)}[w^{\tilde{a}}] \mathop{\mathbb{E}}_{g^{\tilde{a}} \sim G(s,\tilde{a})}[g^{\tilde{a}}]\right]\right]$$

$$= \mathop{\mathbb{E}}_{s \sim d_1}\left[\mathop{\mathbb{V}}_{a \sim \pi_b(s)}\left[\sum_{\tilde{a} \in \mathcal{A}} \rho^+(a|s) \bar{W}(\tilde{a}|s, a) \bar{R}(s, \tilde{a})\right]\right]$$

where in the last step, we apply Assumption 1 to combine the expressions involving $\mathbb{E}_{r \sim R(s,a)}[r]$ and $\mathbb{E}_{g^{\tilde{a}} \sim G(s,\tilde{a})}[g^{\tilde{a}}]$ (both are equal to $\bar{R}(s, \tilde{a})$) into a single summation over $\tilde{a} \in \mathcal{A}$.

We further apply the law of total variance on $(2')$:

$$(2') = \mathop{\mathbb{E}}_{s \sim d_1} \mathop{\mathbb{E}}_{a \sim \pi_b(s)}\left[\mathop{\mathbb{V}}_{\substack{r \sim R(s,a) \\ \boldsymbol{g} \sim G(s,\cdot)}} \mathop{\mathbb{E}}_{\boldsymbol{w} \sim W(s,a)}\left[w^a \rho^a r + \sum_{\tilde{a} \in \mathcal{A} \backslash \{a\}} w^{\tilde{a}} \rho^{\tilde{a}} g^{\tilde{a}}\right]\right] \dots (3)$$

$$+ \mathop{\mathbb{E}}_{s \sim d_1} \mathop{\mathbb{E}}_{a \sim \pi_b(s)}\left[\mathop{\mathbb{E}}_{\substack{r \sim R(s,a) \\ \boldsymbol{g} \sim G(s,\cdot)}} \mathop{\mathbb{V}}_{\boldsymbol{w} \sim W(s,a)}\left[w^a \rho^a r + \sum_{\tilde{a} \in \mathcal{A} \backslash \{a\}} w^{\tilde{a}} \rho^{\tilde{a}} g^{\tilde{a}}\right]\right] \dots (3')$$

Then we have

$$(3) = \mathop{\mathbb{E}}_{s \sim d_1} \mathop{\mathbb{E}}_{a \sim \pi_b(s)}\left[\mathop{\mathbb{V}}_{r \sim R(s,a)}\left[\rho^a \mathop{\mathbb{E}}_{w^a \sim W(a|s,a)}[w^a] r\right] + \sum_{\tilde{a} \in \mathcal{A} \backslash \{a\}} \mathop{\mathbb{V}}_{\boldsymbol{g} \sim G(s,\cdot)}\left[\rho^{\tilde{a}} \mathop{\mathbb{E}}_{w^{\tilde{a}} \sim W(\tilde{a}|s,a)}[w^{\tilde{a}}] g^{\tilde{a}}\right]\right]$$

$$= \mathop{\mathbb{E}}_{s \sim d_1} \mathop{\mathbb{E}}_{a \sim \pi_b(s)}\left[\rho^+(a|s)^2 \bar{W}(a|s, a)^2 \mathop{\mathbb{V}}_{r \sim R(s,a)}[r] + \sum_{\tilde{a} \in \mathcal{A} \backslash \{a\}} \rho^+(\tilde{a}|s)^2 \bar{W}(\tilde{a}|s, a)^2 \mathop{\mathbb{V}}_{g^{\tilde{a}} \sim G(s,\tilde{a})}[g^{\tilde{a}}]\right]$$

$$= \mathop{\mathbb{E}}_{s \sim d_1} \mathop{\mathbb{E}}_{a \sim \pi_b(s)}\left[\sum_{\tilde{a} \in \mathcal{A}} \rho^+(\tilde{a}|s)^2 \bar{W}(\tilde{a}|s, a)^2 \sigma_R(s, \tilde{a})^2\right] + \mathop{\mathbb{E}}_{s \sim d_1} \mathop{\mathbb{E}}_{a \sim \pi_b(s)}\left[\sum_{\tilde{a} \in \mathcal{A} \backslash \{a\}} \rho^+(\tilde{a}|s)^2 \bar{W}(\tilde{a}|s, a)^2 \Delta_\sigma(s, \tilde{a})\right]$$

where in the last step we substitute $\sigma_G(s, \tilde{a})^2 = \sigma_R(s, \tilde{a})^2 + \Delta_\sigma(s, \tilde{a})$.

Letting $g^a = r$ for clarity, we have

$$(3') = \mathop{\mathbb{E}}_{s \sim d_1} \mathop{\mathbb{E}}_{a \sim \pi_b(s)} \mathop{\mathbb{E}}_{\substack{r \sim R(s,a) \\ \boldsymbol{g} \sim G(s,\cdot)}}\left[\sum_{\tilde{a} \in \mathcal{A}} \mathop{\mathbb{V}}_{w^{\tilde{a}} \sim W(\tilde{a}|s,a)}[\rho^{\tilde{a}} g^{\tilde{a}} w^{\tilde{a}}] + 2 \sum_{\substack{a_i \neq a_j \\ a_i, a_j \in \mathcal{A}}} \mathop{\text{Cov}}_{\boldsymbol{w} \sim W(s,a)}(\rho^{a_i} g^{a_i} w^{a_i}, \rho^{a_j} g^{a_j} w^{a_j})\right]$$

$$
= \underset{s\sim d_1}{\mathbb{E}}\, \underset{a\sim \pi_b(s)}{\mathbb{E}}\, \underset{\substack{r\sim R(s,a)\\ g\sim G(s,\cdot)}}{\mathbb{E}} \left[ \sum_{\tilde{a}\in\mathcal{A}} (\rho^{\tilde{a}})^2 (g^{\tilde{a}})^2 \mathbb{V}_{w^{\tilde{a}}\sim W(\tilde{a}|s,a)}[w^{\tilde{a}}] \right.
$$

$$
\left. + 2 \sum_{\substack{a_i,a_j\in\mathcal{A}}}^{a_i\neq a_j} (\rho^{a_i})(\rho^{a_j})(g^{a_i})(g^{a_j}) \times \mathrm{Cov}\Big( W(a_i|s,a), W(a_j|s,a) \Big) \right]
$$

$$
= \underset{s\sim d_1}{\mathbb{E}}\, \underset{a\sim \pi_b(s)}{\mathbb{E}} \left[ \sum_{\tilde{a}\in\mathcal{A}} \rho^+(\tilde{a}|s)^2\Big(\bar{R}(s,\tilde{a})^2 + \sigma_R(s,\tilde{a})^2\Big)\sigma_W(\tilde{a}|s,a)^2 + \sum_{\tilde{a}\in\mathcal{A}\setminus\{a\}} \rho^+(\tilde{a}|s)^2\Delta_\sigma(s,\tilde{a})\sigma_W(\tilde{a}|s,a)^2 \right.
$$

$$
\left. + 2 \sum_{\substack{a_i,a_j\in\mathcal{A}}}^{a_i\neq a_j} \rho^+(a_i|s)\,\rho^+(a_j|s)\,\bar{R}(s,a_i)\,\bar{R}(s,a_j) \times \mathrm{Cov}\Big( W(a_i|s,a), W(a_j|s,a) \Big) \right]
$$

$$
= \underbrace{\underset{s\sim d_1}{\mathbb{E}}\, \underset{a\sim \pi_b(s)}{\mathbb{E}} \left[ \sum_{\tilde{a}\in\mathcal{A}} \rho^+(\tilde{a}|s)^2\bar{R}(s,\tilde{a})^2\sigma_W(\tilde{a}|s,a)^2 \right]}_{(4)} + \underbrace{\underset{s\sim d_1}{\mathbb{E}}\, \underset{a\sim \pi_b(s)}{\mathbb{E}} \left[ \sum_{\tilde{a}\in\mathcal{A}} \rho^+(\tilde{a}|s)^2\sigma_R(s,\tilde{a})^2\sigma_W(\tilde{a}|s,a)^2 \right]}_{(5)}
$$

$$
+ \underbrace{\underset{s\sim d_1}{\mathbb{E}}\, \underset{a\sim \pi_b(s)}{\mathbb{E}} \left[ \sum_{\tilde{a}\in\mathcal{A}\setminus\{a\}} \rho^+(\tilde{a}|s)^2\Delta_\sigma(s,\tilde{a})\sigma_W(\tilde{a}|s,a)^2 \right]}_{(6)}
$$

$$
+ \underbrace{\underset{s\sim d_1}{\mathbb{E}}\, \underset{a\sim \pi_b(s)}{\mathbb{E}} \left[ 2 \sum_{\substack{a_i,a_j\in\mathcal{A}}}^{a_i\neq a_j} \rho^+(a_i|s)\,\rho^+(a_j|s)\,\bar{R}(s,a_i)\,\bar{R}(s,a_j) \times \mathrm{Cov}\Big( W(a_i|s,a), W(a_j|s,a) \Big) \right]}_{(7)}
$$

Putting together expressions (1) through (7), we have the desired decomposition for $\mathbb{V}[\hat{v}^{\text{C-IS}}]$. $\qquad\square$

*Proof of Corollary 14.* Here we derive the variance of C\*-IS, which is C-IS with $W(\tilde{a}|s,a) = |\mathcal{A}|^{-1}$. Since the weights are constant, expressions $(4)(5)(6)(7)$ all vanish to zero because the variance and covariance associated with $W(\cdot|s,a)$ are both zero. We focus on simplifying the second, third, and fourth terms in the variance decomposition (which correspond to expressions (2) and (3) in the proof above; we denote the two parts of (3) as (3.1) and (3.2)). First, note that

$$
\begin{aligned}
\pi_{b^+}(a|s) &= \sum_{\check{a}\in\mathcal{A}} \bar{W}(a|s,\check{a})\pi_b(\check{a}|s)\\
&= \sum_{\check{a}\in\mathcal{A}} |\mathcal{A}|^{-1}\pi_b(\check{a}|s)\\
&= |\mathcal{A}|^{-1}\left(\sum_{\check{a}\in\mathcal{A}} \pi_b(\check{a}|s)\right)\\
&= |\mathcal{A}|^{-1}
\end{aligned}
$$

Then,

$$
\begin{aligned}
(2) &= \mathbb{E}_{s\sim d_1}\left[\mathbb{V}_{a\sim\pi_b(s)}\left[\sum_{\tilde{a}\in\mathcal{A}} \rho^+(\tilde{a}|s)\,\bar{W}(\tilde{a}|s,a)\,\bar{R}(s,\tilde{a})\right]\right]\\
&= \mathbb{E}_{s\sim d_1}\left[\mathbb{V}_{a\sim\pi_b(s)}\left[\sum_{\tilde{a}\in\mathcal{A}} \tfrac{\pi_e(\tilde{a}|s)}{1/|\mathcal{A}|}\,(1/|\mathcal{A}|)\,\bar{R}(s,\tilde{a})\right]\right]\\
&= \mathbb{E}_{s\sim d_1}\left[\mathbb{V}_{a\sim\pi_b(s)}\left[\sum_{\tilde{a}\in\mathcal{A}} \pi_e(\tilde{a}|s)\bar{R}(s,\tilde{a})\right]\right]\\
&= \mathbb{E}_{s\sim d_1}\left[\mathbb{V}_{a\sim\pi_b(s)}\left[V^{\pi_e}(s)\right]\right]\\
&= 0
\end{aligned}
$$

$$
\begin{aligned}
(3.1) &= \mathbb{E}_{s\sim d_1}\mathbb{E}_{a\sim\pi_b(s)}\left[\sum_{\tilde{a}\in\mathcal{A}} \rho^+(\tilde{a}|s)^2\,\bar{W}(\tilde{a}|s,a)^2\,\sigma_R(s,\tilde{a})^2\right]\\
&= \mathbb{E}_{s\sim d_1}\mathbb{E}_{a\sim\pi_b(s)}\left[\sum_{\tilde{a}\in\mathcal{A}} \tfrac{\pi_e(\tilde{a}|s)^2}{(1/|\mathcal{A}|)^2}\,(1/|\mathcal{A}|)^2\,\sigma_R(s,\tilde{a})^2\right]
\end{aligned}
$$

$$= \mathbb{E}_{s \sim d_1} \Big[ \sum_{a \in \mathcal{A}} \pi_b(a|s) \sum_{\tilde{a} \in \mathcal{A}} \pi_e(\tilde{a}|s)^2 \, \sigma_R(s, \tilde{a})^2 \Big]$$

$$= \mathbb{E}_{s \sim d_1} \Big[ \sum_{\tilde{a} \in \mathcal{A}} \big( \sum_{a \in \mathcal{A}} \pi_b(a|s) \big) \, \pi_e(\tilde{a}|s)^2 \, \sigma_R(s, \tilde{a})^2 \Big]$$

$$= \mathbb{E}_{s \sim d_1} \Big[ \sum_{\tilde{a} \in \mathcal{A}} \pi_e(\tilde{a}|s)^2 \, \sigma_R(s, \tilde{a})^2 \Big]$$

$$= \mathbb{E}_{s \sim d_1} \Big[ \sum_{a \in \mathcal{A}} \pi_b(a|s)^2 \, \rho(a|s)^2 \, \sigma_R(s, a)^2 \Big]$$

$$= \mathbb{E}_{s \sim d_1} \mathbb{E}_{a \sim \pi_b(s)} \Big[ \pi_b(a|s) \, \rho(a|s)^2 \, \sigma_R(s, a)^2 \Big]$$

$$(3.2) = \mathbb{E}_{s \sim d_1} \mathbb{E}_{a \sim \pi_b(s)} \Big[ \sum_{\tilde{a} \in \mathcal{A} \backslash \{a\}} \rho^+(\tilde{a}|s)^2 \, \bar{W}(\tilde{a}|s, a)^2 \, \Delta_\sigma(s, \tilde{a}) \Big]$$

$$= \mathbb{E}_{s \sim d_1} \mathbb{E}_{a \sim \pi_b(s)} \Big[ \sum_{\tilde{a} \in \mathcal{A} \backslash \{a\}} \frac{\pi_e(\tilde{a}|s)^2}{(1/|\mathcal{A}|)^2} \, (1/|\mathcal{A}|)^2 \, \Delta_\sigma(s, \tilde{a}) \Big]$$

$$= \mathbb{E}_{s \sim d_1} \Big[ \sum_{a \in \mathcal{A}} \pi_b(a|s) \sum_{\tilde{a} \in \mathcal{A} \backslash \{a\}} \pi_e(\tilde{a}|s)^2 \, \Delta_\sigma(s, \tilde{a}) \Big]$$

$$= \mathbb{E}_{s \sim d_1} \Big[ \sum_{\tilde{a} \in \mathcal{A} \backslash \{a\}} \big( \sum_{a \in \mathcal{A}} \pi_b(a|s) \big) \, \pi_e(\tilde{a}|s)^2 \, \Delta_\sigma(s, \tilde{a}) \Big]$$

$$= \mathbb{E}_{s \sim d_1} \Big[ \sum_{\tilde{a} \in \mathcal{A} \backslash \{a\}} \pi_e(\tilde{a}|s)^2 \, \Delta_\sigma(s, \tilde{a}) \Big]$$

$\square$

## C.4 C-PDIS: Bias Analyses

**Theorem 8** (Unbiasedness of C-PDIS). *In the MDP setting, when both Assumptions 2 and 4 hold, the C-PDIS estimator is unbiased, $\mathbb{E}[\hat{v}^{\text{C-PDIS}}] = v(\pi_e)$.*

First, we introduce a few definitions useful for the proof. Let the $t$-step state distribution be denoted by $d_t^\pi(s) = \Pr(s_t = s \mid s_1 \sim d_1, a_{t'} \sim \pi(s_{t'}))$. Similarly, the $t$-step state-action distribution is $d_t^\pi(s, a) = \Pr(s_t = s, a_t = a \mid s_1 \sim d_1, a_{t'} \sim \pi(s_{t'}))$. Note that $d_{t+1}^\pi(s') = \mathbb{E}_{s, a \sim d_t^\pi}[p(s'|s, a)]$. Let $d_t = d_t^{\pi_b}$ denote the distribution under the behavior policy. Recall the horizon-$t$ value functions: $V_{t:T}^{\pi_e}(s) = \mathbb{E}_{\pi_e}[\sum_{t'=t}^T \gamma^{t'-1} r_{t'}|s_t = s]$, $Q_{t:T}^{\pi_e}(s, a) = \mathbb{E}_{\pi_e}[\sum_{t'=t}^T \gamma^{t'-1} r_{t'}|s_t = s, a_t = a]$. Based on the Bellman equation, $Q_{t:T}^{\pi_e}(s, a) = r(s, a) + \gamma \mathbb{E}_{s' \sim p(s, a)}[V_{(t+1):T}^{\pi_e}(s')]$. Note that $v(\pi_e) = \mathbb{E}_{s \sim d_1}[V_{1:T}^{\pi_e}(s)]$. Also recall Assumption 2, the perfect annotation function in the MDP setting satisfies $\mathbb{E}_{g \sim G_t(s, a)}[g] = Q_{t:T}^{\pi_e}(s, a)$ for $(s, a)$ at step $t$ of the trajectory.

Recall the recursive definition of C-PDIS (Definition 4): $\hat{v}^{\text{C-PDIS}} = v_T$, with $v_0 = 0$, $v_{T-t+1} = w_t^{a_t} \rho_t^{a_t} (r_t + \gamma v_{T-t}) + \sum_{\tilde{a} \in \mathcal{A} \backslash \{a_t\}} w_t^{\tilde{a}} \rho_t^{\tilde{a}} q_t^{\tilde{a}}$ for $t = T...1$, where $\rho_t^{\tilde{a}} = \frac{\pi_e(\tilde{a}|s_t)}{\pi_{b^+}(\tilde{a}|s_t)}$.

*Proof.* We show this via backward induction on a sequence of horizon-$t$ value functions of $\pi_e$ denoted by $V_{t:T}^{\pi_e}(s)$. The goal is to show that $\mathbb{E}_{\tau^+, \boldsymbol{w}}[v_{T-t}] = \mathbb{E}_{s \sim d_{t+1}}[V_{(t+1):T}^{\pi_e}(s)]$ for all $t = T...0$ (for clarity, the subscript of expectation of the estimator may be omitted and assumed to be $\mathbb{E}_{\tau^+, \boldsymbol{w}}$ with $\tau$ generated by the behavior policy $\pi_b$ unless otherwise specified).

*Base case.* It is trivially true that $\mathbb{E}[v_0] = \mathbb{E}_{s \sim d_{T+1}}[V_{(T+1):T}^{\pi_e}(s)] = 0$ since there are no more steps after $t = T$ and $s \sim d_{T+1}$ can be seen as a dummy absorbing state.

*Inductive step.* Suppose $\mathbb{E}[v_{T-t}] = \mathbb{E}_{s \sim d_{t+1}}[V_{(t+1):T}^{\pi_e}(s)]$ holds. For factual state-action pair $(s, a) \sim d_t$ occurring at step $t$, we have

$$\mathbb{E}_{s, a \sim d_t}[r_t + \gamma v_{T-t}] = \mathbb{E}_{s, a \sim d_t}\Big[ r_t + \gamma \mathbb{E}_{s' \sim d_{t+1}}[V_{(t+1):T}^{\pi_e}(s')] \Big]$$

$$= \mathbb{E}_{s, a \sim d_t}\Big[ r_t + \gamma \mathbb{E}_{s' \sim p(\cdot|s, a)}[V_{(t+1):T}^{\pi_e}(s')] \Big]$$

$$= \mathbb{E}_{s, a \sim d_t}[Q_{t:T}^{\pi_e}(s, a)]$$

To show the case for $t - 1$,

$$\mathbb{E}[v_{T-t+1}] = \mathbb{E}\left[w_t^{a_t}\rho_t^{a_t}(r_t + \gamma v_{T-t}) + \sum_{\tilde{a}\in\mathcal{A}\setminus\{a_t\}} w_t^{\tilde{a}}\rho_t^{\tilde{a}}q_t^{\tilde{a}}\right]$$

$$= \mathop{\mathbb{E}}_{s\sim d_t}\mathop{\mathbb{E}}_{a\sim\pi_b(s)}\left[\mathbb{E}_{w_t^a\sim W(a|s,a)}[w_t^a]\frac{\pi_e(a|s)}{\pi_{b^+}(a|s)}\mathbb{E}_{s,a\sim d_t}[r_t + \gamma v_{T-t}]\right.$$

$$\left.+ \sum_{\tilde{a}\in\mathcal{A}\setminus\{a\}}\mathbb{E}_{w_t^{\tilde{a}}\sim W(\tilde{a}|s,a)}[w_t^{\tilde{a}}]\frac{\pi_e(\tilde{a}|s)}{\pi_{b^+}(\tilde{a}|s)}\mathbb{E}_{g_t^{\tilde{a}}\sim G_t(s,\tilde{a})}[g_t^{\tilde{a}}]\right]$$

$$\overset{(1)}{=} \mathop{\mathbb{E}}_{s\sim d_t}\left[\sum_{a\in\mathcal{A}}\pi_b(a|s)\left(\sum_{\tilde{a}\in\mathcal{A}}\bar{W}(\tilde{a}|s,a)\frac{\pi_e(\tilde{a}|s)}{\pi_{b^+}(\tilde{a}|s)}Q_{t:T}^{\pi_e}(s,\tilde{a})\right)\right]$$

$$\overset{(2)}{=} \mathop{\mathbb{E}}_{s\sim d_t}\left[\sum_{\tilde{a}\in\mathcal{A}}\left(\sum_{a\in\mathcal{A}}\pi_b(a|s)\bar{W}(\tilde{a}|s,a)\frac{\pi_e(\tilde{a}|s)}{\pi_{b^+}(\tilde{a}|s)}Q_{t:T}^{\pi_e}(s,\tilde{a})\right)\right]$$

$$\overset{(3)}{=} \mathop{\mathbb{E}}_{s\sim d_t}\left[\sum_{\tilde{a}\in\mathcal{A}}\left(\left(\sum_{a\in\mathcal{A}}\pi_b(a|s)\bar{W}(\tilde{a}|s,a)\right)\frac{\pi_e(\tilde{a}|s)}{\pi_{b^+}(\tilde{a}|s)}Q_{t:T}^{\pi_e}(s,\tilde{a})\right)\right]$$

$$\overset{(4)}{=} \mathop{\mathbb{E}}_{s\sim d_t}\left[\sum_{\tilde{a}\in\mathcal{A}}\cancel{\pi_{b^+}(\tilde{a}|s)}\frac{\pi_e(\tilde{a}|s)}{\cancel{\pi_{b^+}(\tilde{a}|s)}}Q_{t:T}^{\pi_e}(s,\tilde{a})\right]$$

$$= \mathop{\mathbb{E}}_{s\sim d_t}\left[\sum_{\tilde{a}\in\mathcal{A}}\pi_e(\tilde{a}|s)Q_{t:T}^{\pi_e}(s,\tilde{a})\right]$$

$$= \mathop{\mathbb{E}}_{s\sim d_t}\mathop{\mathbb{E}}_{\tilde{a}\sim\pi_e}[Q_{t:T}^{\pi_e}(s,\tilde{a})] = \mathop{\mathbb{E}}_{s\sim d_t}[V_{t:T}^{\pi_e}(s)]$$

where in (1) we replace $\mathbb{E}_{g_t^{\tilde{a}}\sim G_t(s,\tilde{a})}[g_t^{\tilde{a}}]$ with $Q_{t:T}^{\pi_e}(s,\tilde{a})$ following Assumption 2 and combine it with $\mathbb{E}_{s,a\sim d_t}[r_t + \gamma v_{T-t}] = Q_{t:T}^{\pi_e}(s,\tilde{a})$ (as shown above) in a single summation over $\tilde{a}\in\mathcal{A}$, in (2) we swap the order of summations, in (3) we take out common factors that do not depend on the inner summation $a$, and in (4) we use Definition 1 for $\pi_{b^+}(\tilde{a}|s)$, which cancels out the denominator in the importance ratio on the next line. The proof techniques are similar to that in Appendix C.2.

Since $\mathbb{E}[v_{T-t}] = \mathbb{E}_{s\sim d_{h+1}}[V_{(t+1):T}^{\pi_e}(s)]$ implies $\mathbb{E}[v_{T-t+1}] = \mathbb{E}_{s\sim d_t}[V_{t:T}^{\pi_e}(s)]$, and the base case $\mathbb{E}[v_0] = \mathbb{E}_{s\sim d_{T+1}}[V_{(T+1):T}^{\pi_e}(s)]$ is true, by mathematical induction, we have the desired property that $\mathbb{E}[\hat{v}^{\text{C-PDIS}}] = \mathbb{E}[v_T] = \mathbb{E}_{s\sim d_1}[V_{1:T}^{\pi_e}(s)] = v(\pi_e)$. $\qquad\square$

# D Extended Discussions on Practical Implications

## D.1 Using an Approximate MDP Model to Correct Annotation Bias

In the sequential RL setting, if the annotation function reflects the expected returns under the behavior policy, i.e. $G_t = Q_{t:T}^{\pi_b}$, then annotations have a nonzero bias of $\epsilon_{G_t} = Q_{t:T}^{\pi_b} - Q_{t:T}^{\pi_e} \neq 0$ (since $Q_{t:T}^{\pi_e} \neq Q_{t:T}^{\pi_b}$ for meaningful OPE problems where $\pi_e \neq \pi_b$), and consequently, C-PDIS using such annotations becomes biased. While such annotations may aid in model selection (as shown in experiments), here we further propose a procedure to convert the annotations so that they better reflect the evaluation policy.

From the offline data, we first learn an approximate model of the MDP $\hat{\mathcal{M}}$, which includes the transition model $\hat{p}(s'|s, a)$ and reward model $\hat{r}(s, a)$. $\hat{\mathcal{M}}$ is then used to evaluate both $\pi_b$ and $\pi_e$ using a model-based approach, leading to estimated horizon-specific Q-functions, $\hat{Q}_{t:T}^{\pi_b}$ and $\hat{Q}_{t:T}^{\pi_e}$. We then approximate the annotation error for state $s_t$ occurring at horizon $t$ and counterfactual action $\tilde{a}$ as $\hat{\epsilon}_{G_t}(s_t, \tilde{a}) = \hat{Q}_{t:T}^{\pi_b}(s_t, \tilde{a}) - \hat{Q}_{t:T}^{\pi_e}(s_t, \tilde{a})$. Given the factual state-action pair $(s_t^{(i)}, a_t^{(i)})$ and the counterfactual annotation $g_t^{(i),\tilde{a}} \sim Q_{t:T}^{\pi_b}(s_t^{(i)}, \tilde{a})$ for action $\tilde{a}$, we convert the annotation as

$$\hat{g}_t^{(i),\tilde{a}} = g_t^{(i),\tilde{a}} - \hat{\epsilon}_{G_t}(s_t, \tilde{a})$$

One may verify that $\hat{g}_t^{(i),\tilde{a}} \sim Q_{t:T}^{\pi_e}(s_t^{(i)}, \tilde{a})$ in expectation. Note that this approach for annotation conversion is only possible if the counterfactual action has support in the offline data (i.e., $(s_t^{(i)}, \tilde{a})$ is seen in the data); otherwise, we suggest using the annotation as collected.

## D.2 Imputing Missing Annotations

Consider the following scenario: for two instances of the same factual $(s, a)$ in the dataset, only one instance has a counterfactual annotation but not the other. This means that the weights for the factual and counterfactual are $(0.5, 0.5)$ and $(1, 0)$ (assuming a binary action space), and the weight distribution $W(\cdot|s, a)$ has a nonzero variance and covariance. Specifically, in the example above with two samples, $\sigma_W(a|s, a)^2 = \sigma_W(\tilde{a}|s, a)^2 = \text{Cov}(W(a|s, a), W(\tilde{a}|s, a)) = 0.0625$. These variances appear in the variance decomposition of C-IS as shown in Theorem 13 and may lead to an overall larger variance compared to IS.

To address this issue, we suggest a procedure to impute the missing annotations using other available annotations when possible. Given all annotations $\boldsymbol{g}_{\mathcal{D}} = \{g_t^{(i),\tilde{a}} : c_t^{(i),\tilde{a}} = 1\}$ associated with dataset $\mathcal{D} = \{\tau^{(i)}\}_{i=1}^N$, we first build an approximate annotation model $\hat{G}$ by solving a regression problem on $\{((s_t^{(i)}, \tilde{a}), g_t^{(i),\tilde{a}}) : c_t^{(i),\tilde{a}} = 1\}$ using all available annotations $\boldsymbol{g}_{\mathcal{D}}$. For discrete state and action spaces, this is essentially to averaging the annotations for each state-action pair. Then, $\hat{G}$ can be used to impute the missing annotations as $\hat{\boldsymbol{g}} = \{\hat{G}(s_t^{(i)}, \tilde{a}) : c_t^{(i),\tilde{a}} = 0, \tilde{a} \in \mathcal{A} \setminus \{a_t^{(i)}\}\}$ if there is "support" for annotations of the same state and action, i.e., $c_t^{(i),\tilde{a}} = 0$ (annotation is missing) and $\sum_{i'=1}^N \sum_{t'=1}^T \mathbb{1}[s_{t'}^{(i')} = s_t^{(i)}] c_{t'}^{(i'),\tilde{a}} > 0$ (annotation has support and thus can be imputed). This approximate annotation model $\hat{G}$ may be biased, but generally its bias can be outweighed by the benefit of variance reduction for having equal weights (we also observed this empirically in Appendices E.1 and E.2).

## D.3 Optimizing Weighting Schemes to Minimize Variance

Using a worked example, we illustrate that optimizing weights for variance reduction is a highly non-trivial problem. Suppose we have a bandit problem with one state $\{s\}$, two actions $\{\nearrow, \searrow\}$, and the reward function is $R(s, \nearrow) = \mathcal{N}(r_0, \sigma_0^2)$, $R(s, \searrow) = \mathcal{N}(r_1, \sigma_1^2)$. Furthermore, suppose all annotations are available, allowing us to use constant weights and eliminate the weight variances, and suppose the annotation function is identical to the reward function, i.e., $G = R$. We can simplify the variance decomposition of C-IS to be

$$\mathbb{V}[\hat{v}^{\text{C-IS}}] = \mathbb{V}_{s \sim d_1}[V^{\pi_e}(s)] + \mathbb{E}_{s \sim d_1}\left[\mathbb{V}_{a \sim \pi_b(s)}\left[\sum_{\tilde{a} \in \mathcal{A}} \rho^+(\tilde{a}|s) \bar{W}(\tilde{a}|s, a) \bar{R}(s, \tilde{a})\right]\right]$$
$$+ \mathbb{E}_{s \sim d_1}\mathbb{E}_{a \sim \pi_b(s)}\left[\sum_{\tilde{a} \in \mathcal{A}} \rho^+(\tilde{a}|s)^2 \bar{W}(\tilde{a}|s, a)^2 \sigma_R(s, \tilde{a})^2\right] \tag{4}$$

We denote policies in terms of the probabilities assigned to the two actions.

**Setting 1.** Consider $\pi_b(s) = [1, 0]$, $\pi_e(s) = [\alpha, 1 - \alpha]$. The offline dataset contains only $(s, a = \nearrow)$ with counterfactual annotations $(s, \tilde{a} = \searrow)$, and we assume they are each assigned a weight of $w$ and $(1 - w)$. The augmented behavior policy is then $\pi_{b^+}(s) = [w, 1 - w]$. The first and second terms of Eqn. (4) are zero because no variance is associated with sampling state-action pairs (only one possibility under $\pi_b$). The third term becomes (state $s$ is omitted from the expressions)

$$\frac{\pi_e(\nearrow)}{\pi_{b^+}(\nearrow)} \bar{W}(\nearrow \mid \nearrow)^2 \sigma_0^2 + \frac{\pi_e(\searrow)}{\pi_{b^+}(\searrow)} \bar{W}(\searrow \mid \nearrow)^2 \sigma_1^2$$

$$= \frac{\alpha}{w} \times w^2 \times \sigma_0^2 + \frac{1 - \alpha}{1 - w} \times (1 - w)^2 \times \sigma_1^2$$

$$= w\alpha\sigma_0^2 + (1 - w)(1 - \alpha)\sigma_1^2$$

$$= \left(\alpha\sigma_0^2 - (1 - \alpha)\sigma_1^2\right)w + (1 - \alpha)\sigma_1^2$$

This is a linear function of $w$, and where the variance achieves the minimum value depends on the slope $\left(\alpha\sigma_0^2 - (1 - \alpha)\sigma_1^2\right)$.

- If $\alpha\sigma_0^2 = (1 - \alpha)\sigma_1^2$, the slope is zero, and $w$ does not affect variance.
- If $\alpha\sigma_0^2 < (1 - \alpha)\sigma_1^2$, the slope is negative, and $w^* \to 1$ achieves minimum variance of $\alpha\sigma_0^2$.
- If $\alpha\sigma_0^2 > (1 - \alpha)\sigma_1^2$, the slope is positive, and $w^* \to 0$ achieves minimum variance of $(1 - \alpha)\sigma_1^2$.

**Setting 2.** Consider $\pi_b(s) = [\beta, 1 - \beta]$, $\pi_e(s) = [\alpha, 1 - \alpha]$. In the offline dataset, $(s, a = \nearrow)$ appears with probability $\beta$ and $(s, a = \searrow)$ appears with probability $1 - \beta$. When the factual data is $(s, a = \nearrow)$, the counterfactual annotation is for $(s, \tilde{a} = \searrow)$, and let the weights be $w_0$ and $(1 - w_0)$; when the factual data is $(s, a = \searrow)$, the counterfactual annotation is for $(s, \tilde{a} = \nearrow)$, and let the weights be $w_1$ and $(1 - w_1)$. The augmented behavior policy is then

$$\pi_{b^+}(s) = \begin{cases} \beta w_0 + (1 - \beta)(1 - w_1) \\ \beta(1 - w_0) + (1 - \beta)w_1 \end{cases}.$$

The first term of Eqn. (4) is zero because there is only one state. The third term is

$$\beta\left(\frac{\pi_e(\nearrow)}{\pi_{b^+}(\nearrow)} \bar{W}(\nearrow \mid \nearrow)^2 \sigma_0^2 + \frac{\pi_e(\searrow)}{\pi_{b^+}(\searrow)} \bar{W}(\searrow \mid \nearrow)^2 \sigma_1^2\right)$$

$$+ (1 - \beta)\left(\frac{\pi_e(\searrow)}{\pi_{b^+}(\searrow)} \bar{W}(\nearrow \mid \searrow)^2 \sigma_0^2 + \frac{\pi_e(\nearrow)}{\pi_{b^+}(\nearrow)} \bar{W}(\nearrow \mid \searrow)^2 \sigma_1^2\right)$$

$$= \beta\left(\frac{\alpha}{\beta w_0 + (1 - \beta)(1 - w_1)} \times w_0^2 \times \sigma_0^2 + \frac{1 - \alpha}{\beta(1 - w_0) + (1 - \beta)w_1} \times (1 - w_0)^2 \times \sigma_1^2\right)$$

$$+ (1 - \beta)\left(\frac{\alpha}{\beta w_0 + (1 - \beta)(1 - w_1)} \times (1 - w_1)^2 \times \sigma_0^2 + \frac{1 - \alpha}{\beta(1 - w_0) + (1 - \beta)w_1} \times w_1^2 \times \sigma_1^2\right)$$

$$= \frac{\beta w_0^2 + (1 - \beta)(1 - w_1)^2}{\beta w_0 + (1 - \beta)(1 - w_1)} \times \alpha\sigma_0^2 + \frac{\beta(1 - w_0)^2 + (1 - \beta)w_1^2}{\beta(1 - w_0) + (1 - \beta)w_1} \times (1 - \alpha)\sigma_1^2$$

Next, we will attempt to simplify the second variance term:

$$\mathbb{V}_{a \sim \pi_b(s)}\left[\sum_{\tilde{a} \in \mathcal{A}} \rho^+(\tilde{a} \mid s) \bar{W}(\tilde{a} \mid s, a) \bar{R}(s, \tilde{a})\right].$$

First note that

$$\mathbb{E}_{a \sim \pi_b(s)}\left[\sum_{\tilde{a} \in \mathcal{A}} \rho^+(\tilde{a} \mid s) \bar{W}(\tilde{a} \mid s, a) \bar{R}(s, \tilde{a})\right]$$

$$= \frac{\pi_e(\nearrow)}{\pi_{b^+}(\nearrow)} \bar{W}(\nearrow \mid \nearrow)r_0 + \frac{\pi_e(\searrow)}{\pi_{b^+}(\searrow)} \bar{W}(\searrow \mid \nearrow)r_1$$

$$= \frac{\alpha}{w} \times w \times r_0 + \frac{1 - \alpha}{1 - w} \times (1 - w) \times r_1$$

$$= \alpha r_0 + (1 - \alpha)r_1$$

Then, the second variance term becomes

$$\beta\left(\frac{\alpha}{\beta w_0 + (1-\beta)(1-w_1)} \times w_0 \times r_0 + \frac{1-\alpha}{\beta(1-w_0)+(1-\beta)w_1} \times (1-w_0) \times r_1 - \big(\alpha r_0 + (1-\alpha)r_1\big)\right)^2$$

$$+ (1-\beta)\left(\frac{\alpha}{\beta w_0 + (1-\beta)(1-w_1)} \times (1-w_1) \times r_0 + \frac{1-\alpha}{\beta(1-w_0)+(1-\beta)w_1} \times w_1 \times r_1 - \big(\alpha r_0 + (1-\alpha)r_1\big)\right)^2$$

$$= \beta\left[\alpha r_0\Big(\frac{w_0}{\beta w_0 + (1-\beta)(1-w_1)} - 1\Big) + (1-\alpha)r_1\Big(\frac{(1-w_0)}{\beta(1-w_0)+(1-\beta)w_1} - 1\Big)\right]^2$$

$$+ (1-\beta)\left[\alpha r_0\Big(\frac{1-w_1}{\beta w_0 + (1-\beta)(1-w_1)} - 1\Big) + (1-\alpha)r_1\Big(\frac{w_1}{\beta(1-w_0)+(1-\beta)w_1} - 1\Big)\right]^2$$

$$= \beta\left[\alpha^2 r_0^2\Big(\frac{w_0}{\beta w_0 + (1-\beta)(1-w_1)} - 1\Big)^2 + (1-\alpha)^2 r_1^2\Big(\frac{(1-w_0)}{\beta(1-w_0)+(1-\beta)w_1} - 1\Big)^2\right.$$

$$\left. + 2\alpha(1-\alpha)r_0 r_1\Big(\frac{w_0}{\beta w_0 + (1-\beta)(1-w_1)} - 1\Big)\Big(\frac{(1-w_0)}{\beta(1-w_0)+(1-\beta)w_1} - 1\Big)\right]$$

$$+ (1-\beta)\left[\alpha^2 r_0^2\Big(\frac{1-w_1}{\beta w_0 + (1-\beta)(1-w_1)} - 1\Big)^2 + (1-\alpha)^2 r_1^2\Big(\frac{w_1}{\beta(1-w_0)+(1-\beta)w_1} - 1\Big)^2\right.$$

$$\left. + 2\alpha(1-\alpha)r_0 r_1\Big(\frac{1-w_1}{\beta w_0 + (1-\beta)(1-w_1)} - 1\Big)\Big(\frac{w_1}{\beta(1-w_0)+(1-\beta)w_1} - 1\Big)\right]$$

$$= \alpha^2 r_0^2\left[\beta\Big(\frac{w_0}{\beta w_0 + (1-\beta)(1-w_1)} - 1\Big)^2 + (1-\beta)\Big(\frac{1-w_1}{\beta w_0 + (1-\beta)(1-w_1)} - 1\Big)^2\right]$$

$$+ (1-\alpha)^2 r_1^2\left[\beta\Big(\frac{1-w_0}{\beta(1-w_0)+(1-\beta)w_1} - 1\Big)^2 + (1-\beta)\Big(\frac{w_1}{\beta(1-w_0)+(1-\beta)w_1} - 1\Big)^2\right]$$

$$+ 2\alpha(1-\alpha)r_0 r_1\left[\Big(\frac{w_0}{\beta w_0 + (1-\beta)(1-w_1)} - 1\Big)\Big(\frac{1-w_0}{\beta(1-w_0)+(1-\beta)w_1} - 1\Big)\right.$$

$$\left. + \Big(\frac{1-w_1}{\beta w_0 + (1-\beta)(1-w_1)} - 1\Big)\Big(\frac{w_1}{\beta(1-w_0)+(1-\beta)w_1} - 1\Big)\right]$$

As shown above, the variance expression is a rather complicated function of $(w_0, w_1)$. To solve for its minimum, one may take the derivatives with respect to each of $w_0$ and $w_1$ and solve for the zeros. The final solution will depend on the problem parameters, including $r_0, r_1, \sigma_0^2, \sigma_1^2, \alpha, \beta$. Here, we do not solve for the final solution, and note that this approach may not be applicable to real-world problems due to its explicitly dependence on problem parameters that may be unknown. We encourage future work to explore different methods to find the variance-minimizing weighting schemes.

# E    Extended Experiments

## E.1    Synthetic Domains - Bandits

### E.1.1    Two-State Bandits

We expand on the experiments shown in Section 5.1, where we consider a class of bandit problems with two states $\{s_1, s_2\}$ (drawn with equal probability), two actions $\mathcal{A} = \{\nearrow, \searrow\}$ (recall Figure 3), and corresponding reward distributions $R(s_i, a) \sim \mathcal{N}(\bar{R}_{(s_i, a)}, \sigma^2)$. Without loss of generality, we assume $\nearrow$ is always taken from $s_2$ by both $\pi_b$ and $\pi_e$. For $s_1$, we consider deterministic policies (in which one action is always taken) as well as stochastic policies (which take the two actions with probabilities that sum to 1); see column/row header in Table 3. Given $(\pi_b, \pi_e)$, we draw $1,000$ samples following $\pi_b$ and then evaluate $\pi_e$ using various estimators, including standard IS, the naive baseline of adding counterfactual annotations as new samples (Section 3.1), and C*-IS. We assume that counterfactual annotations are only available for $s_1$, and all annotations are drawn from the true reward function. We measure the bias, standard deviation (square root of variance), and root mean-squared error (RMSE) of the estimators with respect to $v(\pi_e)$. In Table 3 we consider three settings of the rewards: (i) $\bar{R}_{(s_1, \nearrow)} = 1, \bar{R}_{(s_1, \searrow)} = 2$, i.e., both actions lead to a positive reward. (ii) $\bar{R}_{(s_1, \nearrow)} = -1, \bar{R}_{(s_1, \searrow)} = 1$, i.e., one action leads to a positive reward and the other leads to a negative reward, (iii) $\bar{R}_{(s_1, \nearrow)} = -1, \bar{R}_{(s_1, \searrow)} = -2$, i.e., both actions lead to a negative reward. In all cases, the reward for state $s_2$ is set to $0$ for both actions.

**Naive baseline fails due to bias.** The naive baseline often has a nonzero bias and worse RMSE than IS regardless of whether the offline data has support. This is consistent with our analyses and example provided in Appendix B.

**Bias reduction in support-deficient settings.** In the first two rows of each sub-table in Table 3, $\pi_b$ is deterministic and the untaken action has poor support in the offline data. IS is often biased for these cases; note that in cases where $\pi_e$ assigns a small probability to the unsupported action (e.g., row 2, column 4, $\pi_e$ takes $\nearrow$ with probability 0.1), the bias is small and shows up as $0$ after rounding. In particular, when rewards as all positive (the first table of Table 3), the bias is negative, and when rewards are all negative (the third table of Table 3), the bias is positive. In the second table of Table 3, the direction of bias depends on the reward of the unsupported action. In the last three rows, IS is unbiased when the offline data has full support. In contrast, C*-IS is unbiased in all cases (but for rounding errors), and can reduce bias compared to IS in support-deficient settings by making use of counterfactual annotations. RMSE of C*-IS is often (but not always) reduced compared to IS even though variance can sometimes increase. This is consistent with our analyses in Appendix C.2.

**Variance reduction in well-supported settings.** In the last three rows of each sub-table in Table 3, the offline data has full support because $\pi_b$ is stochastic, and IS is unbiased in these cases. C*-IS is also unbiased in these cases and achieves lower variance, leading to lower RMSE than IS. This is consistent with our analyses in Appendix C.3.

### E.1.2    One-State Bandits

To simplify further experiments, we modify the two-state bandits above so that $s_2$ is drawn with probability $0$ — equivalently, we have a set of one-state bandits (with only $s_1$) with two actions $\mathcal{A} = \{0, 1\}$ where $0$ corresponds to $\nearrow$ and $1$ corresponds to $\searrow$, and corresponding reward distributions $R_0 \sim \mathcal{N}(\bar{R}_0, \sigma_0^2)$ and $R_1 \sim \mathcal{N}(\bar{R}_1, \sigma_1^2)$. For $\pi_b$ and $\pi_e$, we consider the same set of policies as before, and the same evaluation setup. We omit the naive approach in this setting because it is equivalent to our proposed approach when there is only one state (the state distribution is not affected by adding counterfactual annotations directly).

Results for this setting are summarized in Table 4 and **show similar trends as above**. Next, we use the one-state bandit to study the effect of weights $w$ on OPE performance.

Table 3: Summary of performance on the **two-state** bandit problem for various $\pi_b$ (rows) and $\pi_e$ (columns), where each policy is denoted by its probabilities assigned to the two actions from $s_1$. Each cell of the table corresponds to a $(\pi_b, \pi_e)$ combination, for which we report (bias, std, RMSE) for three estimators: IS in the top row, naive in the middle row, and C*-IS in the bottom row.

$$\bar{R}_{(s_1,\nearrow)} = 1, \bar{R}_{(s_1,\searrow)} = 2$$

| $\pi_b$ \ $\pi_e$ | [ 1 , 0 ] | | | [ 0 , 1 ] | | | [0.5, 0.5] | | | [0.1, 0.9] | | | [0.8, 0.2] | | |
|---|---|---|---|---|---|---|---|---|---|---|---|---|---|---|---|
| [1.0, 0.0] | 0 | 0.7 | 0.7 | -1 | 0.4 | 1 | -0.5 | 0.5 | 0.7 | -0.9 | 0.4 | 1 | -0.2 | 0.6 | 0.6 |
| | 0.2 | 1.1 | 1.2 | 0.3 | 2 | 2 | 0.3 | 0.9 | 1 | 0.3 | 1.8 | 1.8 | 0.2 | 0.9 | 0.9 |
| | 0 | 0.7 | 0.7 | 0 | 1.1 | 1.1 | 0 | 0.9 | 0.9 | 0 | 1.1 | 1.1 | 0 | 0.7 | 0.7 |
| [0.0, 1.0] | -0.5 | 0.4 | 0.6 | 0 | 1.1 | 1.1 | -0.2 | 0.6 | 0.7 | 0 | 1 | 1 | -0.4 | 0.4 | 0.6 |
| | 0.2 | 1.1 | 1.1 | 0.3 | 2 | 2 | 0.3 | 0.9 | 1 | 0.3 | 1.8 | 1.8 | 0.2 | 0.8 | 0.9 |
| | 0 | 0.7 | 0.7 | 0 | 1.1 | 1.1 | 0 | 0.9 | 0.9 | 0 | 1.1 | 1.1 | 0 | 0.7 | 0.7 |
| [0.5, 0.5] | 0 | 1 | 1 | 0 | 1.8 | 1.8 | 0 | 1 | 1 | 0 | 1.6 | 1.6 | 0 | 0.8 | 0.8 |
| | 0.2 | 1.1 | 1.1 | 0.3 | 2 | 2 | 0.3 | 0.9 | 1 | 0.3 | 1.8 | 1.8 | 0.2 | 0.8 | 0.9 |
| | 0 | 0.7 | 0.7 | 0 | 1.1 | 1.1 | 0 | 0.9 | 0.9 | 0 | 1.1 | 1.1 | 0 | 0.7 | 0.7 |
| [0.1, 0.9] | 0.1 | 2.6 | 2.6 | 0 | 1.2 | 1.2 | 0 | 1.3 | 1.3 | 0 | 1.1 | 1.1 | 0.1 | 2 | 2 |
| | 0.2 | 1.1 | 1.1 | 0.3 | 2 | 2 | 0.3 | 0.9 | 1 | 0.3 | 1.8 | 1.8 | 0.2 | 0.8 | 0.9 |
| | 0 | 0.7 | 0.7 | 0 | 1.1 | 1.1 | 0 | 0.9 | 0.9 | 0 | 1.1 | 1.1 | 0 | 0.7 | 0.7 |
| [0.8, 0.2] | 0 | 0.8 | 0.8 | 0.1 | 3.2 | 3.2 | 0 | 1.6 | 1.6 | 0 | 2.9 | 2.9 | 0 | 0.8 | 0.8 |
| | 0.2 | 1.1 | 1.2 | 0.3 | 2 | 2 | 0.3 | 0.9 | 1 | 0.3 | 1.7 | 1.8 | 0.2 | 0.8 | 0.9 |
| | 0 | 0.7 | 0.7 | 0 | 1.1 | 1.1 | 0 | 0.9 | 0.9 | 0 | 1.1 | 1.1 | 0 | 0.7 | 0.7 |

$$\bar{R}_{(s_1,\nearrow)} = -1, \bar{R}_{(s_1,\searrow)} = 1$$

| $\pi_b$ \ $\pi_e$ | [ 1 , 0 ] | | | [ 0 , 1 ] | | | [0.5, 0.5] | | | [0.1, 0.9] | | | [0.8, 0.2] | | |
|---|---|---|---|---|---|---|---|---|---|---|---|---|---|---|---|
| [1.0, 0.0] | 0 | 0.7 | 0.7 | -0.5 | 0.4 | 0.6 | -0.2 | 0.5 | 0.5 | -0.4 | 0.4 | 0.6 | -0.1 | 0.6 | 0.6 |
| | -0.1 | 1.1 | 1.1 | 0.2 | 1.1 | 1.1 | 0 | 0.9 | 0.9 | 0.2 | 1.1 | 1.1 | -0.1 | 1 | 1 |
| | 0 | 0.7 | 0.7 | 0 | 0.7 | 0.7 | 0 | 0.4 | 0.4 | 0 | 0.6 | 0.6 | 0 | 0.6 | 0.6 |
| [0.0, 1.0] | 0.5 | 0.4 | 0.6 | 0 | 0.7 | 0.7 | 0.3 | 0.5 | 0.5 | 0.1 | 0.6 | 0.7 | 0.4 | 0.4 | 0.6 |
| | -0.1 | 1.1 | 1.1 | 0.2 | 1.1 | 1.2 | 0 | 0.9 | 0.9 | 0.2 | 1.1 | 1.1 | -0.1 | 1 | 1 |
| | 0 | 0.7 | 0.7 | 0 | 0.7 | 0.7 | 0 | 0.4 | 0.4 | 0 | 0.6 | 0.6 | 0 | 0.6 | 0.6 |
| [0.5, 0.5] | 0 | 1.1 | 1.1 | 0 | 1.1 | 1.1 | 0 | 0.9 | 0.9 | 0 | 1 | 1 | 0 | 0.9 | 0.9 |
| | -0.1 | 1.1 | 1.1 | 0.2 | 1.2 | 1.2 | 0 | 1 | 1 | 0.2 | 1.1 | 1.1 | -0.1 | 1 | 1 |
| | 0 | 0.7 | 0.7 | 0 | 0.7 | 0.7 | 0 | 0.4 | 0.4 | 0 | 0.6 | 0.6 | 0 | 0.6 | 0.6 |
| [0.1, 0.9] | 0 | 2.4 | 2.4 | 0 | 0.7 | 0.7 | 0 | 1.4 | 1.4 | 0 | 0.8 | 0.8 | 0 | 2 | 2 |
| | -0.1 | 1.1 | 1.1 | 0.2 | 1.1 | 1.2 | 0 | 0.9 | 0.9 | 0.2 | 1.1 | 1.1 | -0.1 | 1 | 1 |
| | 0 | 0.7 | 0.7 | 0 | 0.7 | 0.7 | 0 | 0.4 | 0.4 | 0 | 0.6 | 0.6 | 0 | 0.6 | 0.6 |
| [0.8, 0.2] | 0 | 0.8 | 0.8 | 0.1 | 1.8 | 1.8 | 0 | 1.1 | 1.1 | 0.1 | 1.7 | 1.7 | 0 | 0.8 | 0.8 |
| | -0.1 | 1.1 | 1.1 | 0.2 | 1.1 | 1.2 | 0 | 0.9 | 0.9 | 0.1 | 1.1 | 1.1 | -0.1 | 1 | 1 |
| | 0.1 | 0.7 | 0.7 | 0 | 0.7 | 0.7 | 0 | 0.4 | 0.4 | 0 | 0.6 | 0.6 | 0 | 0.5 | 0.5 |

$$\bar{R}_{(s_1,\nearrow)} = -1, \bar{R}_{(s_1,\searrow)} = -2$$

| $\pi_b$ \ $\pi_e$ | [ 1 , 0 ] | | | [ 0 , 1 ] | | | [0.5, 0.5] | | | [0.1, 0.9] | | | [0.8, 0.2] | | |
|---|---|---|---|---|---|---|---|---|---|---|---|---|---|---|---|
| [1.0, 0.0] | 0 | 0.7 | 0.7 | 1 | 0.4 | 1.1 | 0.5 | 0.5 | 0.7 | 0.9 | 0.4 | 1 | 0.2 | 0.6 | 0.7 |
| | -0.1 | 1.1 | 1.1 | -0.3 | 2 | 2 | -0.2 | 1 | 1 | -0.3 | 1.7 | 1.8 | -0.2 | 0.9 | 0.9 |
| | 0 | 0.7 | 0.7 | 0.1 | 1.1 | 1.1 | 0.1 | 0.9 | 0.9 | 0.1 | 1.1 | 1.1 | 0 | 0.8 | 0.8 |
| [0.0, 1.0] | 0.5 | 0.4 | 0.6 | 0.1 | 1.1 | 1.1 | 0.3 | 0.7 | 0.7 | 0.1 | 1 | 1 | 0.4 | 0.4 | 0.6 |
| | -0.1 | 1.1 | 1.1 | -0.3 | 2 | 2 | -0.2 | 1 | 1 | -0.3 | 1.7 | 1.8 | -0.2 | 0.9 | 0.9 |
| | 0 | 0.7 | 0.7 | 0.1 | 1.1 | 1.1 | 0.1 | 0.9 | 0.9 | 0.1 | 1.1 | 1.1 | 0 | 0.8 | 0.8 |
| [0.5, 0.5] | 0 | 1.1 | 1.1 | 0.1 | 1.8 | 1.8 | 0.1 | 1 | 1 | 0.1 | 1.6 | 1.6 | 0 | 0.9 | 0.9 |
| | -0.1 | 1.1 | 1.1 | -0.3 | 2 | 2 | -0.2 | 1 | 1 | -0.3 | 1.7 | 1.8 | -0.2 | 0.8 | 0.9 |
| | 0 | 0.7 | 0.7 | 0.1 | 1.1 | 1.1 | 0.1 | 0.9 | 0.9 | 0.1 | 1.1 | 1.1 | 0 | 0.8 | 0.8 |
| [0.1, 0.9] | 0 | 2.4 | 2.4 | 0.1 | 1.2 | 1.2 | 0 | 1.3 | 1.3 | 0.1 | 1.1 | 1.1 | 0 | 1.9 | 1.9 |
| | -0.1 | 1.1 | 1.1 | -0.3 | 2 | 2 | -0.2 | 1 | 1 | -0.3 | 1.7 | 1.8 | -0.2 | 0.9 | 0.9 |
| | 0 | 0.7 | 0.7 | 0.1 | 1.1 | 1.1 | 0.1 | 0.9 | 0.9 | 0.1 | 1.1 | 1.1 | 0 | 0.8 | 0.8 |
| [0.8, 0.2] | 0 | 0.8 | 0.8 | 0.1 | 3.1 | 3.1 | 0 | 1.5 | 1.5 | 0.1 | 2.8 | 2.8 | 0 | 0.8 | 0.8 |
| | -0.1 | 1.1 | 1.1 | -0.3 | 2 | 2 | -0.2 | 1 | 1 | -0.3 | 1.8 | 1.8 | -0.2 | 0.8 | 0.9 |
| | 0.1 | 0.7 | 0.7 | 0 | 1.1 | 1.1 | 0.1 | 0.9 | 0.9 | 0 | 1.1 | 1.1 | 0.1 | 0.8 | 0.8 |

Table 4: Summary of performance on the **one-state** bandit problem for various $\pi_b$ (rows) and $\pi_e$ (columns), where each policy is denoted by its probabilities assigned to the two actions from $s_1$. Each cell of the table corresponds to a $(\pi_b, \pi_e)$ combination, for which we report (bias, std, RMSE) for three estimators: IS in the top row, naive in the middle row, and C*-IS in the bottom row.

$$\bar{R}_0 = 1, \bar{R}_1 = 2$$

| $\pi_e$ \ $\pi_b$ | [ 1 , 0 ] | [ 0 , 1 ] | [0.5, 0.5] | [0.1, 0.9] | [0.8, 0.2] |
|---|---|---|---|---|---|
| [1.0, 0.0] | 0.03 0.5 0.5
0.03 0.5 0.5 | -2 0 2
0.02 0.47 0.47 | -0.98 0.25 1.01
0.03 0.34 0.35 | -1.8 0.05 1.8
0.02 0.43 0.43 | -0.37 0.4 0.55
0.03 0.41 0.41 |
| [0.0, 1.0] | -1 0 1
0.02 0.47 0.47 | 0.03 0.5 0.5
0.03 0.5 0.5 | -0.48 0.25 0.54
0.03 0.34 0.35 | -0.07 0.45 0.45
0.03 0.45 0.45 | -0.79 0.1 0.8
0.02 0.39 0.39 |
| [0.5, 0.5] | 0.01 1.23 1.23
0.03 0.47 0.47 | 0.09 2.17 2.17
0.02 0.5 0.5 | 0.05 0.71 0.71
0.03 0.34 0.35 | 0.08 1.86 1.86
0.02 0.46 0.46 | 0.02 0.69 0.69
0.03 0.39 0.39 |
| [0.1, 0.9] | 0.08 3.46 3.46
0.01 0.47 0.47 | 0.02 0.88 0.88
0.04 0.5 0.5 | 0.05 1.45 1.45
0.03 0.34 0.35 | 0.03 0.59 0.59
0.03 0.45 0.45 | 0.07 2.64 2.64
0.02 0.39 0.39 |
| [0.8, 0.2] | 0.03 0.75 0.75
0.04 0.48 0.48 | 0.05 4.28 4.28
0.01 0.49 0.49 | 0.04 1.92 1.92
0.03 0.34 0.35 | 0.05 3.8 3.81
0.02 0.44 0.44 | 0.03 0.65 0.65
0.03 0.4 0.4 |

$$\bar{R}_0 = -1, \bar{R}_1 = 1$$

| $\pi_e$ \ $\pi_b$ | [ 1 , 0 ] | [ 0 , 1 ] | [0.5, 0.5] | [0.1, 0.9] | [0.8, 0.2] |
|---|---|---|---|---|---|
| [1.0, 0.0] | 0.03 0.5 0.5
0.03 0.5 0.5 | -1 0 1
0.02 0.47 0.47 | -0.48 0.25 0.54
0.03 0.34 0.35 | -0.9 0.05 0.9
0.02 0.43 0.43 | -0.17 0.4 0.43
0.03 0.41 0.41 |
| [0.0, 1.0] | 1 0 1
0.02 0.47 0.47 | 0.03 0.5 0.5
0.03 0.5 0.5 | 0.52 0.25 0.57
0.03 0.34 0.35 | 0.13 0.45 0.47
0.03 0.45 0.45 | 0.81 0.1 0.81
0.02 0.39 0.39 |
| [0.5, 0.5] | 0.06 1.17 1.17
0.03 0.47 0.47 | 0.06 1.28 1.28
0.02 0.5 0.5 | 0.06 1.12 1.12
0.03 0.34 0.35 | 0.06 1.23 1.23
0.02 0.46 0.46 | 0.06 1.13 1.13
0.03 0.39 0.39 |
| [0.1, 0.9] | -0.04 3.37 3.37
0.01 0.47 0.47 | 0.03 0.64 0.64
0.04 0.5 0.5 | -0.01 1.86 1.86
0.03 0.34 0.35 | 0.02 0.8 0.8
0.03 0.45 0.45 | -0.03 2.76 2.76
0.02 0.39 0.39 |
| [0.8, 0.2] | 0.02 0.74 0.74
0.04 0.48 0.48 | 0.06 2.42 2.42
0.01 0.49 0.49 | 0.04 1.46 1.46
0.03 0.34 0.35 | 0.06 2.22 2.23
0.02 0.44 0.44 | 0.03 0.95 0.96
0.03 0.4 0.4 |

$$\bar{R}_0 = -1, \bar{R}_1 = -2$$

| $\pi_e$ \ $\pi_b$ | [ 1 , 0 ] | [ 0 , 1 ] | [0.5, 0.5] | [0.1, 0.9] | [0.8, 0.2] |
|---|---|---|---|---|---|
| [1.0, 0.0] | 0.03 0.5 0.5
0.03 0.5 0.5 | 2 0 2
0.02 0.47 0.47 | 1.02 0.25 1.05
0.03 0.34 0.35 | 1.8 0.05 1.8
0.02 0.43 0.43 | 0.43 0.4 0.58
0.03 0.41 0.41 |
| [0.0, 1.0] | 1 0 1
0.02 0.47 0.47 | 0.03 0.5 0.5
0.03 0.5 0.5 | 0.52 0.25 0.57
0.03 0.34 0.35 | 0.13 0.45 0.47
0.03 0.45 0.45 | 0.81 0.1 0.81
0.02 0.39 0.39 |
| [0.5, 0.5] | 0.06 1.17 1.17
0.03 0.47 0.47 | -0.01 2.1 2.1
0.02 0.5 0.5 | 0.02 0.7 0.71
0.03 0.34 0.35 | 0 1.8 1.8
0.02 0.46 0.46 | 0.04 0.67 0.67
0.03 0.39 0.39 |
| [0.1, 0.9] | -0.04 3.37 3.37
0.01 0.47 0.47 | 0.05 0.86 0.86
0.04 0.5 0.5 | 0 1.41 1.41
0.03 0.34 0.35 | 0.04 0.58 0.58
0.03 0.45 0.45 | -0.02 2.57 2.57
0.02 0.39 0.39 |
| [0.8, 0.2] | 0.02 0.74 0.74
0.04 0.48 0.48 | 0.09 4.02 4.02
0.01 0.49 0.49 | 0.06 1.8 1.8
0.03 0.34 0.35 | 0.08 3.57 3.57
0.02 0.44 0.44 | 0.04 0.63 0.63
0.03 0.4 0.4 |

**Using equal weights (in C\*-IS) is a reasonable heuristic though not always variance-minimizing.** In the results above, C\*-IS assumed the weights are split equally for factual and counterfactual data. Next, we explore the effect of different weighting schemes on the performance of C-IS, applied to the same class of bandit problems described above. Note that, consistent with Theorem 1, C-IS remains unbiased regardless of weights, except potentially at extreme values of weights (0 or 1) that ignore either the factual data or counterfactual annotations. Across different settings (Figure 7a-d), we found that the ideal weighting scheme is problem-specific, and certain weights may lead to higher variance compared to standard IS (e.g., lower left region of Figure 7b). Encouragingly, these results also demonstrate that C\*-IS, though not always variance-minimizing, consistently achieves lower variance than standard IS (when the data has full support and the estimator is unbiased). Additionally, variance in the weight distributions directly contributes to variance in the resulting estimator (Figure 7e), corroborating our analysis in Theorem 13. Overall, our results suggest that C\*-IS, which uses constant weights split equally among actions, is a promising heuristic for using C-IS in practice.

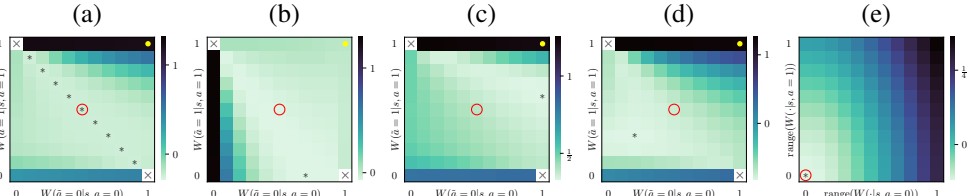

Figure 7: Heatmaps of log std for C-IS using different weights applied to various problems (with different bandit parameters, $\pi_b$, $\pi_e$, and annotation quality; exact parameter settings are specified below). $W(\tilde{a}|s,a)$ is the weight $w^{\tilde{a}}$ assigned to action $\tilde{a}$ given the factual sample $(s,a)$. Asterisks $*$ indicate the weights under which the variance is minimized for each problem. Gray crosses $\times$ indicate where the estimator is biased. Red circles $\bigcirc$ correspond to C\*-IS. Yellow dots in the upper right corners correspond to standard IS (where applicable). (a-b) Constant weights where we sweep the factual weight within the range $[0,1]$ for both actions. (c-d) Similar to (a), where the annotations have a larger (c) or smaller (d) variance than the reward function. (e) Weights are drawn from uniform distributions centered at $0.5$. We sweep the range of the uniform distributions within $[0,1]$.

Parameter settings: (a) Base setting, $R = [1,2], \sigma_R = [1,1], \pi_b = [0.1,0.9], \pi_e = [0.8,0.2], \sigma_G = \sigma_R$. (b) Annotations are not useful, $R = [1,2], \sigma_R = [1,1], \pi_b = [0.9,0.1], \pi_e = [0.95,0.05], \sigma_G = \sigma_R$. (c) Annotations has larger variance than rewards, $R = [1,2], \sigma_R = [1,1], \pi_b = [0.1,0.9], \pi_e = [0.8,0.2], \sigma_G = 2\sigma_R$. (d) Annotations has smaller variance than rewards, $R = [1,2], \sigma_R = [1,1], \pi_b = [0.1,0.9], \pi_e = [0.8,0.2], \sigma_G = 0.5\sigma_R$. (e) $R = [1,2], \sigma_R = [1,1], \pi_b = [0.1,0.9], \pi_e = [0.8,0.2], \sigma_G = \sigma_R$.

**Imputing missing annotations can reduce variance.** Finally, we explore the impact of missing annotations on the performance of C-IS. As shown in Figure 8a, obtaining more counterfactual annotations generally helps to reduce the variance. However, as noted in our variance analyses, if annotations for the same factual $(s,a)$ are sometimes missing, we cannot directly apply C\*-IS with equal weights; this may lead to increased variance due to variance in the weights. Here, we use a simple strategy to impute the missing annotations with the average of other annotations (for the same counterfactual $\tilde{a}$ of the same factual $(s,a)$). As shown in Figure 8b, this reduces the variance further especially when not all annotations are available.

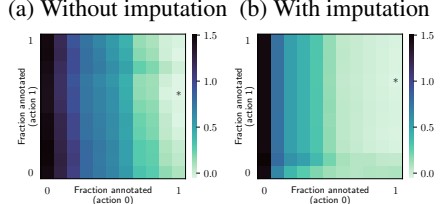

(a) Without imputation  (b) With imputation

Figure 8: Heatmaps of log std for C-IS where some counterfactual annotations may be missing. We vary the fraction of acquired annotations for the two actions independently within $[0,1]$. (a) Equal weights (as in C\*-IS) are used when annotations are available; otherwise, the factual weight is set to $1$ when annotations are missing. (b) The missing annotations are imputed using other similar annotations. Imputing missing annotations allows for C\*-IS to be applied directly and it achieves lower variance. Parameter settings are the same as Figure 7a.

### E.2 Healthcare Domain - Sepsis Simulator

#### E.2.1 Experimental Setup

**Simulator Description.** The patient state is characterized by five variables: a binary indicator for diabetes status, and four ordinal-valued vital signs (heart rate, blood pressure, oxygen concentration, glucose). Following prior work [22], we used a discrete state space with $|\mathcal{S}| = 1440$. The action space corresponds to the administration of vasopressors and takes on a binary value (on/off), which may increase or decrease the values of certain vital signs (with pre-specified probabilities) at the next time step. The action space in the original simulator formulation involves combinations of 3 treatments: antibiotics, vasopressors, and mechanical ventilation; we focus on only vasopressors for the purpose of illustration. The episode ends when the patient is discharged or died; discharge only occurs when all vitals are normal and all treatments are turned off, whereas death occurs if three or more vitals are abnormal. Rewards are sparse and only assigned at the end of each episode, with $+1$ for survival and $-1$ for death. Episodes that reach the maximum length of 20 are truncated with zero terminal reward.

**Evaluation Setup.** Following prior work [22], we collected 50 offline datasets from the sepsis simulator (using different random seeds) each with 1000 episodes by following an $\epsilon$-greedy behavior policy with respect to the optimal policy where $\epsilon = 0.1$. For the evaluation policies, we created a set of deterministic policies by perturbing the optimal policy such that each policy takes the non-optimal actions in a randomly selected subset of states. We varied the number of "action-flipped" states in $\{50, 100, 200, 300, 400\}$ and generated 5 different evaluation policies for each; we also included the optimal policy in the candidate set, resulting in a total of 26 candidate evaluation policies. These represent policies that may be derived from offline data by typical RL approaches that aim to learn deterministic policies, and are of diverse quality where approximately half are superior to the behavior policy while the other half are inferior (Figure 9). As the baseline estimator that only makes use of offline data, we applied standard PDIS which does not rely on counterfactual annotations. To apply C*-PDIS, we assume all counterfactual annotations are collected in our main experiments in which they may be drawn from different annotation functions; we explore the impact of missing annotations in subsequent sensitivity analyses. For C*-PDIS, we considered three different annotation functions: (i) $G = Q^{\pi_e}$, the Q-function of the evaluation policy $\pi_e$; (ii) $G = Q^{\pi_b}$, the Q-function of the behavior policy, and (iii) $G = Q^{\pi_b} \mapsto Q^{\pi_e}$, where we apply the bias correction procedure discussed in Appendix D.1. We also compare to two naive baselines (given perfect annotations): "naive unweighted" simply adds counterfactual annotations as new trajectories and has the same issue discussed in Section 3.1, whereas "naive weighted" reweights the annotations at the trajectory level instead of per-decision. More formally, assuming a binary action space $\mathcal{A} = \{0, 1\}$ (without loss of generality), given a trajectory of length $T$ with counterfactual annotations at each step, $\tau = [s_t, a_t, r_t]_{t=1}^T$, $\boldsymbol{g} = \{g_t^{1-a_t}\}_{t=1}^T$ where $1 - a_t$ is the counterfactual action for $a_t$, the naive weighted estimator is defined as

$$(1 - \textstyle\sum_{t=1}^T w_t)\rho_{1:T}(\textstyle\sum_{t=1}^T r_t) + \textstyle\sum_{t=1}^T \left(w_t \rho_{1:t-1} \rho_t^{1-a_t}(\textstyle\sum_{t'=1}^{t-1} r_{t'} + g_t^{1-a_t})\right)$$

Intuitively, this first converts each annotation into a sub-trajectory that terminates at the step of annotation with the counterfactual action $[s_1, a_1, r_1, \cdots, s_t, 1 - a_t, g_t]$, and then performs IS on each sub-trajectory (including the original trajectory), and finally computes a weighted sum of these $T + 1$ estimates (1 factual estimate, $T$ counterfactual estimates) using weights $(1 - \sum_{t=1}^T w_t)$, $w_1, \cdots, w_T$. The reason why "naive weighted" does not work is more subtle: while reweighting the (partial) trajectories constructed from the counterfactual annotations correctly maintains the initial state distribution, it does not correctly maintain the intermediate state distributions.

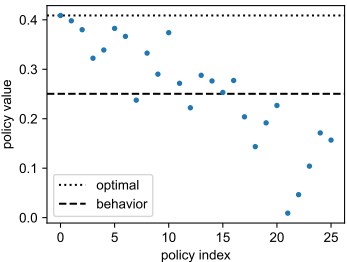 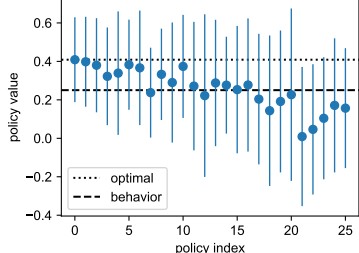

Figure 9: The true value $v(\pi_e)$ of each of the 26 evaluation policies. The two dashed line are, respectively, the value of the optimal policy and the value of the behavior policy $v(\pi_b)$ (which is $\epsilon$-greedy with respect to the optimal policy where $\epsilon = 0.1$). On the right, we additionally plot error bars representing $\pm$ standard deviation of the values at initial states for each of the policies. The average std of initial state values over all 26 policies is $0.312$.

### E.2.2 Results

**C\*-PDIS outperforms all baselines in all metrics for the ideal setting.** As shown in Table 5, when all counterfactuals are available and annotated with the evaluation policy's Q-function ($G = Q^{\pi_e}$), C\*-PDIS outperforms baseline PDIS (without annotations) in all metrics, suggesting that it provides more accurate OPE estimates. In contrast, the two naive approaches fail to provide accurate estimates and often underperform standard PDIS.

**C\*-PDIS is robust to biased annotations.** Under the more realistic scenario where $G = Q^{\pi_b}$, i.e., annotations summarize the future returns under $\pi_b$ rather than $\pi_e$, we observe a degradation in all metrics compared to the ideal case, though C\*-PDIS is still superior to PDIS (Table 5). Applying the bias correction procedure $G = Q^{\pi_b} \mapsto \hat{Q}^{\pi_e}$ (see Appendix D.1) helps recover performance to be closer to the ideal case, and is especially helpful for $\pi_e$ that are far away from $\pi_b$ (Figure 10).

Table 5: Comparison of baseline and proposed estimators in terms of OPE performance (RMSE, ESS), ranking performance (Spearman's rank correlation) and binary classification performance (accuracy, FPR, FNR) on the sepsis simulator, reported as mean $\pm$ std from 50 repeated runs. **Bolded** results are the best for each metric, whereas highlighted results outperform all baselines. The upper table shows the overall results; the lower table shows the breakdown by ordinary IS and weighted IS (OIS vs WIS) applied to each approach.

| | Estimator | ↓ **RMSE** | ↑ **ESS** | ↑ **Spearman** | ↑ **%Accuracy** | ↓ **%FPR** | ↓ **%FNR** | |
|---|---|---|---|---|---|---|---|---|
| **Baseline** | PDIS (w/o annot.) | $0.113_{\pm 0.038}$ | $76.8_{\pm 44.0}$ | $0.596_{\pm 0.110}$ | $76.5_{\pm 3.5}$ | $33.7_{\pm 8.7}$ | $15.9_{\pm 4.6}$ | |
| | Naive unweighted $(G=Q^{\pi_e})$ | $0.128_{\pm 0.006}$ | $207.2_{\pm 91.5}$ | $0.089_{\pm 0.089}$ | $50.0_{\pm 6.0}$ | $11.6_{\pm 8.3}$ | $78.1_{\pm 13.6}$ | |
| | Naive weighted $(G=Q^{\pi_e})$ | $0.097_{\pm 0.006}$ | $300.8_{\pm 117.6}$ | $0.420_{\pm 0.097}$ | $64.3_{\pm 4.7}$ | $24.0_{\pm 12.7}$ | $44.3_{\pm 11.4}$ | |
| **Proposed** | C\*-PDIS $(G=Q^{\pi_e})$ | **0.013** $_{\pm 0.005}$ | **994.0** $_{\pm 10.1}$ | **0.995** $_{\pm 0.003}$ | **95.7** $_{\pm 3.1}$ | 4.5 $_{\pm 6.9}$ | **4.2** $_{\pm 5.3}$ | } ★ ideal case |
| | C\*-PDIS $(G=Q^{\pi_b})$ | 0.070 $_{\pm 0.003}$ | **994.0** $_{\pm 10.1}$ | 0.961 $_{\pm 0.011}$ | 86.8 $_{\pm 8.2}$ | 22.0 $_{\pm 20.1}$ | 8.2 $_{\pm 11.3}$ | } relaxing |
| | C\*-PDIS $(G=Q^{\pi_b} \mapsto \hat{Q}^{\pi_e})$ | 0.028 $_{\pm 0.007}$ | **994.0** $_{\pm 10.1}$ | 0.979 $_{\pm 0.010}$ | 90.1 $_{\pm 5.4}$ | **4.2** $_{\pm 6.6}$ | 14.1 $_{\pm 9.7}$ | Assump. 2 |

| | Estimator | ↓ **RMSE** | ↑ **ESS** | ↑ **Spearman** | ↑ **%Accuracy** | ↓ **%FPR** | ↓ **%FNR** |
|---|---|---|---|---|---|---|---|
| **Breakdown OIS vs WIS** | PDOIS (w/o annot.) | $0.079_{\pm 0.054}$ | $76.8_{\pm 44.0}$ | $0.868_{\pm 0.087}$ | $79.8_{\pm 5.3}$ | $6.4_{\pm 7.3}$ | $30.3_{\pm 8.8}$ |
| | PDWIS (w/o annot.) | $0.136_{\pm 0.033}$ | $76.8_{\pm 44.0}$ | $0.523_{\pm 0.178}$ | $73.2_{\pm 5.1}$ | $61.1_{\pm 13.9}$ | $1.6_{\pm 3.4}$ |
| | C\*-PDOIS $(G=Q^{\pi_e})$ | **0.013** $_{\pm 0.005}$ | **994.0** $_{\pm 10.1}$ | **0.995** $_{\pm 0.003}$ | $95.6_{\pm 3.2}$ | $4.5_{\pm 6.9}$ | $4.3_{\pm 5.5}$ |
| | C\*-PDWIS $(G=Q^{\pi_e})$ | **0.013** $_{\pm 0.005}$ | **994.0** $_{\pm 10.1}$ | **0.995** $_{\pm 0.003}$ | **95.7** $_{\pm 3.0}$ | $4.5_{\pm 6.9}$ | **4.1** $_{\pm 5.1}$ |
| | C\*-PDOIS $(G=Q^{\pi_b})$ | $0.070_{\pm 0.003}$ | **994.0** $_{\pm 10.1}$ | $0.962_{\pm 0.012}$ | $86.7_{\pm 8.3}$ | $20.2_{\pm 20.1}$ | $8.3_{\pm 11.4}$ |
| | C\*-PDWIS $(G=Q^{\pi_b})$ | $0.070_{\pm 0.003}$ | **994.0** $_{\pm 10.1}$ | $0.961_{\pm 0.012}$ | $86.9_{\pm 8.1}$ | $19.8_{\pm 20.1}$ | $8.1_{\pm 11.2}$ |
| | C\*-PDOIS $(G=Q^{\pi_b} \mapsto \hat{Q}^{\pi_e})$ | $0.028_{\pm 0.007}$ | **994.0** $_{\pm 10.1}$ | $0.979_{\pm 0.010}$ | $90.1_{\pm 5.4}$ | **4.2** $_{\pm 6.6}$ | $14.1_{\pm 9.7}$ |
| | C\*-PDWIS $(G=Q^{\pi_b} \mapsto \hat{Q}^{\pi_e})$ | $0.028_{\pm 0.007}$ | **994.0** $_{\pm 10.1}$ | $0.979_{\pm 0.010}$ | $90.1_{\pm 5.4}$ | **4.2** $_{\pm 6.6}$ | $14.1_{\pm 9.7}$ |

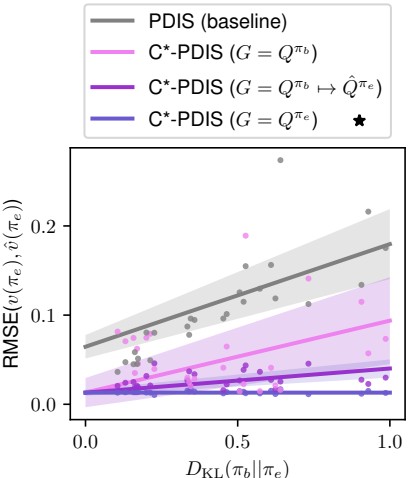

Figure 10: RMSE of C*-PDIS vs. distance to $\pi_b$ (in terms of KL divergence) for each $\pi_e$, plotted with linear trend lines. OPE error increases as $\pi_e$ becomes more different from behavior.

**Variance reduction of C*-PDIS outweighs the effect of noisy annotations.** To understand the robustness of our estimator to annotation noise, we perturbed the annotations with varying amounts of noise. Specifically, each annotation is added with a noise value drawn from zero-mean Gaussian distributions with a pre-specified standard deviation which we vary. As the level of annotation noise increases (Figure 11-left), performance degradation is minimal even at the highest level of noise tested (with a std of 1, which is large relative to the reward range $[-1, 1]$, and most notably larger than $0.31$, the std of initial state values of this domain). Our estimator remains competitive relative to the baseline PDIS, suggesting that the benefit of variance reduction from additional data (through counterfactual annotations) outweighs the variance increase from annotation noise, even when annotations are much noisier than factual data. The same trend holds when only $10\%$ of the counterfactual annotations are collected (Figure 11-right).

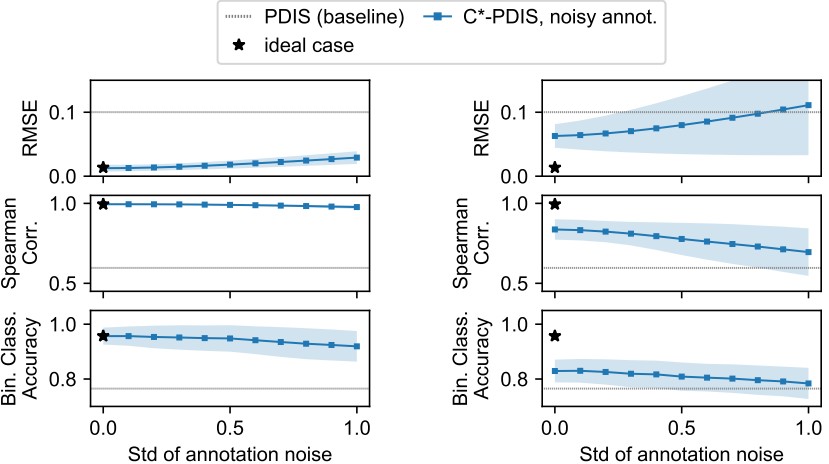

Figure 11: Performance of our proposed C*-PDIS estimator is generally robust to noisy annotations. Trend lines show average of 50 runs $\pm$ one std. When all annotations are available (left), the performance degradation of C*-PDIS is minimal even at the highest level of noise tested. When only $10\%$ of the counterfactuals are annotated (right), performance degradation is more noticeable and eventually becomes worse than baseline PDIS without annotations.

**Collecting more annotations and imputing missing annotations improves performance.** As the amount of available annotations increases (Figure 12), our approach interpolates between baseline PDIS and the ideal case of C*-PDIS with an monotonic improvement in performance. Furthermore, imputing annotations (as described in Appendix D.2) achieves better performance, suggesting it is a promising strategy to handle missing annotations when not all annotations can be obtained in practice. The same trend holds under different amounts of annotation noise (Figure 12 left vs right).

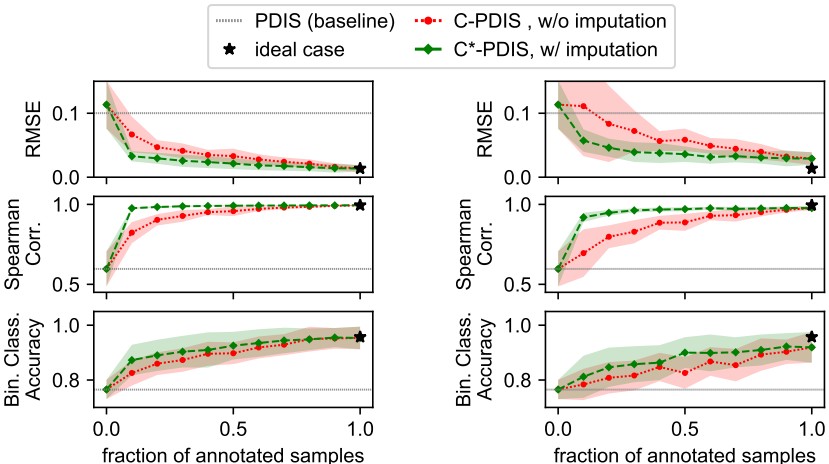

Figure 12: Performance of our proposed C*-PDIS estimator is robust to missing annotations, especially when the missing annotations are imputed. Trend lines show average of 50 runs ± one std. As the fraction of annotated samples increases, performance interpolates between baseline PDIS and the ideal case where all counterfactual annotations are available. The imputed version outperforms the unimputed version and maintains a competitive performance (relative to the ideal case) even in the presence of high degrees of missingness. The same general trend holds across the two settings with different amounts of annotation noise (left: std of 0.2, right: std of 1.0), though the imputed annotations have a larger bias when annotations are noisier, leading to slightly worse performance even when all annotations are available.

