# OpenReview forum: "Counterfactual-Augmented Importance Sampling for Semi-Offline Policy Evaluation"
_NeurIPS.cc/2023/Conference — NeurIPS 2023 poster_

### Official Review · Reviewer_rPY7 · 2023-06-13

**Soundness:** 3 good
**Presentation:** 4 excellent
**Contribution:** 3 good
**Rating:** 7
**Confidence:** 4

**Summary:**

The paper proposed a novel method for off-policy evaluation with offline data that entails human input. Specifically, the paper assumes access to human annotations for counterfactual state-action pairs as counterfactual immediate reward (in bandits) or Q values (in RL) under the behaviour policy (as a relaxation to Q values under the evaluation policy). They used a bandit example to show that simply incorporating counterfactual annotations into the offline data can shift state distribution and therefore bias overall estimates. To keep the state distribution unchanged, they further assume access to a human-privided weight for each visited / annotated counterfactual state-action pair. Effectively this creates an augmented behaviour policy as if smoothed towards a uniform policy. They are then able to reweigh the importance sampling ratios and conduct OPE as a weighted sum of the conventional backwards bootstrapping and human annotation. They then move on to give theoretic analyses about the bias and variance of the proposed OPE estimator. Experiments conducted on a synthetic bandit problem and a healthcare problem with simulator demonstrated overall improved performance than vanilla IS and ablated versions of the proposed method itself

**Strengths:**

1. proposed a novel method for OPE with offline data, which seems increasingly timely as machine learning is marching into real life
2. experiments have tested thorough aspects of the proposed idea

**Weaknesses:**

1. even if the paper goes into length to keep state distribution not affected by incorporating human annotations, it still does not address the distribution shift between the state distribution induced by the behaviour policy and by the evaluation policy
2. expert annotations for counterfactual state-action pairs along with their weights are also expensive to get (though argubly safer / easier than rolling out the evaluation policy in real world) and inevitably subjective
3. unclear how the promising results could extend to real-world scenarios

**Questions:**

1. how would human experts provide Q values under the behaviour policy in practice? It does not seem any easier than providing
Q values under the evaluation policy both of which entail a sequence of counterfactual state-action pairs that are not their own choices
2. Appendix D.2, to relax Q under the evaluation policy to Q under the behaviour policy in the annotation, why simply considering a bias term in the annotation is enough? It seems to me you would also need to reweigh the annotation based on how relatively likely the counterfactual action is selected. Yes you have indeed reweighed for this between the evaluation policy and the augmented behaviour policy, but now you have a third policy, i.e. the original behaviour policy before augmentation (under which you have the Q values). I would imagine in addition to the IS ratio as you have in place already, you would also need to reweigh the Q value with e.g. product of subsequent IS ratios between the evaluation policy and the original behaviour policy. However, I see in your experiments you showed that the performance is not much affected with biased annotations, so I suppose not reweighing the biased annotations is not a major issue, but it would still be good to see further justification for that.

**Limitations:**

please see above

---

> ### Author Rebuttal · Authors · 2023-08-09
>
> We thank Reviewer rPY7 for the time spent reading our paper (including the appendix) and their helpful feedback. We are also encouraged by their overall positive assessment of our paper. Below we address the reviewer's questions and concerns.
>
> **1. The proposed method does not address state distribution shift between $\pi_e$ and $\pi_b$**
>
> Vanilla IS (and PDIS) are unbiased under the common support assumption (Assumption 3 in our paper) and do not require explicit handling of the state distribution shift due to different policies. Our paper points out the bias issue of naive incorporation of counterfactual annotations (where states with more annotations are over-represented) and provides a simple reweighting solution to address it.
>
> - We also acknowledged in related works (L562-L564) that there is a family of approaches based on importance weighting on marginalized state distributions $d_{\pi}(s)$. We believe that similar modifications to the MIS/DICE estimator family can be made, but require separate full analyses on their bias/variance, which is outside the scope of our current paper.
>
> **2. Is it even practical to collect counterfactual annotations? “How would human experts provide Q values under the behaviour policy in practice?”** (see also: [overall response](/forum?id=dsH244r9fA&noteId=jKEOBV6cyp), [noPk](/forum?id=dsH244r9fA&noteId=l1RyMRJL8s))
>
> One domain that the authors have the most experience in is healthcare, where the goal is to optimize sequential treatment policies (e.g. Komorowski et al. 2018). In this domain, clinicians are constantly evaluating alternative treatment paths in their minds when making treatment decisions but we only observe what was actually done; counterfactual annotations can be seen as a mechanism to elicit this information in their thought process that is otherwise not recorded. We envision clinicians to be asked questions such as “given that the patient received treatment A, if treatment B was used instead, what do you think would happen to the patient”.
>
> [1] Komorowski et al. “The Artificial Intelligence Clinician learns optimal treatment strategies for sepsis in intensive care”. Nature Medicine, 2018.
>
> - Re subjectivity of annotations: while an inevitable property of human annotations, we want to point out two potential remedies that could improve the usefulness of our approach: (i) we can average the annotations from multiple annotators to reduce annotation variance, (ii) our approach is relatively robust to high levels of annotation noise (as seen in Fig 5 experiments), and informally speaking, if the annotation noise is similar to observed reward variance, then the variance reduction we get is as if we have collected more offline data. We also believe there is opportunity for future research especially in the HCI space (which we mention on L530-L534) to improve annotation solicitation (e.g. better question phrasing to reduce subjectivity in response).
>
> **3. Question about Appendix D.2, why simply considering a bias term in the annotation is enough?**
>
> Thank you for raising this detailed question, we are happy to provide more clarification to your question (main points are summarized below).
> > “why simply considering a bias term in the annotation is enough? ... you would also need to reweigh the Q value with e.g. product of subsequent IS ratios between the evaluation policy and the original behaviour policy…”
>
> We would like to refer the reviewer to the unbiasedness proof of C-PDIS, which we have sketched in L272-L279 and illustrated in Figure 4, and fully derived in Appendix C.4 (starting L725 on page 21). In Figure 4 (ii) and (iii), what we require for unbiasedness is that both the factual and counterfactual branches provide unbiased estimates of Q under $\pi_e$, for factual $a_t$ and counterfactual $\tilde{a}$ respectively. When using the bias-correction approach in Appx D.2, we are directly mapping the annotation from Q of $π_b$ (the original behavior policy) to Q of $π_e$ (the evaluation policy) by estimating their difference (i.e. the annotation bias). Indeed, an alternative approach to correct this bias is to apply IS, as proposed by the reviewer by using the “product of subsequent IS ratios between the evaluation policy and the original behaviour policy”. However, the main challenge here is that we only have the annotation for $\tilde{a}$ but no subsequent sub-trajectory information (recall Fig 4(iii)). We have experimented with a version of this approach where we search for all matching sub-trajectories that start with $(s_t, \tilde{a})$ and use the average subsequent IS ratio to reweight, but we found that it introduced a lot more variance (due to limited number of matching sub-trajectories) than the model-based bias-correction procedure described in the paper.

---

> > ### Comment · Reviewer_rPY7 · 2023-08-14
> >
> > Thank you authors for your response.
> >
> > 1. I acknowledge that my state distributional shift concern is addressed. The authors did point out that reweighing with marginalized state distributions is an orthogonal direction.
> >
> > 2. I'm half-convinced by the accuracy and ease of getting counterfactual human annotations, they are mind processes after all. But this aside, the technical part of the paper seems of value and self-contained.
> >
> > 3. Regarding the bias term in appendix D.2, so you mean you are not learning/correcting for this term but only pointing out its existence?

---

> > > ### Author Response · Authors · 2023-08-14
> > >
> > > Dear Reviewer rPY7, thank you for the reply.
> > >
> > > For **3.**: yes we are learning/correcting the bias term. Sorry if our initial response wasn't clear. In Appendix D.2,
> > >
> > > - the first paragraph (L772-L777) points out the bias issue,
> > > - the second paragraph (L778-L786) describes a model-based procedure to learn and correct this bias. In the equation after L783, $\hat{g}^{(i),\tilde{a}}_t = g^{(i),\tilde{a}}_t - \hat{\epsilon}\_{G_t}(s_t, \tilde{a})$, the learned bias is subtracted from the annotation $g$ so that it has the correct, “unbiased” expectation.
> > > - We also acknowledge (L785) when this is not possible (e.g. due to missing support, when the annotated counterfactual action did not appear in offline data), in which case we recommend using the obtained annotation as is.

---

> > > > ### Comment · Reviewer_rPY7 · 2023-08-18
> > > >
> > > > Thank you. I keep my rating, due to lack of real-world validations.

---

> > > > > ### Author Response · Authors · 2023-08-18
> > > > >
> > > > > Dear Reviewer rPY7, thank you for the reply and we're glad that we've answered all your questions!
> > > > >
> > > > > Regarding real-world validations: we wholeheartedly agree with you that real-world human experiments would be the necessary next step to realize the impact of our work; we have explicitly mentioned this under limitations and as future work (L520). We hope readers will find our insights valuable (especially the failure of the naive approach, and practical considerations of how to use our proposed estimator most effectively), and we encourage researchers and practitioners in different research areas (e.g., RL, HCI, healthcare, education) to build upon the ideas in our work.

---

### Official Review · Reviewer_fzzo · 2023-07-06

**Soundness:** 2 fair
**Presentation:** 4 excellent
**Contribution:** 2 fair
**Rating:** 5
**Confidence:** 4

**Summary:**

- Propose a setting in OPE where experts can annotate the value of counterfactual actions.
- A new importance sampling scheme for the setting is proposed.
- Theoretical analysis and experiments confirm superiority over the unweighted case, etc.

**Strengths:**

1. The proposed setting of expert annotating the counterfactual values seems like a direction worth exploring
1. Clearly written with some theoretical guarantees. In particular, the bias and variance are carefully discussed for cases where assumptions such as the coverage are not met, and benefit such as bias reduction compared to importance sampling (IS) is shown.
1. A useful heuristic is proposed (equal weighting)

**Weaknesses:**

1. The method itself and the baselines are fairly naive.
    1. Comparisons with standard OPE baselines, such as the direct and doubly robust methods as well as the IS method, may be of value to better estimate the benefit of annotating counterfactuals for readers.
    1. Also have a question about the claimed problem with the naive IS method, which is pointed out in section 3.1. See Question 2 below.
1. Other similar annotation augmentation approaches (e.g., [1,2], but not limited to) are not compared neigher theoretically nor experimentally. I believe that the proposal to add counterfactual annotations is a costly and significant change to the problem setting and should be widely compared to similar approaches widely.

[1] Srivastava, Megha, Tatsunori Hashimoto, and Percy Liang. "Robustness to spurious correlations via human annotations." International Conference on Machine Learning. PMLR, 2020.

[2] Kaushik, Divyansh, Eduard Hovy, and Zachary Lipton. "Learning The Difference That Makes A Difference With Counterfactually-Augmented Data." International Conference on Learning Representations. 2019.

**Questions:**

1. What is the exact definition of "naive weighted"? What is the theoretical advantage of the proposed method?
1. The problem with the naive IS method is shown in section 3.1, but is this not simply because the state visitation probability $d_\pi(s)$ is not corrected? The problem can be solved by simply applying any OPE method that takes $d_\pi(s)$ into account, such as DualDICE [3], to the augmented dataset.

[3] Nachum, Ofir, et al. "Dualdice: Behavior-agnostic estimation of discounted stationary distribution corrections." Advances in neural information processing systems 32 (2019).

**Limitations:**

1. The proposed method requires costly accurate annotation, which is properly investigated theoretically and experimentally.

---

> ### Author Rebuttal · Authors · 2023-08-09
>
> Thank you for taking the time to read and evaluate our work. Below, we provide clarifications and answers to your questions.
>
> **1. “The method itself and the baselines are fairly naive”** (see also: [overall response](/forum?id=dsH244r9fA&noteId=jKEOBV6cyp), [71Lt](/forum?id=dsH244r9fA&noteId=tJFHuJCKwM))
>
> In this paper, we establish a formal framework of ‘semi-offline’ evaluation (offline data + counterfactual annotations), and propose a new estimator for this setting based on importance sampling. While the C-IS estimator is simple, our analysis **reveals new theoretical insights and important practical considerations** for practitioners to keep in mind, e.g. it’s better to impute missing annotations and use equal weights. Therefore, we believe our paper makes non-trivial contributions to the community and is an important step to enabling RL applications in high-stakes domains such as healthcare.
>
> - Comparison to other OPE methods: We believe the relevant comparison is C-IS against IS (and C-PDIS against PDIS), as **improvement of C-PDIS over PDIS directly illustrates the benefit of counterfactual annotations**. In other words, whenever good counterfactual annotations are available, our results (theoretical + empirical) suggest that the counterfactual-augmented version C-PDIS should be preferred over vanilla PDIS. Results for other OPE methods are not shown as they are not directly relevant to supporting our main claim on the utility of counterfactual annotations (and how to incorporate them to IS-based estimators). In addition, past benchmarking works have found that the best OPE method is environment-specific and has recommended multiple OPE methods should be used in practice [Fu et al. 2021, Voloshin et al. 2021]. We believe there is opportunity to augment other classes of OPE methods with counterfactual annotations and have highlighted this potential in related work (in Appendix A.1 on page 12).
>
>     [1] Voloshin et al. “Empirical Study of Off-Policy Policy Evaluation for Reinforcement Learning”. NeurIPS D&B 2021.
>     [2] Fu et al. “Benchmarks for Deep Off-Policy Evaluation”. ICLR 2021.
>
> **2. Relationship to other annotation/augmentation approaches**
>
> We appreciate these additional references. It is certainly true that the annotations required in our approach will share similar challenges to those in the cited works (e.g. subjectivity, inter-rater disagreement), and we will expand our discussion on this in the revised version. Here, we would like to point out a few key distinctions between our approach and these past work: compared to Kaushik et al. ICLR 2019 (annotators alter text to match a counterfactual target label) and Megha et al. ICML 2020 (humans provide annotations of potential unmeasured confounder), our formulation of counterfactual annotations is more naturally suited to sequential decision making in RL, focused on the unobserved trajectories in the offline setting. This is an underexplored area with great potential to enable practical RL in high-stakes domains.
>
> **3. What’s the exact definition of naive weighted?**
>
> We believe the reviewer is referring to the “naive weighted” baseline that was mentioned on L361 in the sepsis simulator experiments, whose results are shown in Table 2 on page 9. We have informally defined “naive weighted” on L361 as an approach that “reweights the annotations at the trajectory level instead of per-decision” (we weren’t able to elaborate in the main text due to space constraint). More formally (wlog), assuming a binary action space $\\{0,1\\}$, given a trajectory of length $T$ with counterfactual annotations at each step, $\tau = [s_t, a_t, r_t]\_{t=1}^{T}$, $\boldsymbol{g} = \\{g_t^{1-a_t}\\}\_{t=1}^{T}$ where $1-a_t$ is the counterfactual action for $a_t$, the naive weighted estimator is defined as
>
> $(1-\sum\_{t=1}^{T} w_t) \rho_{1:T} (\sum\_{t=1}^{T} r_t) + \sum\_{t=1}^{T} \big( w_t \rho_{1:t-1} \rho_t^{1-a_t} (\sum\_{t’=1}^{t-1} r_{t’} + g_t) \big)$
>
> Intuitively, this first converts each annotation into a sub-trajectory that terminates at the step of annotation with the counterfactual action $[s_1, a_1, r_1, \cdots, s_t, 1-a_t, g_t]$, and then performs IS on each sub-trajectory (including the original trajectory), and finally computes a weighted sum of these T+1 estimates (1 factual estimate, T counterfactual estimates) using weights $(1-\sum\_{t=1}^{T} w_t)$, $w_1, \cdots, w_T$.
>
> We have elaborated the reason why it does not work in Appendix E.2, L886-L891 on page 28:
> > The reason why “naive weighted” does not work is more subtle: while reweighting the (partial) trajectories constructed from the counterfactual annotations correctly maintains the initial state distribution, it does not correctly maintain the intermediate state distributions.
>
> **4. Why not apply OPE methods that take state distribution d_π(s) into account?**
>
> We chose to focus on importance sampling based OPE methods due to their simplicity of implementation and common use, as well as comprehensive theoretical guarantees as discussed in past work [1][2]. In the paper, we do provide a solution to the bias issue with the naive incorporation of counterfactual annotations to IS. On the other hand, we acknowledged in related works (L562-L564) that there is a family of approaches based on importance weighting on marginalized state distributions $d_{\pi}(s)$. We believe that similar modifications to the MIS/DICE estimator family can be made - and are also a valid solution to the overall problem - but require separate full analyses on their bias/variance properties, which is outside the scope of our current paper.
>
> [1] Levine et al. “Offline Reinforcement Learning: Tutorial, Review, and Perspectives on Open Problems”. arXiv 2020.
> [2] Precup. “Eligibility traces for off-policy policy evaluation”. 2000.

---

> > ### Comment · Reviewer_fzzo · 2023-08-18
> >
> > My concern is about the nontriviality of this work.
> > It is obviously better to incorporate annotation-augmented data if available; thus it is difficult to believe that the lack of such practice is due to people not knowing that it is better to incorporate annotation-augmented data.
> > Rather, it would be normal to assume that this is because of the high cost or inaccuracy of obtaining such annotation.
> > Therefore, the difference of augmentation with annotation, i.e., the difference with PDIS alone, does not seem to be worthy for such a top venue.
> >
> > In my understanding, the motivation that might answer the above question is described in Section 3.1.
> > Author(s) describes an issue when weighting (IS) is simply applied to the augmented data.
> > However, it does not take the difference in state visitation probability $d_\pi(s)$ into account (since they compare only within the "per-decision" approaches).
> > Therefore, a natural question would be "What if simply apply (trajectory-level) weighting methods such as DICE to the augmented datasets?"
> > If such a strategy was enough, the contribution of this work would be limited.
> > Contrasting this with such a naive baseline would make the contribution of this paper easier to understand.
> >
> > According to the Author(s)' response (3), the compared baselines ("naive weighted") are also per-decision, and my concern is not clear.

---

> > > ### Author Response · Authors · 2023-08-18
> > >
> > > Thank you Reviewer fzzo for the reply. We appreciate your engagement in discussions and would like address your concerns regarding our work's contributions:
> > >
> > > - **"it is difficult to believe that the lack of such practice is due to people not knowing that it is better to incorporate annotation-augmented data"**: to our knowledge, we are the first to pose the task of semi-offline evaluation using the specific combination of *'offline data + counterfactual annotations'* in the RL OPE setting - we're happy to discuss the distinctions wrt further related work if exists, in addition to the references you already provided. We therefore believe what we propose constitutes a *new task* of interest to the NeurIPS community and has the potential to be *adopted by practitioners* in high-stakes offline RL domains to as a safe mechanism to gain more confidence in new policies.
> > >
> > > - **"high cost or inaccuracy of obtaining such annotation"**: We agree that, as with any data annotations, there will be cost and variability associated with our proposed counterfactual annotations. However, compared to online deployment which is the gold-standard for evaluation in RL, collecting such annotations is arguably a safer, less costly alternative, especially in high-stakes domains such as healthcare or autonomous driving, and is an important step to realize the real-world impact of ML in these domains.
> > >   - Re cost:
> > >
> > >     The success of modern supervised learning is built upon large labeled/annotated datasets, which are associated with high costs. We do not believe cost alone should deter research, but rather it needs to be justified with respect to the potential benefits and ethical/safety considerations. In high-stakes RL domains, the dangers and uncertainty of online evaluation (e.g. recommending potentially suboptimal or wrong treatments to clinicians) can often make it justifiable to collect counterfactual annotations instead.
> > >
> > >     That said, there are also ways to reduce cost: one way is to incorporate the counterfactual annotation prompt(s) into existing supervised annotation pipelines, e.g. when annotating a patient (retrospectively/offline), we can simultaneously ask clinician annotators to provide disease diagnosis(es) as well as predicted effect(s) of alternative treatment(s); another possibility is to collect counterfactual annotations on-the-fly in a "semi-online" fashion, e.g. at the end of daily grand rounds, ask clinicians to record the predicted effect(s) of alternative treatment(s) they did not select for patients they just saw.
> > >
> > >   - Re inaccuracy: it is certainly true that human annotations involve the challenges of subjectivity, ambiguity, and annotator bias; we also agree that our new framing of "counterfactual reasoning" in annotations may introduce additional variability. It is thus important to incorporate existing remedies (provide clear annotation guidelines and training sessions, and recruit diverse representative annotator teams) and to conduct human experiments in the domains of interest so we can understand any additional challenge our approach introduces and their downstream impact. For example, in the context of healthcare, it would be interesting to measure the inter-rater reliability in disease diagnosis labels (used in supervised learning) vs counterfactual annotations (used in semi-offline eval) across multiple annotators -- which in our opinion is a research question complementary to our present paper and worthy of a separate investigation. That said, our experiments (Sec 5.2) provide some initial evidence that imperfect counterfactual annotations can still be valuable despite being noisy or biased.
> > >
> > > - **"difference with PDIS alone ... does not seem to be worthy for such a top venue"**: we'd like to point out that our contribution is not simply the C-PDIS estimator *alone* - our contributions include (1) framing the 'semi-offline evaluation' task, (2) the C-PDIS estimator which is a simple modification of existing IS estimators, (3) perhaps most importantly, theoretical analyses that justify how to best apply this estimator in practice. Specifically, (3) includes the bias-correction procedure (L293; Appx D.2) as well as the recommendation of using equal weights after imputing missing annotations (L286; Appx D.1). These insights -- rooted in our extensive theoretical analyses (Sec 4) and otherwise unknown to readers -- are substantiated by empirical evidence (Sec 5) demonstrating their advantages over applications of C-PDIS without incorporating these strategies. Our work thus provide unique findings and valuable guidance for practitioners.
> > >
> > > We further respond to the question about DICE-based approaches in a follow-up comment below.

---

> > > > ### Author Response · Authors · 2023-08-18
> > > >
> > > > - **"What if simply apply (trajectory-level) weighting methods such as DICE to the augmented datasets?"**: We do not believe the existence of a potential solution based on stationary distribution correction discredits our proposed method as being a valid solution to the problem; rather, the two are orthogonal directions (as acknowledged by Reviewer rPY7).
> > > >
> > > >   We also want to highlight that these DICE-based approaches *cannot be simply applied*: below, we illustrate several difficulties/unclearness in an attempt to apply stationary distribution weighting methods to this setting. For context, this family of estimators are often define the policy value estimate as $\mathbb{E}\_{(s,a) \sim d_{π_b}, r \sim R(s,a)}[w_{π_e/π_b}(s,a)r] = \mathbb{E}\_{(s,a,r) \sim d_{D}}[w_{π_e/π_b}(s,a)r]$ and the goal is to estimate the stationary distribution ratio $w\_{π_e/π_b}(s,a) = d_{π_e}(s,a)/d_{π_b}(s,a) = \big(d_{π_e}(s) π_e(a|s) \big) / \big(d_{π_b}(s) π_b(a|s) \big)$.
> > > >
> > > >   - First, in the bandit case (e.g. Sec 3.1 example), it should be clear how reweighting using $d_π(s)$ easily solves the problem, which we do not elaborate here. The main challenge comes in when trying to apply it in the MDP setting.
> > > >   - **Challenge 1:** *how to handle immediate reward vs cumulative reward*: recall the definition of the estimator above, which reweighs immediate rewards $r \sim R(s,a)$. However, the counterfactual annotations $g \sim G(s,a)$ correspond to cumulative reward, and each $(s,a)$ may be associated with both factual sequences of immediate rewards and counterfactual annotations. It is not straightforward how to modify $\mathbb{E}\_{(s,a,r) \sim d_{D}}[w_{π_e/π_b}(s,a)r]$ to accomodate this distinction. **Potential solution:** One may want to re-define the MDP to have an absorption state after the counterfactual annotation step.
> > > >   - **Challenge 2:** *the change in $d_π(s)$ due to annotations is different from the change in $d_π(s)$ due to policy shift*: the ratio $w_{π_e/π_b}(s,a)$ learned by DICE addresses the shift in $d_π(s)$ under $π_b$ vs $π_e$, however the counterfactual annotations introduce an additional shift in the data distribution that is not accounted for by either $π_b$ or $π_e$, but a separate process that controls which samples are annotated. **Potential solution:** Some redefinition of $π_{b^+}$ similar to our notion of augmented behavior policy that incorporates the annotation process may be necessary, meaning we may need to use $d_{π_{b^+}}$ instead of $d_{π_{b}}$.
> > > >   - **Challenge 3:** *it's unclear existing methods can learn the “correct $w_{π_e/π_b}(s,a)$ ratio"*: suppose we convert the augmented dataset containing full trajectories plus partial trajectories ending with annotations. This is not a naturally occurring dataset generatable by any policy (even $π_{b^+}$) in the original MDP, and we must define some notion of “augmented MDP” (perhaps with new absorption states, modified initial state distribution and/or transition probabilities). In other words, the data distribution follows $π_{b^+} \times M^+$. However, we want to reweigh this data to the distribution $π_e \times M$ i.e. the target policy under the original MDP. It is unclear whether the ratio $d_{π_e \times M} / d_{π_{b^+} \times M^{+}}$ still satisfies the same theoretical guarantees, of either the backward-Bellman recurrence in e.g. Liu et al. Theorem 1&4 or the forward-Bellman recurrence in Nachum et al. Sec 3.1.
> > > >
> > > >   In either case, we believe these modifications necessary for applying DICE are no longer a *simple, straightforward* application (we’re also not sure if our potential solutions completely address all issues and lead to an unbiased estimator). We therefore believe DICE (or other MIS) approaches + counterfactual annotations is a standalone direction worthy of its own investigation.
> > > >
> > > >     [Liu et al.] "Breaking the Curse of Horizon: Infinite-Horizon Off-Policy Estimation". NeurIPS 2018.
> > > >     [Nachum et al.] "DualDICE: Behavior-agnostic estimation of discounted stationary distribution corrections." NeurIPS 2019.
> > > >
> > > > Once again, thank you for your time and continued engagement in discussions! We are happy to incorporate your valuable feedback with respect to clarifying the significance of our contributions.

---

> > > > > ### Comment · Reviewer_fzzo · 2023-08-20
> > > > >
> > > > > I appreciate Author(s)' effort to clear my doubt about the existence of trivial alternative reweighting schemes.
> > > > > I agree with these potential solutions and the challenges.
> > > > > Once I had thought that one does not have to care about $\pi_b$ but only $\pi_{b^+};$ though the annotation would be done on $Q^{\pi_b}$ (not on $Q^{\pi_b^+}$), which makes the problem setting different from the simple application of DICE to the augmented dataset.
> > > > > I raised my score to 5.
> > > > >
> > > > > My current main concern is only about the reality of obtaining faithful annotations, i.e., they might be biased (or obtained with large noise) even compared to $Q^{\pi_b}.$
> > > > > Also, annotations for early times (small $t$) would be more difficult.
> > > > > In order to make this approach realistic, it will be necessary to evaluate the robustness against such issues or address them.

---

> > > > > > ### Author Response · Authors · 2023-08-20
> > > > > >
> > > > > > Dear Reviewer fzzo, we are sincerely grateful for the time and effort you dedicated to thoroughly reviewing our (somewhat lengthy) response!
> > > > > >
> > > > > > Your last comment raises important points beyond the core research question we seek to address in our present paper, but are nonetheless crucial — and we are excited to pursue them as our future work. We agree that “annotations might be biased (or obtained with large noise) even compared to $Q^{\pi_b}$”, and we believe it’s important to empirically measure if/how various factors influence annotation quality, including $t$ (as you mentioned), which state-action is being annotated (and its associated inherent stochasticity), how often they appear in data (under $π_b$), etc.
> > > > > >
> > > > > > More broadly speaking, there could be a trade-off between quality vs utility of the annotations: for instance, annotating the terminal step is easy but probably doesn’t add much, whereas annotating the initial state is harder but more useful. Given an understanding of the factors driving annotation error/variability, it would then be important to study how to select which counterfactual annotations to query under a fixed annotation budget (whether in terms of time or monetary costs), in order to maximize the gains compared to OPE based on offline data alone.
> > > > > >
> > > > > >   > **Side note - thoughts on “annotations for early times would be more difficult”**: We agree that *intuitively* (at least this was what we thought initially), annotations for small $t$ are more difficult as they involve reasoning about longer sequences of counterfactual actions, but at the same time they are more useful because they provide more coverage (see Fig 1-right). However, anecdotally when we consulted our medical collaborator(s), they expressed it might actually be easier to annotate small $t$: for large $t$ one needs to follow through the factual action sequence to a certain point and then switch to counterfactuals afterwards, but for small $t$ such as the initial state, one only need to process the information from the initial state and think about the subsequent counterfactuals only. Thus, we hypothesize annotations at all steps might indeed involve similar cognitive loads, but of course this hypothesis needs to be verified experimentally with real human annotators.
> > > > > >
> > > > > > We will expand our discussion on Sec 4.4 Practical Considerations to include these additional points so that readers can more clearly appreciate that these are research directions worth exploring. Many of these questions delve into the realm of HCI and are outside the scope of this paper’s methodological contributions for offline RL and OPE, and as we embark on answering these questions in healthcare domains, we believe our paper will inspire fellow researchers and practitioners to explore these questions within their own domains of interest.

---

### Official Review · Reviewer_noPk · 2023-07-10

**Soundness:** 4 excellent
**Presentation:** 4 excellent
**Contribution:** 4 excellent
**Rating:** 6
**Confidence:** 4

**Summary:**

This paper offers a new OPE estimator leveraging additional annotations of counterfactual actions. The paper provided an extremely thorough and comprehensive discussion of the theoretical property of this estimator. This helps us develop an understanding of how it operates. This paper also discussed fully the limitations of the approach and how the estimator would behave if different assumptions have been violated.

I generally find this to be a good paper, with limited experiments and not enough real-world motivation, but I can potentially see an application of this method, even though, for now, I don’t yet know what it could be.

**Strengths:**

The theoretical discussion of the estimator is very clear and very thorough. The paper discussed how bias of the annotator functions would impact the bias of the estimator. The paper also discussed when and how annotations could help reduce bias and variance of the estimator. The paper is also very clearly written. The proofs in the appendix are easy to follow (although I have not checked all of them). The experiments in the appendix are well documented.

**Weaknesses:**

There are a few weaknesses of this paper that prevents it from having a higher impact.
1. No human experiment has been offered. There is no actual investigation of what human inter-annotator variance would look like. It is somewhat known in a few important domains, such as dialogue evaluation (perhaps not surprisingly, this is a huge application field where RL is applied to natural language processing), inter-annotator agreement on certain metrics can be huge [1] (Table 2). The authors didn’t collect real human annotations at all, and the paper lacks key data that can shed light on whether this approach can work with real human annotations. Also, collecting $\omega$ would seem to be difficult as well. It is unfortunate that the authors didn’t opt to do this, but I don’t think this eclipses other amazing parts of this paper.
2. The experiments are too synthetic. This is a complaint similar to but slightly different from the previous point. Most RL/Bandit tasks have real-world motivations. By using a synthetic bandit experiment and Sepsis, the authors did not convince me that these two tasks would clearly benefit from human annotations. When are human annotations desirable in an RL/bandit task? How hard is it to obtain human annotations, especially for a sequential setting where some level of “imagine what the policy would do in the future” is involved? Can the authors shed some light on what kind of real-world task they hope to apply this OPE estimator to?

[1] Lowe, Ryan, et al. "Towards an automatic turing test: Learning to evaluate dialogue responses." arXiv preprint arXiv:1708.07149 (2017).


**Questions:**

1. Section 4.2 discusses the variance of the annotation function that would impact the variance of C*-IS. It assumes that the annotation function has the same variance as the reward function. Can the authors provide some insights on what happens when the annotation function has a lower variance than the reward function (this is possible because human annotations are malleable) or higher variance?

**Limitations:**

The authors addressed the limitations in the paper.

---

> ### Author Rebuttal · Authors · 2023-08-09
>
> We are encouraged that the reviewer appreciated our clear/thorough theoretical analyses, as well as well-documented experiments. We also want to point out that we have provided full code implementation in the supplement. Below, we address the feedback from the reviewer:
>
> **1. Regarding human experiments**
>
> We resonate strongly with the reviewer that, in order to further realize the utility and potential impact of our work, human experiments on the annotation quality/variability would be necessary; we have explicitly mentioned this under limitations and as future work (L520). **We are grateful that the reviewer acknowledges the contributions of our paper even without a user study.** Importantly, our work sheds light on desirable properties of useful counterfactual annotations, which points to exciting directions of future research in RL as well as HCI (L530-L534); there is also opportunity to draw lessons from the NLP community about human annotation quality/variability (such as ref [1] mentioned by Reviewer noPk).
>
> Re the side note about “collection of w”: please see [overall response](/forum?id=dsH244r9fA&noteId=jKEOBV6cyp) paragraph **4.** (and we also responded to Reviewer [71Lt](/forum?id=dsH244r9fA&noteId=LuHvfFOj0Q) regarding this); to summarize, in the paper we explored different weight settings both theoretically and empirically, and we suggest using equal weights as per C*-IS as a useful heuristic.
>
> **2. What are the real-world application domains of our approach?** (see also: [overall response](/forum?id=dsH244r9fA&noteId=jKEOBV6cyp), [rPY7](/forum?id=dsH244r9fA&noteId=6fGRmonfcP))
>
> Our approach is best suited for high-stakes RL domains where online evaluation is risky, but domain experts can provide additional feedback on new policies. One domain that the authors have the most experience in is healthcare, where the goal is to optimize sequential treatment policies (e.g. Komorowski et al. 2018). In this domain, clinicians are constantly evaluating alternative treatment paths in their minds when making treatment decisions but we only observe what was actually done; counterfactual annotations can be seen as a mechanism to elicit this information in their thought process that is otherwise not recorded. We envision clinicians to be asked questions such as “given that the patient received treatment A, if treatment B was used instead, what do you think would happen to the patient”.
>
> The sepsis simulator experiment is our best attempt at using an existing RL domain with strong healthcare motivations, where we have full control over the simulation parameters (various policy configurations etc) and online access (to evaluate OPE methods with ground-truth). While admittedly imperfect (which we acknowledged as a limitation on L528), our simulation experiments do attempt to cover various realistic scenarios including missing/noisy/biased annotations and show the robustness of our approach.
>
> [1] Komorowski et al. “The Artificial Intelligence Clinician learns optimal treatment strategies for sepsis in intensive care”. Nature Medicine, 2018.
>
> **3. What happens if annotation variance differs from reward variance?** (see also: [overall response](/forum?id=dsH244r9fA&noteId=jKEOBV6cyp), [Qcjd](/forum?id=dsH244r9fA&noteId=uEp9MwDHCX))
>
> For clarity, in Theorems 6&13 we assume annotation variance $\sigma_G^2$ is the same as reward variance $\sigma_R^2$. More generally, we can assume some relationship between the two such that $\sigma_G(s,a)^2 = \sigma_R(s,a)^2 + \Delta_{\sigma}(s,a)^2$. Then, in the proof of Theorem 13 (Appendix C.3, page 13), the majority of the derivation stays the same except for term (3) where we now apply a different assumption on annotation variance (L712). The resulting variance decomposition will have an additional term $\mathbb{E}\_{s \sim d_1} \mathbb{E}\_{a \sim \pi_b(s)} [\sum\_{\tilde{a} \sim \mathcal{A} \setminus \\{a\\}} \rho^{+}(\tilde{a}|s)^2 \bar{W}(\tilde{a}|s,a)^2 \Delta_{\sigma}(s,\tilde{a})^2]$ related to the difference in variance $\Delta_{\sigma}^2$. Note that this also suggests that when the annotations have a larger variance than the rewards, we may want to assign it a smaller weight, and vice versa -- which makes sense intuitively.
>
> Similarly, the modified version of Theorem 6 will contain an additional third term:
>
> $\mathbb{V}[\hat{v}^{\textup{C*-IS}}] = \mathbb{V}\_{s\sim d_1}[V^{\pi_e}(s)] + \mathbb{E}\_{s\sim d_1} \mathbb{E}\_{a\sim \pi(s)}[\pi_b(a|s) \rho(a|s)^2 \sigma_R(s,a)^2]+ \mathbb{E}\_{s\sim d_1}[\sum\_{\tilde{a} \sim \mathcal{A} \setminus \\{a\\}} \pi_e(\tilde{a}|s)^2 \Delta_{\sigma}(s,\tilde{a})^2]$
>
> Where the original Theorem 6 can be obtained by setting $\Delta_{\sigma}^2$ to 0, making the last term vanish.
>
> We will add the full derivations to the appendix.

---

> > ### Comment · Reviewer_noPk · 2023-08-21
> >
> > The additional derivation is highly appreciated. This is in fact, quite interesting, and now the impact of human annotator variance is easily seen in the estimator. I hope the authors would consider adding a synthetic experiment that shows different settings of annotator variance (technically, lower human annotator variance than reward variance could also make the estimator variance lower, right? Similar to control variate techniques?) It would be nice to see this being shown in an experiment (even though the derivation already helps a lot).
> >
> > I will keep my score as is. I think the lack of real-world datasets or applications is definitely concerning, but the paper has made enough contributions to convince me that it can generate more discussions and inspire follow-up works.

---

> > > ### Author Response · Authors · 2023-08-21
> > >
> > > Dear Reviewer noPK, thank you for the reply!
> > >
> > > Re your suggestion on additional experiments with different settings of annotation variance: absolutely - this is a great idea! We already have some preliminary versions of this,
> > > - Fig 7(c-d): bandits where annotations have a larger or smaller variance than the reward function,
> > > - Fig 5-center: SepsisSim with varying annotation noise,
> > >
> > > but we agree this is worth exploring more systematically (e.g. in the SepsisSim experiments, we will add direct comparisons of annotator variance against reward variance), and we will include those results at revision.

---

### Official Review · Reviewer_Qcjd · 2023-07-22

**Soundness:** 3 good
**Presentation:** 2 fair
**Contribution:** 2 fair
**Rating:** 4
**Confidence:** 3

**Summary:**

The paper considers OPE with annotation to improve the performance of OPE. The paper explains why naively incorporating such information may lead to a biased estimate. And the paper provides a theoretical analysis of the bias and variance. Finally, the paper demonstrates the performance of the proposed method using synthetic bandit and healthcare experients.

**Strengths:**

OPE is well-known to suffer from high variance. And incorporating other information is important and relevant. The paper studies an interesting and important question and proposes useful solutions with theoretical guarantees.

**Weaknesses:**

The clarity of the paper needs to improve. The main concept of this paper, annotation, is first introduced in Section 2. And In the Introduction, the authors only mention this notion in Figure 1. I think the authors should introduce and discuss "annotation" in the introduction to facilitate the reader's understanding. There are some other confusing points that may hinder comprehension, which I list in the Question section.

Assumptions 1 and 2 seem very strong. Could the authors discuss the impact of the bias of the annotation?

It seems in the current formulation, a behavior trajectory with annotations is very similar to many behavior trajectories. At least in the bandit case, I think they are equivalent. If it is correct, it will affect the paper's novelty and contribution.

Related to the clarity, "distribution shifts" are mentioned in the abstract, but never show up again in the paper. I think it is better to discuss the issues of distribution shifts induced by different policies in RL.



**Questions:**

1. IN Section 3, are weights state-dependent? And it seems the weights are random. How should I understand it?

2. Why the variance does not depend on the variance of the annotation var(g)?

**Limitations:**

The authors discussed some limitations in Section 4.4.

---

> ### Author Rebuttal · Authors · 2023-08-09
>
> Thank you for taking the time to read and evaluate our work. Below, we list specific updates on which we plan to focus our efforts, as well as provide answers to the questions raised.
>
> **1. “The clarity of the paper needs to improve”**
>
> Thank you for your detailed suggestions on improving the clarity of the paper. In our revision we will address the following points:
>
> > **Paper should introduce the concept of annotation earlier**
> >
> > We agree. We plan to update the introduction to (i) provide concrete examples of what a counterfactual annotation means, specifically in the context of healthcare: “asking doctors what they think would happen to the patient if the opposite treatment was used instead, assuming the standard care procedure was followed afterwards”, and (ii) incorporate related works on annotations/augmentations mentioned by Reviewer fzzo, and point out that our definition of counterfactual annotation is specific to RL and differs from existing notions of annotations.
>
> > **Are weights state-dependent?**
> >
> > No. Weights are specific to **each sample**, not necessarily to each state. Here’s a clarifying example: given an MDP with a single state $s$ and two actions {↗, ↘}. Suppose we have two samples: sample #1 is simply ($s$,↗) with no annotation for ↘, so the weights to the two actions must be $[1,0]$; sample #2 is also ($s$,↗) but with an annotation for ↘, and the weights can be any two non-negative number that sums to 1 as specified by the user, e.g. $[0.8, 0.2]$. (Note that this doesn’t mean the weights are “random”, as the user may *deliberately* set weights to be equal $[0.5,0.5]$, or to $[1,0]$ which ignores the annotation if they believe it is of poor quality.) The average weight (L159) of these two samples is $\bar{W}(\$↗$|s,$↗$) = 0.9$, $\bar{W}($↘$|s,$↗$) = 0.1$, and is used in computing the augmented behavior policy per Definition 1. We hope this example, together with the description in Sec 3.2 as well as our code implementation in the supplement, can help clarify the definitions, and we will add this example to the appendix.
>
> > **Paper should discuss the issues of “distribution shift” induced by different policies in RL**
> >
> > Absolutely, we will expand the discussion of distribution shift in (offline) RL and how it is the main challenge of OPE, providing additional references such as [1][2][3][4]. Thank you for this suggestion.
> >
> > [1] Kumar, et al. "Stabilizing off-policy q-learning via bootstrapping error reduction." NeurIPS 2019.
> > [2] Kumar, et al. "Conservative q-learning for offline reinforcement learning." NeurIPS 2020.
> > [3] Laroche, et al. "Safe policy improvement with baseline bootstrapping." ICML 2019.
> > [4] Parbhoo, et al. "Generalizing off-policy evaluation from a causal perspective for sequential decision-making." arXiv 2022.
>
> **2. Assumptions seem very strong, what’s the impact of annotation bias?**
>
> We agree with the reviewer that Assumptions 1&2 are rather strong, and this is exactly why our paper also presents analyses and experiments for **when the assumptions are violated, i.e., annotations are biased**.
> - Under Sec 4.1, in Proposition 3 we provide detailed theoretical analysis of the effect of bias due to imperfect annotations, where the bias is $\epsilon_G$.
> - In Sec 4.4 (L293) and Appendix D.2 we further provide an approach to directly correct for annotation bias using a model-based approach.
> - In Sec 5.2 experiments on the sepsis simulator (L374), we empirically studied the impact of bias and we found our approach to be robust to annotation bias.
>
> **3. Behavior trajectory with annotations vs many behavior trajectories, are they the same?**
>
> No, they're not the same. If we understand the reviewer correctly, the interpretation of “many behavior trajectories” corresponds to the naive approach that we describe in Sec 3.1 - this approach is the first thing that came to our mind as well, but as it turns out it's not correct as it does not handle state distributions correctly: states that receive more annotations will be overrepresented in the augmented dataset, and this will bias the final estimate.
>
> **4. What happens if annotation variance differs from reward variance?** (see also: [overall response](/forum?id=dsH244r9fA&noteId=jKEOBV6cyp), [noPk](/forum?id=dsH244r9fA&noteId=l1RyMRJL8s))
>
> For clarity, in Theorems 6&13 we assume annotation variance $\sigma_G^2$ is the same as reward variance $\sigma_R^2$. More generally, we can assume some relationship between the two such that $\sigma_G(s,a)^2 = \sigma_R(s,a)^2 + \Delta_{\sigma}(s,a)^2$, which leads to an additional term that depends on $\Delta_{\sigma}(s,a)^2$ in Theorems 6&13 variance decomposition expressions. **Due to space constraints, please see paragraph 2 of the [overall response](/forum?id=dsH244r9fA&noteId=jKEOBV6cyp) (above) for detailed derivations, which we plan to add to the appendix.**

---

> > ### Comment · Reviewer_Qcjd · 2023-08-14
> > **Thanks for your explanation and clarification**
> >
> > Thanks for the author's explanation and clarification especially for 3.

---

> > > ### Author Response · Authors · 2023-08-16
> > >
> > > Dear Reviewer Qcjd, we really appreciate your timely reply! Given that we have provided answers to all the points the reviewer mentioned under "Weaknesses" and "Questions", we kindly ask the reviewer to reassess the paper, or outline any further concerns that the reviewer might have.
> > >
> > > Once again, we extend our sincere thanks to you for your time and valuable feedback!

---

### Official Review · Reviewer_71Lt · 2023-07-28

**Soundness:** 3 good
**Presentation:** 3 good
**Contribution:** 2 fair
**Rating:** 6
**Confidence:** 4

**Summary:**

This paper studies offline policy evaluation in reinforcement learning, where the behavior policy, some interaction trajectories, and some human annotated trajectories (modified from interaction trajectories) are available. The paper points out that, since the number of human annotated trajectories are not the same across interaction trajectories, naively treating these human annotated trajectories the same as interaction trajectories can introduce bias (for example, in bandits, the context distribution changes), even if human annotations are perfect. So the paper proposes to weight the trajectories such that for each interaction trajectory, the weights of this trajectory and the trajectories  annotated from this trajectory sum to one. This ensures estimating the value of a policy from the correct state distribution.

The paper analyzed the bias and variance of the proposed estimator in the bandit setting and the bias of the estimator in the RL setting, under some specific settings (sometimes when the weights are the same and every trajectory is annotated, sometimes human annotations are perfect) and conducted empirical evaluation to show that the proposed estimator can reduce estimation error.



**Strengths:**

The paper is well-written and easy to follow.

The paper studies an interesting problem of off-policy evaluation in reinforcement learning with the help of both interaction data and human annotations. This setting makes sense when data is sparse and human annotations are accurate and affordable.

The paper both theoretically and empirically analyze the bias and variance of the proposed estimator.




**Weaknesses:**

My major concern is about the novelty of the paper. The paper did not answer the question of how to set the weights of interaction data and human annotations. To me, it seems that weighting is an obvious approach to try when correcting for the state distribution. I believe there are other works using similar idea. For example [1]. The interesting question to me is how to set the weights for optimal estimation error reduction. Unfortunately, this is not discussed in the paper.

It would be nice to discuss the relationship between this work and policy evaluation from multiple loggers (e.g. [2]). To me, they are similar problems since human annotations are like collected from another logging policy (except we need to correct for the state distribution).

I am a bit confused about the definition of C^\star-IS. The authors mentioned that C^\star-IS corresponds to w = 1/|A|. The authors also mentioned that the weight should be zero when there are no annotations. This means only when we have full information data can we use C^\star-IS. First, this seems a less interesting setting to analyze since the fundamental problem of missing data in RL is gone. Second, in the experiments, some actions were not annotated, but C^\star-IS is still used. There seems to be a contradiction?

Minor:

In line 50, the paper mentioned that the estimator requires a weaker condition. To me, this is an overclaim, since in reality, only the support condition is weaker, but an additional strong assumption that human annotations are perfect is also required.

[1] Learning to Rank with Selection Bias in Personal Search Xuanhui Wang, Michael Bendersky, Donald Metzler, Marc Najork SIGIR 16

[2] Effective Evaluation using Logged Bandit Feedback from Multiple Loggers A. Agarwal and S. Basu and T. Schnabel and T. Joachims KDD 17

**Questions:**

See weaknesses.

**Limitations:**

The authors adequately addressed the limitations.

---

> ### Author Rebuttal · Authors · 2023-08-09
>
> We thank the reviewer for their overall positive assessment of our paper and valuable feedback, including providing additional references. We are encouraged that the reviewer found our paper to be well-written and recognized its potential impact. We address the main questions below:
>
> **1. How to set weights for optimal estimation error reduction?** (see also: [overall response](/forum?id=dsH244r9fA&noteId=jKEOBV6cyp), [noPk](/forum?id=dsH244r9fA&noteId=l1RyMRJL8s))
>
> We agree with the reviewer that this is indeed an interesting question that arises from our approach, and **in the paper we have provided a partial answer**.
> - We explored this empirically in Fig 7 (Appendix E.1, page 27), where we sweep the weights and measure the resulting estimator variance. We found that (L839) “the ideal weighting scheme is problem-specific, and certain weights may lead to higher variance compared to standard IS”. We also found that (L841) “C*-IS, though not always variance-minimizing, consistently achieves lower variance than standard IS”. Unfortunately due to space constraints we couldn’t include the full results and summarized as (L350) “Equal weights (in C*-IS) is a good heuristic though not always ‘optimal’ ” in the text. Since it is an important takeaway of our paper, we will expand upon this part given the one extra content page at camera-ready.
> - We also considered this problem analytically (solving for variance-minimizing weights in the bandit case, which turned into a quartic equation) and numerically (optimizing variance objective using SGD). However, unless problem-specific parameters are known, we have yet to come up with a general way to minimize variance. Therefore, we suggest C*-IS (with equal weights) as a heuristic and highlighted this as a direction for future research (L311) that would build upon our semi-offline evaluation framework.
>
> **2. Clarifying C\*-IS and missing annotations, is there a contradiction?**
>
> You are correct in noting that (i) C*-IS corresponds to w = 1/|A|, (ii) weights should be zero when there are no annotations, and (iii) in the experiments, some actions were not annotated i.e. missing. However, there is **no contradiction** because we proposed an approach to impute missing annotations, described in Sec 4.4 (L286-292) and Appendix D.1, and applied it in our experiments (L385).
> - Our variance analysis (Sec 4.2, Appendix C.3) suggests that variance may increase if non-uniform weights are used (illustrated empirically in Fig 8 on page 27 of Appendix), motivating the idea of imputing missing annotations. This can reduce variance at the potential cost of increasing bias (L768-770), and in experiments we have empirically observed it to be effective overall. This is an important part of the approach to make it more practical, and in the final version we plan to expand upon this part to make sure readers do not miss it.
>
> **3. “My major concern is about the novelty of the paper”** (see also: [overall response](/forum?id=dsH244r9fA&noteId=jKEOBV6cyp), [fzzo](https://openreview.net/forum?id=dsH244r9fA&noteId=SS3U84sY4C))
>
> In this paper, we establish a formal framework of ‘semi-offline’ evaluation (offline data + counterfactual annotations), and propose a new estimator for this setting based on importance sampling. While the C-IS estimator is simple (using reweighting, similar to e.g. ref [1] in the review), our analysis **reveals new theoretical insights and important practical considerations** for practitioners to keep in mind, e.g. it’s better to impute missing annotations and use equal weights. Therefore, we believe our paper makes non-trivial contributions to the community and is an important step to enabling RL applications in high-stakes domains such as healthcare.
>
> **4. Relationship to OPE with multiple loggers**
>
> This is indeed an interesting connection. We agree with the reviewer that a possible interpretation of counterfactual annotations is “data” from another logging policy i.e., the “annotation policy”. There is also a nice parallel in the policy definitions between Agarwal et al. (ref [2] in the review) vs our paper: in Agarwal et al. (Definition 5.1) the average policy $\pi_{avg}$ is the weighted average of multiple logging policies, whereas our augmented behavior policy (Definition 1, L162) $\pi_{b^+}$ can be seen as the weighted average of the behavior policy and “annotation policy”. However, there are two key distinctions between our framework vs Agarwal et al.: (i) our formulation of annotation is a single number rather than a (sub-)trajectory in the RL setting, (ii) we need to correct for state distributions that a naive approach fails to account for. We will add this discussion to the related works section (currently in Appendix A.1).
>
> **5. Regarding the language around weaker condition**
>
> Thank you for this suggestion. We will change the claim "weaker condition" -> "weaker condition on support" so it is more accurate.

---

> > ### Comment · Reviewer_71Lt · 2023-08-16
> >
> > I thank the authors for explanation and clarification. The rebuttal addressed most of my concerns. I would like to keep my evaluation. I would not increase my evaluation because I still think that the proposed method is not that novel (in particular, weighting seems obvious to me and the authors did not answer how to set the weights theoretically) and that there are no real-world experiments.

---

> > > ### Author Response · Authors · 2023-08-16
> > >
> > > Dear reviewer 71Lt, thank you very much for the response! We are glad that we are able to address most of your concerns and clarify a few misunderstandings. We would like to share a few more thoughts in response to your latest comment:
> > >
> > > - **"weighting seems obvious"**: while in retrospect our weighting approach may be seen as a "straightforward" modification, our paper brings important contributions because
> > >
> > >   (1) we are the first to formalize a new "semi-offline evaluation" task using offline data + counterfactual annotations,
> > >
> > >   (2) under this setting, we point out the bias issue with the naive approach that *many people* may instinctively adopt without reading our paper -- "just add the annotations as new data points" -- and propose a theoretically sound, simple-to-implement solution.
> > >
> > > - **"did not answer how to set the weights theoretically"**: we agree with the reviewer that this is an important question, and we do not claim our paper provides a complete answer to this non-trivial question. Instead, our paper provides useful guidance as to how to use our estimator effectively.
> > >
> > >   - Our experiments show that the C-IS estimator combined with our proposed heuristic (of using equal weights) can reduce both bias and variance, which speaks to the practical utility of our proposed estimator even without a theoretically satisfactory answer.
> > >
> > >   - To further demonstrate non-trivial nature of this problem, in addition to Sec 4 + Appx C theoretical analyses (which lays important groundwork for answering this question theoretically) and subsets of the experiments (Appx E, Fig 7) which showed no $w$ is always best, we will update the appendix to include our analytical derivations (finding the derivative and solve for its zeros) and numerical optimization experiments (using gradient descent) that optimize $w$ to minimize Theorem 13 variance expression when *all* problem parameters are known, with the caveat that in practical problems where not all the problem parameters are known, neither can be applied.
> > >
> > >   - On L311 we stated "We believe that optimizing the weights can further improve OPE performance and is an interesting direction for future work" and we will make sure to emphasize this again in our limitations section.
> > >
> > > - **no real-world experiments**: we have explicitly acknowledged this fact in our limitations section (L519). As pointed out by Reviewers noPk and rPY7, the technical part our paper (formalizing the problem setting and estimator, detailed theoretical analyses, and proof-of-concept experiments) is a self-contained, complete contribution. In our future work, we are excited to collaborate with experts in other research areas (e.g., HCI, healthcare) to carry out human experiments and design methods that can solicit meaningful/useful counterfactual annotations, and we believe those are outside of the scope of the main research question we seek to address in our current paper.
> > >
> > > Lastly, we would like to express our gratitude for your constructive and insightful feedback, and we hope these clarifications can help better contextualize the contributions of our paper.

---

### Author Rebuttal · Authors · 2023-08-09

We thank all reviewers for taking time to read our paper and providing valuable feedback.

Overall, reviewers found our paper to be well-written (71Lt, noPk, fzzo), addressing an interesting (71Lt), important (Qcjd) and timely problem (rPY7), and proposing a novel solution (rPY7) with comprehensive theoretical guarantees (noPk) and thorough experiments (rPY7). Reviewers also commented that the paper has carefully considered limitations and the effects when assumptions are violated (noPk), suggesting our work points to a direction worth exploring (fzzo).

Below we respond to each review with a separate rebuttal, and here we summarize the common themes mentioned by multiple reviewers.

**1. Clarifying our contributions: concerns about “novelty” (71Lt), proposed method is “fairly naive” (fzzo)**

In this paper, we establish a formal framework of ‘semi-offline’ evaluation (offline data + counterfactual annotations), and propose a new estimator for this setting based on importance sampling. While the C-IS estimator is simple, our analysis **reveals new theoretical insights and important practical considerations** for practitioners to keep in mind, e.g. it’s better to impute missing annotations and use equal weights. Therefore, we believe our paper makes non-trivial contributions to the community and is an important step to enabling RL applications in high-stakes domains such as healthcare.

**2. What happens if annotation variance differs from reward variance?** (Qcjd, noPk)

For clarity, in Theorems 6&13 we assume annotation variance $\sigma_G^2$ is the same as reward variance $\sigma_R^2$. More generally, we can assume some relationship between the two such that $\sigma_G(s,a)^2 = \sigma_R(s,a)^2 + \Delta_{\sigma}(s,a)^2$. Then, in the proof of Theorem 13 (Appendix C.3, page 13), the majority of the derivation stays the same except for term (3) where we now apply a different assumption on annotation variance (L712). The resulting variance decomposition will have an additional term $\mathbb{E}\_{s \sim d_1} \mathbb{E}\_{a \sim \pi_b(s)} [\sum\_{\tilde{a} \sim \mathcal{A} \setminus \\{a\\}} \rho^{+}(\tilde{a}|s)^2 \bar{W}(\tilde{a}|s,a)^2 \Delta_{\sigma}(s,\tilde{a})^2]$ related to the difference in variance $\Delta_{\sigma}^2$. Note that this also suggests that when the annotations have a larger variance than the rewards, we may want to assign it a smaller weight, and vice versa -- which makes sense intuitively.

Similarly, the modified version of Theorem 6 will contain an additional third term:

$\mathbb{V}[\hat{v}^{\textup{C*-IS}}] = \mathbb{V}\_{s\sim d_1}[V^{\pi_e}(s)] + \mathbb{E}\_{s\sim d_1} \mathbb{E}\_{a\sim \pi(s)}[\pi_b(a|s) \rho(a|s)^2 \sigma_R(s,a)^2]+ \mathbb{E}\_{s\sim d_1}[\sum\_{\tilde{a} \sim \mathcal{A} \setminus \\{a\\}} \pi_e(\tilde{a}|s)^2 \Delta_{\sigma}(s,\tilde{a})^2]$

Where the original Theorem 6 can be obtained by setting $\Delta_{\sigma}^2$ to 0, making the last term vanish.

We will add the full derivations to the appendix.

**3. Is it even practical to collect counterfactual annotations?** (noPk, rPY7)

Our approach is best suited for high-stakes RL domains where online evaluation is risky, but domain experts can provide additional feedback on new policies. One domain that the authors have the most experience in is healthcare, where the goal is to optimize sequential treatment policies (e.g. Komorowski et al. 2018). In this domain, clinicians are constantly evaluating alternative treatment paths in their minds when making treatment decisions but we only observe what was actually done; counterfactual annotations can be seen as a mechanism to elicit this information in their thought process that is otherwise not recorded. As acknowledged by Reviewer noPk, even without results from a user study, our paper provides valuable contributions to the community, where we formalize a framework for semi-offline evaluation using counterfactual annotations and study the theoretical properties of a new estimator for this setting. Importantly, our work sheds light on desirable properties of useful counterfactual annotations, which points to exciting directions of future research in RL as well as HCI (L530-L534); there is also opportunity to draw lessons from the NLP community about human annotation quality/variability (such as ref [1] mentioned by Reviewer noPk).

Komorowski et al. “The Artificial Intelligence Clinician learns optimal treatment strategies for sepsis in intensive care”. Nature Medicine, 2018.

**4. How to set weights for optimal estimation error reduction?** (71Lt, noPk)

This is indeed an interesting question that arises from our approach, and **in the paper we have provided a partial answer**.
- We explored this empirically in Fig 7 (Appendix E.1, page 27), where we sweep the weights and measure the resulting estimator variance. We found that (L839) “the ideal weighting scheme is problem-specific, and certain weights may lead to higher variance compared to standard IS”. We also found that (L841) “C*-IS, though not always variance-minimizing, consistently achieves lower variance than standard IS”. Unfortunately due to space constraints we couldn’t include the full results and summarized as (L350) “Equal weights (in C*-IS) is a good heuristic though not always ‘optimal’ ” in the text. Since it is an important takeaway of our paper, we will expand upon this part given the one extra content page at camera-ready.
- We also considered this problem analytically (solving for variance-minimizing weights in the bandit case, which turned into a quartic equation) and numerically (optimizing variance objective using SGD). However, unless problem-specific parameters are known, we have yet to come up with a general way to minimize variance. Therefore, we suggest C*-IS (with equal weights) as a heuristic and highlighted this as a direction for future research (L311) that would build upon our semi-offline framework.

---

### Decision · Program_Chairs · 2023-09-21

**Decision:**

Accept (poster)

**Comment:**

This paper is proposing a weighted Importance Sampling based off-policy evaluation that weighs between factual and counterfactual trajectories. They analyze bias-variance properties of two main IS estimators they propose under the assumption that counterfactual annotations are accurate.

Overall, the approach of weighting matches state distributions across factual and counterfactual trajectories and is a fairly natural approach. Concerns raised by reviewers are:

    1. Method not novel enough (intuitive idea)
    2. Assumption of no bias in counterfactual human annotations is unrealistic
    3. No real-world evaluation. The experiments consist of a synthetic bandit setting and a Sepsis healthcare simulator.

Overall, I agree with points 2. and 3. Authors have addressed some of the concerns regarding the assumption of no-bias during discussion. And the submission could be significantly enhanced with better empirical evaluation.

Overall, due to the comprehensive nature of the theoretical analysis, which I encourage authors to add in the final version, and two experiments, I will currently recommend a weak accept.